# What makes Chain-of-Thought Prompting Effective?
# A Counterfactual Study

**Aman Madaan**♠*    **Katherine Hermann**◇    **Amir Yazdanbakhsh**◇*

♠Language Technologies Institute, Carnegie Mellon University
◇Google DeepMind
amadaan@cs.cmu.edu    {hermannk, ayazdan}@google.com

## Abstract

The effectiveness of Chain-of-thought prompting (CoT) has been widely recognized, but the underlying mechanisms behind its success, the reason why *it just works* for a wide range of tasks, remains an open question. To investigate this, we employ a counterfactual prompting approach, systematically manipulating elements of examples used in a few-shot prompt, and testing the consequences on model behavior. This allows us to understand the relative contributions of prompt elements such as symbols (digits, entities) and patterns (equations, sentence structure) on in-context learning. Our experiments with three different large language models (LLMs) reveal several key findings. First, the specific symbols used in the prompt do not significantly impact the model's performance. However, consistent patterns in examples and specifying text in style frequently found on the web are crucial. Second, our findings suggest that the necessity of accurate few-shot examples depends on their role in communicating task understanding. We identify tasks where inaccurate few-shot examples hurt and, surprisingly, tasks where they improve performance. Additionally, we find that the intermediate steps in CoT may not necessarily facilitate learning *how* to solve a task, but instead efficiently convey task understanding (*what*) to the model. Furthermore, CoT leverages LLMs to fill in missing commonsense information, particularly helping difficult reasoning problems and long-tail questions[1].

## 1   Introduction

Large language models (LLMs) have demonstrated remarkable performance in various complex tasks using a small number of examples—a paradigm known as few-shot learning (Brown et al., 2020;

---

* Equal Contributions. Work done when Aman was a student researcher at Google Research, Brain Team (now Google DeepMind).

[1]Code and data available at https://github.com/reasoning-machines/prompt-lib/

Chowdhery et al., 2022). This progress has been significantly boosted by chain-of-thought prompting (CoT) and its variants (Wei et al., 2022b; Kojima et al., 2022; Zhou et al., 2022), which have been proven to further enhance the LLMs' capabilities (Ling et al., 2017; Nye et al., 2021; Cobbe et al., 2021; Patel et al., 2021; BIG-bench Collaboration, 2022).

Despite its demonstrated effectiveness, the underlying mechanisms behind CoT still need to be fully understood. A common explanation draws a parallel with human thinking, in which individuals often reflect on a problem before arriving at a solution (Ling et al., 2017; Wei et al., 2022b,a). While this analogy is intuitive, it does not fully explain the reasons for CoT's success, including when and how the CoT mechanism operates. Since LLMs are trained to predict the next token in a given context, there might be a more systematic explanation behind the successes and failures of CoT. This study aims to explore the mechanism behind CoT, providing insights into its operation.

Our approach involves modifying different components of the examples utilized in the few-shot prompt, and assessing the impact of these changes on the final performance (Figure 1). Specifically, we pinpoint the key elements of an example in few-shot prompting as: **Symbols** (e.g., digits, dates) and **Patterns** (e.g., equations, templates, sentence structure). We then apply *counterfactual prompting* (Goyal et al., 2019) where all components except one are held constant— for instance, replacing symbols like numbers with Greek letters. The effect of each component is then assessed by comparing the performance differences between prompt variations. Our experimental approach spans four diverse reasoning tasks and is implemented across three major language models—PaLM, GPT-3, and CODEX, yielding several surprising findings:

❶ Our study reveals that the specific symbols employed in the prompt have minimal impact on the

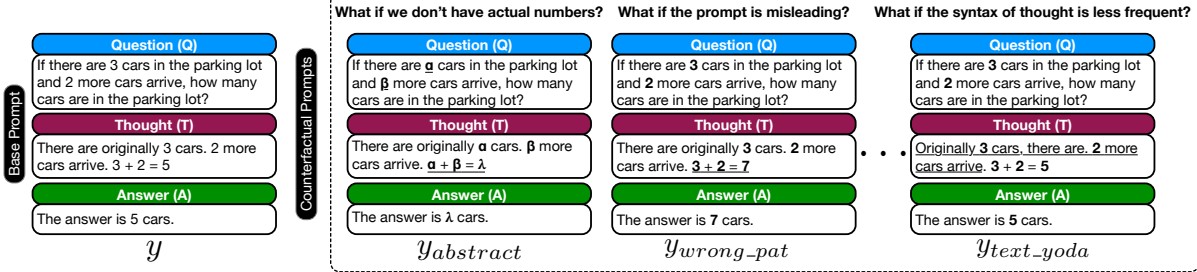

Figure 1: We evaluate the effectiveness of chain-of-thought (CoT) prompting by conducting experiments on diverse datasets, and modifying various aspects of the prompt. By identifying symbols and patterns (templates) specific to each dataset, we create counterfactual prompts and compare their output ($y_{cf}$) to that of the original CoT prompt ($y$). Furthermore, we analyze the attention patterns used by the model when reasoning about the same question using the original and counterfactual prompts, to gain a deeper understanding of the underlying mechanisms.

model's performance. For instance, substituting numbers in the prompt (e.g., 1, 2, 3) with Greek alphabets such as $\alpha, \beta, \gamma$ does not significantly affect the model's performance (Section 3). Nevertheless, maintaining a consistent pattern in examples and specifying a text style commonly seen on the web is crucial (Sections 4 and 5).

❷ Our findings reveal the nuanced role that the accuracy of few-shot examples plays in task understanding within prompts. We discover that the significance of this accuracy is contingent on how much it contributes to comprehending the task at hand. For instance, in tasks such as mathematical reasoning, the presence of incorrect equations doesn't impede performance significantly, primarily because the task objective—solving a mathematical problem—remains clear, regardless of these errors. Conversely, in tasks like sports reasoning, embedding incorrect information in the prompt can obscure task comprehension, negatively impacting performance. This analysis allows us to differentiate between tasks where the accuracy of few-shot examples is pivotal for understanding, and those where it is less critical. Interestingly, we also unearth instances where including incorrect information paradoxically enhances performance by inadvertently facilitating task comprehension (Section 4).

❸ Crucially, we find that symbols and patterns work in unison to bolster CoT in two primary ways: by generating missing information (e.g., through extraction of commonsense knowledge), and reinforcing task understanding (Ouyang et al., 2022) (e.g., by outlining the specific methodology for generating answers). We posit that the successful interplay of symbols and patterns, as facilitated by CoT prompts, plays a more central role in task

success than the model's inherent reasoning capabilities (Section 6)

## 2 Counterfactual Prompting for CoT

**Chain-of-thought prompting.** In the few-shot prompting setup, the input to the model is a prompt, which consists of $k$ in-context examples in the form of $\langle \text{input} \mapsto x_i, \text{output} \mapsto y_i \rangle$ tuples, each of which is related to the target task.

Chain-of-thought prompting (CoT, as proposed by Wei et al. (2022b)) includes an additional intermediate step in the form of a "thought" $t_i$, creating triplets $\langle x_i, t_i, y_i \rangle$. The "thought" $t_i$ describes the intermediate steps and/or results required to derive the output $y_i$ from $x_i$.

For example, given a question such as *John had 6 apples and gave half of them away. How many does he have now?*, instead of directly generating the answer (3), CoT first generates a reasoning step, such as *John had 6 apples and gave away half. Half of 6 = 6 / 2 = 3.* The final answer is then conditioned on this intermediate rationale, and is expected to improve the overall performance of the LLM on the task.

**Counterfactual prompting.** The primary objective of our study is to understand CoT through counterfactual prompting. Each counterfactual prompt $C(p)$ alters only one particular aspect of the in-context examples from the original prompt $p$, while retaining the question asked to the model in its original form. For example, in GSM-8K, a dataset of math word problems (Table 1), we might manipulate the symbol type instantiated in the few-shot examples that appear within a prompt by systematically swapping digits (e.g., 1, 2, 3) for Greek letters (e.g., $\alpha, \beta, \gamma$). This enables us to ask: *what would the model's performance have*

| | |
|---|---|
| **◄ MATHEMATICAL ►** Solve a grade-school level math reasoning problems | |

**Question:** Shawn has five toys. For Christmas, he got two toys each from his mom and dad. How many toys does he have now?
**Thought:** Shawn started with 5 toys. If he got 2 toys each from his mom and dad, then that is 4 more toys. 5 + 4 = 9.
**Symbols:** Numbers: **5**, **4**, **9**
**Patterns:** Equations: **5 + 4 = 9**. The equations typically appear at the end of the thought, and are almost always involved in generating the final answer.

**◄ COMMONSENSE ►** (SPORTS) Verify the accuracy of a statement linking an athlete with a sport.

**Question:** Is the following sentence plausible? "Jamal Murray was perfect from the line."'
**Thought:** Jamal Murray is a basketball player. Being perfect from the line is part of basketball.
**Symbols:** Person and activity: **Jamal Murray**, **Being perfect from the line**
**Patterns:** Consistent sentence structure PERSON *belongs to* SPORT. ACTIVITY *belongs to* SPORT, where *belongs to* is a phrase that connects a sports personality with an activity. The answer is yes if both the person and the activity are associated with the same sport.

**◄ COMMONSENSE ►** (DATE) Reason about dates

**Question:** It is 4/19/1969 today. What is the date 24 hours later in MM/DD/YYYY?
**Thought:** Today is 04/19/1969. 24 hours later is one day after today, which would be 04/20/1969. The answer is 04/20/1969.
**Symbols:** Dates: **04/19/1969**, **04/20/1969**
**Patterns:** Reasoning flows in two steps: **initial calculation** (*Today is 04/19/1969...*), followed by **generation of output** (*The answer is...*)

**◄ SYMBOLIC ►** (SORTING) Sort integers between 1-9

**Question**: 3, 1, 2, 7, 8, 5, 6, 9, 4
**Thought:** 1 < 2 < ... < 9
**Symbols:** Numbers: **2**, **4**, **9**
**Patterns:** Smaller number < larger number (**1 < 2**)

Table 1: **◄ Symbols ►** and **◄ Patterns ►** for different tasks.

*been if all the numbers in the prompt were replaced with symbols?*. By comparing performance on this new version of the prompt $C_{symb\_abs}(p)$ with that of performance on the original prompt $p$, we can learn about the role which using actual digits plays in task performance[2].

**Symbols and Patterns.** In this study, for each dataset, we factor prompts into three distinct components: symbols, patterns, and other surface-level features. For example, in the GSM-8K dataset, symbols are numerical digits (e.g., 5, 4, 2, 13) and patterns (templates) are mathematical equations (e.g., $1 + 2 = 3$). These definitions are chosen to align with the specific characteristics of each dataset to maximize the potential for exciting analysis. Furthermore, our choice of tasks allows us to experiment with various elements of a prompt, such as exploring cases where tasks require explicit patterns (e.g., equations) and implicit patterns (e.g., sentence structures). A detailed description of these components for each task is provided in Table 7 and the corresponding sections of the paper. We also include all the prompts in the Appendix.

**Tasks.** We select tasks for our study based on two criteria: (i) Tasks for which CoT presents ample

improvements over DIRECT prompting as reported in Wei et al. (2022b), and (ii) Tasks that are varied enough to allow us to analyze different symbols, patterns, and text. Consequently, we pick mathematical tasks (GSM-8K, Cobbe et al. (2021)), commonsense tasks (date and sports understanding, BIG-bench Collaboration (2022)), and symbolic tasks (SORTING) as the tasks for our study. For more details on these tasks and datasets, please see Appendix B and Appendix-Table 7. Table 1 provides examples from each dataset.

**Prompts and metrics.** We utilize the same prompts as Wei et al. (2022b) as the base prompts, and modify them for all counterfactual experiments (5-8 examples per prompt). All of the tasks we consider have a single correct answer, and are thus readily evaluated with automated metrics. We use solve rate and accuracy interchangeably to refer to performance on the task.

**Models.** We use LLMs such as PaLM, GPT-3, and CODEX (code-davinci-002), as these models have been shown to perform chain-of-thought reasoning (Wei et al., 2022a) successfully. To provide clear and concise results, we present the results of our experiments using PaLM-62B in the main text. However, to ensure that our findings are not specific to a single model, we also conduct experiments on

---

[2]Appendix-Table 6 outlines a set of major "*what if*" questions that we study in this work.

| Question / Thought | Prompt Type | Solve Rate |
|---|---|---|
| ◄ MATHEMATICAL ► (DIRECT = 10.11%, CoT = 27.37%) | | |
| **Thought:** Shawn started with $\alpha$ toys. If he got $\beta$ toys each from his mom and dad, then that is $\lambda$ more toys. $\alpha + \lambda = \pi$. | $C_{\mathsf{symb\_abs}}(p)$ (Table 25) | 25.70% |
| **Thought:** Shawn started with **5.5** toys. If he got **2.5** toys each from his mom and dad, then that is **5** more toys. **5.5 + 5 = 10.5**. | $C_{\mathsf{symb\_ood}}(p)$ (Table 30) | 28.20% |
| ◄ COMMONSENSE ► (SPORTS) (DIRECT = 71.08%, CoT = 93.67%) | | |
| **Thought:** **Jamal Murray** is a basketball player. **Being ACTIVITY** is part of basketball. | $C_{\mathsf{symb\_abs}}(p)$ (Table 28) | 92.11% |
| **Thought:** **Adair Foster** is a basketball player. **Juggling the paper cups** is part of basketball. | $C_{\mathsf{symb\_ood}}(p)$ (Table 32) | 79.72% |
| ◄ COMMONSENSE ► (DATE) (DIRECT = 31.61%, CoT = 45.18%) | | |
| **Thought:** Today is **DATE**. 24 hours later is one day after today, which would be **DATE**. | $C_{\mathsf{symb\_abs}}(p)$ (Table 24) | 37.41% |
| **Thought:** Today is **04/30/3069**. 24 hours later is one day after today, which would be **04/31/3069**. | $C_{\mathsf{symb\_ood}}(p)$ (Table 31) | 44.50% |
| ◄ SYMBOLIC ► (SORTING) (DIRECT = 46.0%, CoT = 60.6%) | | |
| **Thought:** $\varsigma < \phi < \gamma < \delta < \zeta < \chi < \epsilon < \pi < \upsilon$ | $C_{\mathsf{symb\_abs}}(p)$ (Table 26) | 61.8% |
| **Thought:** **11** < **23** < **34** < **48** < **56** < **63** < **72** < **85** < **95** | $C_{\mathsf{symb\_ood}}(p)$ (Table 33) | 80.0% |

Table 2: Results for counterfactual prompts that replace symbols with abstract symbols or OOD placeholders. DIRECT refers to standard few-shot prompting, without intermediate steps.

a diverse set of publicly available models such as GPT-3 and CODEX. Our results are consistent across all models, as presented in Appendix H.

## 3 Role of Symbols

### 3.1 Method

Symbols refer to token sequences in the prompt which the model employs to reason about the objective of a task and derive the solution. In this work, we adopt natural definitions of symbols for each task. Specifically, we designate numbers in GSM-8K and SORTING, dates in DATE (e.g., 01/01/2021), and athletes (e.g., Michael Jordan) and activities (e.g. dunked the ball) in SPORTS as symbols.

To understand how symbols help the model understand a target task, we devised a collection of counterfactual prompts manipulating symbols of various types. This subsection outlines the methods for two main experiments conducted using these counterfactual prompts for symbols: (a) "*Abstract*" → substituting the symbols with abstract values and (b) "*Out-of-Distribution*" → replacing the symbols with the ones that were not encountered during the training phase.

**Abstract symbols** $\ulcorner C_{\mathsf{symb\_abs}}(p) \urcorner$. In this variant, we substitute symbols with an abstract placeholder. For example in SPORTS, we use the placeholder ACTIVITY to represent a sports activity "scoring a three pointer".

**OOD symbols** $\ulcorner C_{\mathsf{symb\_ood}}(p) \urcorner$. These prompts investigate the effects of incorporating out-of-

distribution (OOD) symbols within prompts. Specifically, we examine the consequences of replacing integers with fractions in GSM-8K, substituting sportspersons with random names in SPORTS, and altering dates to dates beyond the year 3000 AD in DATE.

In our experiments, we manipulate the symbols only in the few-shot examples that appear within a prompt. The question asked to the model at the end of the prompt remains the unaltered original in all conditions. This approach ensures that the model must generalize from the modified symbol type used in the few-shot examples to the original symbol type used in the question, allowing us to evaluate the model's ability to extrapolate its understanding from manipulated context.

### 3.2 Results

This subsection discusses the results from the experiments described above, as presented in Table 2. The statistical significance test for each counterfactual experiment is detailed in Appendix H.

**Abstract symbols** $\ulcorner C_{\mathsf{symb\_abs}}(p) \urcorner$. The results in Table 2 illustrate that substituting symbols with abstract placeholders has little to no impact on model performance. In addition, we observe that in most cases the answers does not contain abstract symbols, however, the exact fraction is dataset dependent. For example, outputs for GSM-8K contain abstract symbols (Greek alphabets) in 12.91% of the cases, whereas abstract symbols (placeholders for names and activities) are not present in the out-

(a) Vanilla CoT Prompt $p$.

(b) Abstract Symbols $C_{symb\_abs}(p)$.

Figure 2: Average attention per token ($0^{th}$ layer, averaged across heads) for the same question using (a) vanilla CoT prompt $p$ and (b) $C_{symb\_abs}(p)$. Both $p$ and $C_{symb\_abs}(p)$ generate the correct answer, relatively attending to same tokens. The phenomenon holds for higher layers (Appendix G).

put for SPORTS. Overall, in cases where the model does generate Greek alphabet symbols, the model regardless reaches the correct answer in 36.5% of the cases.

**OOD symbols** $\ulcorner C_{\mathsf{symb\_ood}}(p)\urcorner$**.** Table 2 demonstrate that substituting symbols with OOD symbols does not have a significant effect on the model's performance when solving GSM-8K and DATE. However, for SPORTS, using OOD symbols significantly influences the model's task understanding, leading to a noticeable decrease in performance. In the SORTING task, we observed a significant improvement in the task solve rate (from 60.6% to 80.0%) by using larger integers in the prompts. These results underscore the importance of providing the model with clear and diverse understanding of the target task.

However, for SPORTS, using OOD symbols significantly influences the model's task understanding, leading to a noticeable decrease in performance. We change a known player's name, **Jamal Murray**, to a fabricated name, **Adair Foster**. We also switch a typical basketball action, **Being AC-TIVITY**, with something unrelated like **Juggling the paper cups**. These changes make it hard to tell that the task is about figuring out if a sports person and an activity are related, leading to a drop in accuracy (Table 2).

**Attention analysis.** Beyond corpus-level statistics like accuracy and agreement scores, it is useful to examine whether the model's behavior is consistent for the same input question across different versions of a prompt (instantiated with different symbol types). To understand this, we analyze the attention patterns for randomly sampled questions using both CoT and $C_{\mathsf{symb\_abs}}(p)$. The results, presented in Figure 2, show the attention patterns for a random question from GSM-8K using $p$ and $C_{\mathsf{symb\_abs}}(p)$. The attention patterns are similar for both prompt instantiations, indicating a consistent reasoning mechanism. We include details on the attention score calculation, per-layer heatmaps, and the limitations of relying solely on attention analysis for model interpretation in Appendix G.

## 4 Role of Patterns

In this study, we define a pattern as a template that is present in all examples within a given task. The role of patterns is mainly to guide the task towards reaching its objective and hence can manifest differently depending on the task. For example, in GSM-8K patterns are structural, whereas in SPORTS patterns represent specific rules.

### 4.1 Method

We have identified the patterns for each task and they are summarized as follows:

- **GSM-8K** → Equations (e.g., $2 + 2 = 4$).
- **SPORTS** → The pattern is a consistent thought structure in the following form: "*person is a sport$_1$ player. activity is part of sport$_2$*". The answer is yes, if **sport$_1$** and **sport$_2$** are the same.
- **DATE** → As in SPORTS, the pattern here is the consistent thought structure. Each thought contains two parts: (a) ⟨**calculation**⟩ in which the information from input (e.g., question) is restated, and intermediate results are derived (e.g., "One day after 06/01/1943 is 06/02/1943") and (b) ⟨**output**⟩ in which the final answer is generated based on the intermediate results (e.g., "10 days before today is 05/23/1943").
- **SORTING** → The thought lists numbers in a sorted order *(1 less than 2)*.

We alter different aspects of patterns within the few-shot prompts to study their importance. Similar to our approach with symbols, all these alterations are made strictly within the prompts,

while keeping the input questions unaltered (i.e., no changes are made to the task).

**Inconsistent pattern.** In $C_{\texttt{pat\_inconsistent}}(p)$ we assess the sensitivity of model performance to the usage of *inconsistent* patterns. For GSM-8K, we construct $C_{\texttt{pat\_inconsistent}}(p)$ by exclusively removing everything except equations. However, in SPORTS, patterns are implicit in the sentence structure (*person is a sport$_1$ player, activity is part of sport$_2$*), making it challenging to create a $C_{\texttt{pat\_inconsistent}}(p)$ scenario. To overcome this, we devise a prompt that incorporates multiple variations of thought. For example, in some cases, we phrase the thought by listing activity first: " *activity is part of sport$_2$* , *person is a sport$_1$ player.* " This methodology effectively eliminates the model's reliance on specific patterns, essentially creating a virtual equivalent of the $C_{\texttt{pat\_inconsistent}}(p)$ setup. We apply similar techniques to DATE and SORTING.

**Pattern-only.** In $C_{\texttt{pat\_only}}(p)$ prompts, we modify the thoughts by preserving solely the essential information conveyed by the patterns. For example, in GSM-8K, the pattern-only prompts exclusively contain mathematical equations. In SPORTS, the pattern strives to establish a connection between a person and an activity, based on whether they involve the same sport (in affirmative cases) or different sports (in negative cases). The $C_{\texttt{pat\_only}}(p)$ prompts retain this information by distilling the thought to "*both are part of the same/different sport*". Similarly, in DATE, we construct thoughts that retain the calculation and answer generation. For example, the statement *the date today is 04/19/1969, there are 24 hours in a day* is transformed into *today = 04/19/1969, 24 hours = day,* where the second expression only provides the answer equation.

**Wrong pattern.** In $C_{\texttt{pat\_wrong}}(p)$, we examine prompts that include misleading or incorrect information while following the standard pattern. For instance, we use incorrect equations for GSM-8K, erroneous date calculations for DATE, and improper ordering for SORTING. Similarly, for SPORTS, we associate a sportsperson and activity with a randomly chosen sport, instead of the correct one. The goal of this experiment is to evaluate the role factual information in the prompt plays in model's ability to generate correct responses.

## 4.2 Results

**Inconsistent pattern.** The use of inconsistent patterns in the $C_{\texttt{pat\_inconsistent}}(p)$ method had a noticeable impact on performance. For instance, in mathematical tasks, the solve rate was 21.46% (Table 3), significantly lower than the 27.37% achieved by CoT. In SPORTS tasks, the solve rate was 79.01%, as compared to CoT's 93.67%. Despite being able to derive relevant facts such as "Nick Foles is a football player" and "the puck is a part of ice hockey," the model failed to utilize these facts to produce correct answers.

**Pattern-only.** Results from the $C_{\texttt{pat\_only}}(p)$ method demonstrated that preserving only the patterns in prompts led to a reduced performance. For mathematical tasks, the solve rate was only 10.01% (Table 3), significantly lower than the 27.37% solve rate of CoT. Similarly, in SPORTS tasks, the solve rate achieved was 74.13%, as opposed to the 93.67% solve rate of CoT. This underscores the importance of the contextual information that accompanies the patterns for optimal performance.

**Wrong pattern.** Introducing incorrect patterns in the $C_{\texttt{pat\_wrong}}(p)$ method led to varying impacts on performance depending on the task. In mathematical tasks, $C_{\texttt{pat\_wrong}}(p)$ achieved a solve rate of 24.39%, nearly the same as the 27.37% solve rate of CoT (Table 3). Likewise, for DATE tasks, the solve rate achieved by $C_{\texttt{pat\_wrong}}(p)$ was 44.84%, closely comparable to CoT's 45.18%. However, for SPORTS tasks, the solve rate sharply declined to 46.02%, which was considerably lower than the 93.67% solve rate of CoT. These results indicate that incorrect patterns can greatly skew the model's understanding, especially for tasks like SPORTS where correct associations between the subject and activity are crucial.

## 5 Additional Surface-level Manipulations

In addition to symbols and patterns, we delve into surface-level manipulations of text. These manipulations encompass changes to tokens that do not directly contribute to task-specific semantics but may nonetheless impact a language model's understanding and performance. In this section, we scrutinize the effects of these surface-level alterations in our prompts and examine their influence on the outcomes.

**Text with altered grammatical style.** First, we examine the impact of Yodish, a syntactically valid

| Question / Thought | Prompt Type | Solve Rate |
|---|---|---|
| ◄ **MATHEMATICAL** ► (DIRECT = 10.11%, CoT = 27.37%) | | |
| **Thought:** Shawn started with 5 toys. If he got 2 toys each from his mom and dad, then that is 4 more toys. | $C_{\texttt{pat\_inconsistent}}(p)$ (Table 39) | 21.46% |
| **Thought:** 5 + 4 = 9. | $C_{\texttt{pat\_only}}(p)$ (Table 40) | 10.01% |
| **Thought:** Shawn started with 5 toys. If he got 2 toys each from his mom and dad, then that is 4 more toys. **5 + 4 = 7.** | $C_{\texttt{pat\_wrong}}(p)$ (Table 37) | 24.39% |
| ◄ **COMMONSENSE** ► (**SPORTS**) (DIRECT = 71.08%, CoT = 93.67%) | | |
| **Thought:** Jamal Murray and being perfect from the line are both part of basketball. | $C_{\texttt{pat\_inconsistent}}(p)$ (Table 45) | 79.01% |
| **Thought:** Both are part of the **same sport**. | $C_{\texttt{pat\_only}}(p)$ (Table 41) | 74.13% |
| **Thought:** Jamal Murray is a soccer player. Being perfect from the line is part of soccer. | $C_{\texttt{pat\_wrong}}(p)$ (Table 46) | 46.02% |
| ◄ **COMMONSENSE** ► (**DATE**) (DIRECT = 31.61%, CoT = 45.18%) | | |
| **Thought:** Today is 04/19/1969. | $C_{\texttt{pat\_inconsistent}}(p)$ (Table 44) | 34.19% |
| **Thought:** ⟨calculation⟩ Today = 04/19/1969. 24 hours = 1 day. ⟨output⟩ 04/19/1969 + 1 = 04/20/1969. | $C_{\texttt{pat\_only}}(p)$ (Table 42) | 33.52% |
| **Thought:** ⟨calculation⟩ Today is 04/19/1969. 24 hours later is one day after today, which ⟨output⟩ would be 03/20/1969. | $C_{\texttt{pat\_wrong}}(p)$ (Table 36) | 44.84% |
| ◄ **SYMBOLIC** ► (**SORTING**) (DIRECT = 46.0%, CoT = 60.6%) | | |
| **Thought** 9 > 8 > 7 > 6 > 5 > 4 > 3 > 2 > 1 | $C_{\texttt{pat\_inconsistent}}(p)$ (Table 43) | 45.0% |
| **Thought:** — (similar to DIRECT) | $C_{\texttt{pat\_only}}(p)$ | 46.0% |
| **Thought:** 1 < 2 < 3 < 4 < 7 < 6 < 5 < 8 < 9 | $C_{\texttt{pat\_wrong}}(p)$ (Table 47) | 64.8% |

Table 3: The accuracy of patterns is not always important, but their absence could be catastrophic. Please note that the $C_{\texttt{pat\_inconsistent}}(p)$ prompts have examples in multiple formats, and we only show one here due to space constraints.

| Question / Thought | Solve Rate |
|---|---|
| ◄ **MATHEMATICAL** ► (DIRECT = 10.11%, CoT = 27.37%) | |
| With 5 toys, Shawn started. 2 toys each from his mom and dad, if he got, then that is 4 more toys. 5 + 4 = 9. | 23.22% |
| ◄ **SPORTS** ► (DIRECT = 71.08%, CoT = 93.67%) | |
| A basketball player, Jamal Murray is. Perfect from the line, is part of basketball being. | 68.26% |
| ◄ **DATE** ► (DIRECT = 31.61%, CoT = 45.18%) | |
| 04/19/1969, today is. Later is one day after today, 24 hours, 04/20/1969, which would be. | 30.75% |

Table 4: Modifying the style of text in the prompts to *Yodish* severely impacts DATE and SPORTS, which rely on the model to generate output in a specific format.

but non-standard style of English, on the model's performance (Kaminski, 2011; Pullum, 2005). In Yodish, the *XSV* sentence structure is prevalent, where *X* is a phrase that complements the verb *V*, and *S* is the subject. For example, the sentence "Bryce Harper is a baseball player" would be re-arranged in Yodish as "A baseball player, Bryce Harper is". This style presents a greater challenge for the model, as it is less frequently encountered in typical training data. This makes it a valuable test case for evaluating how textual structure influences model performance. We experiment with three variations of prompts: (a) $C_{\texttt{text\_yodathoughts}}(p)$: thoughts, (b) $C_{\texttt{text\_yodaquestions}}(p)$: questions, and

(c) $C_{\texttt{text\_yoda}}(p)$: both questions and thoughts in Yodish. As shown in Table 4, this style has varying effects on model performance, from moderate (GSM-8K) to significantly negative (SPORTS and DATE). For example, in SPORTS, the use of Yodish encourages the model to generate the sport (the object) at the start of the sentence. This structure, while grammatically correct, forces the model to process information in a manner closer to direct prompting, as the model has to output the answer *before* the reasoning process.

Using CODEX as the base model, we performed additional experiments with other forms of less common patterns in standard English grammar: (a) passive voice and (b) nested clause. Specifically, we modified the prompts to rewrite the original thought (e.g., *Shawn started with 5 toys. If he got 2 toys each from his mom and dad, then that is 4 more toys. 5 + 4 = 9.*) in passive voice (*5 toys were initially had by Shawn. 4 more toys were received by him, 2 each from his mom and dad. 5 + 4 is 9*) and using nested clauses (*Given that Shawn had 5 toys, and considering he received 4 more from his parents, the total is 9 toys*).

Both these variations led to a drop in performance. For passive voice the solve rates for GSM-8K, SPORTS, and DATE dropped to 53.0% (-12.6%), 90.3% (-8.0%), and 65.9% (-3.3%) respec-

tively. For thoughts written with nested clauses, the solve rates decreased to 55.5% (-10.1%) for GSM-8K, 90.3% (-8.0%) for SPORTS, and 66.4% (-2.8%) for DATE. These results indicate that even within standard English grammar using less common variations cause discernible drops in task solve rates.

**Shuffled and random thoughts.** Finally, we test the impact of altering the order or context of the text. We experiment with random thoughts ($C_{\texttt{text\_rand}}(p)$), where each thought is replaced by a semantically correct but randomly chosen thought from another example, and shuffled thoughts, where we either shuffle the words within a thought ($C_{\texttt{text\_inter\_shuf}}(p)$) or across thoughts ($C_{\texttt{text\_intra\_shuf}}(p)$). These experiments were designed to assess the model's dependency on the logical sequence and context of the text. The considerable decrease in performance indicates the model's reliance on coherent and contextually relevant information. The performance decrease was significant across all models, suggesting a universal dependency on coherent and contextually appropriate text among these models.

## 6 What makes Chain-of-Thought prompting work?

In this section, we summarize our findings and present some key takeaways.

**Takeaway I.** COT helps in reinforcing task understanding.

Symbols and patterns can be significantly altered as long as they communicate task intent (*what* has to be done). In some cases (such as SORTING), deviating from standard patterns may be beneficial if they more effectively communicate the task. While few-shot prompting is often called in-context learning, our findings indicate that prompts serve more as a means of *reminding* the model of the task that needs to be solved.

**Takeaway II.** COT helps in eliciting common-sense knowledge.

Examples in a COT prompt share a key property: they help to fill in the information in the prompt. For instance, in ➥ $\mathcal{Q}3$ of Appendix-Table 16, the model with COT infuses commonsense knowledge

| Task | PaLM-62B | GPT-3 | CODEX | PaLM-540B |
|------|----------|-------|-------|-----------|
| GSM (Table 73) | +6.2% | +11.7% | -4.7% | +5.7% |
| DATE (Table 74) | +14.8% | +12.8% | +1.1% | +5.7% |
| SPORTS (Table 75) | +1% | +16.6% | +0.2% | +2.1% |
| SORTING (table 23) | +9% | +273% | +268% | +24.5% |

Table 5: Comparison (realtive performance difference) of COT and CCOT solve rates.

about animals (e.g., "*5 dogs have 4 legs each . . . 2 cats have 4 legs each . . . 10 birds have 2 legs each*"). For DATE, the model articulates the exact date for "*Christmas Eve*" in the generated thought (➥ $\mathcal{Q}6$-◆ $\mathcal{T}^{\ulcorner}$COT$^{\urcorner}$ (Appendix-Table 16). Further, examples typically repeat known information (e.g., name of the athlete) before generating new, helpful information (name of the sport) conditioned on known information. We find that hampering this property of the prompt, either by removing patterns (Section 4) or altering text structure (Section 5), hurts performance (Appendix-Table 16).

**Takeaway III.** COT helps difficult examples and long-tail questions.

We analyze cases where COT is exclusively correct (i.e., COT is correct and DIRECT prompting is wrong). For GSM-8K, we find that the average number of entities in questions solved exclusively by COT is 3.98 compared to the overall average of 3.62, a statistically significant difference (difference of means t-test $p = 0.04$). Similarly, in the SPORTS dataset, we find that COT is exclusively correct for rare entities: the average number of Google search results for activities for which COT *exclusively* yields correct answers is $\approx 52\times$ lower compared to the ones for which $C_{\texttt{pat\_only}}(p)$ exclusively lands correct answers Appendix-Table 18.

**CCOT: Concise Chain Of Thought** A central observation of our study is that when few-shot examples effectively convey the task, LLMs can successfully harness them. Moreover, a consistent pattern within the prompts that aligns with the task significantly enhances the model's ability to generate the correct response. To investigate whether a consistent pattern and the ability to fill in missing information are sufficient, we create a concise version of the Chain of Thought (COT) prompts,

named CCoT, that retain the essential information while removing unnecessary tokens.

For GSM-8K, we randomly select questions from the training set whose thoughts are shorter than CoT. For SPORTS, a thought such as *Jamal Murray is a basketball player. Being perfect from the line is part of basketball* was streamlined to *Jamal Murray → basketball. perfect from the line → basketball*. Similarly, in *Today is 04/19/1969. 24 hours later is one day after today, which would be 04/20/1969* was converted to *Today is 04/19/1969. 24 hours (one day) later is 04/20/1969.* Table 5 shows that CCoT outperforms CoT while using prompts with fewer tokens. The task solve rate of CCoT remains relatively high as we scale the model to the large version, highlighting the efficiency of CCoT. Additionally, we find that CCoT reduces the input and output tokens by 1.39 and 1.58 times, respectively. We provide additional results and links to each prompt in Table 19.

# 7  Related Work and Discussion

This paper intersects with a growing body of work on prompting and large language model reasoning (Brown et al., 2020; Chowdhery et al., 2022; Scao et al., 2022; Zhang et al., 2022; Dasgupta et al., 2022).

**Role of accurate few-shot examples.** Min et al. (2022) find that label correctness is not crucial for the success of the models, and even random labels might lead to competitive performance. Building on this work, Kim et al. (2022) find that the role of the correctness of the labels might be task-dependent. A concurrent body of work has also explored the reasons behind the effectiveness of chain-of-thought-prompting and shows that even wrong CoT prompts can lead to strong performance (Ye et al., 2022; Wang et al., 2022). Our findings complement and concur with the findings of these works and is mutually reinforcing, but go beyond the notion of the accuracy of examples. Specifically, we manipulate various aspects of symbols and patterns (with correctness being one of the aspects) to examine their role in the success of CoT. Further, in addition to comparing the final results (outcome), we also focus on the mechanism (attention patterns). While Wang et al. (2022) primarily evaluates the effectiveness of CoT for reasoning and question-answering tasks by introducing drastic changes to prompts to illustrate invalid reasoning, our work adopts a broader approach.

We introduce counterfactual studies encompassing both subtle and significant modifications to CoT. Specifically, we assess CoT's effectiveness under two scenarios: (1) where the reasoning flow remains but with incorrect symbols and patterns, and (2) where the reasoning flow is intentionally disrupted, such as through prompt shuffling or the introduction of random texts.

In a related vein, Ye et al. (2022) investigates the effects of incorrect calculations and word omission/masking in CoT. Our work extends this by exploring the influence of out-of-distribution (OOD) symbols, inconsistent patterns, exclusive use of symbols or patterns, and varied grammatical styles. **Few-shot learning or few-shot reminding?** Our results resonate with the work of Reynolds and McDonell (2021); Ouyang et al. (2022), who found that one of the key roles played by the prompt is to *remind* the model of the underlying task. Finally, Xie et al. (2021) show that in-context learning enables a large model to infer a *shared concept* between the examples, possibly leading to better task understanding. Our studies on the role of prompt, especially examples where wrong examples lead to better output (e.g., for SORTING), provide further empirical evidence for these findings. Finally, in concurrence with Razeghi et al. (2022)'s finding that pre-training term frequencies partly account for the success of few-shot methods, our experiments on SPORTS shows that CoT method is helps difficult questions involving personalities and activities less commonly found on the web.

# 8  Conclusions

Our study suggests that the underlying mechanisms behind the effectiveness of Chain-of-thought prompting (CoT) may be more complex than previously thought. We find that even large substitutions, like replacing the digits in few-shot examples with Greek letters, do not affect model performance. However, simple grammar or word-order changes can have catastrophic effects. These results, along with other findings, suggest that the effectiveness of CoT may stem from its ability to efficiently convey task understanding (*what*) to the LLM. Our results indicate that a combination of consistent, easy-to-mimic patterns (templates) and a strong LLM that can fill missing commonsense is the recipe for effective CoT. We hope to use this research to develop better prompting techniques and more robust language models for various tasks.

## Limitations

This work investigates mechanisms that enable the effectiveness of chain of thought techniques in large language models. However, this study does not delve into the underlying interactions between the layers or devise theoretical formulations of the models' reasoning capability, mainly due to the complexity and depth of these models, which hinder faithful probing. Instead, we leverage counterfactual probing, a tractable approach for understanding the behavior of large language models.

**Limitations of counterfactual prompting.** Counterfactual examples can provide valuable insights into the behavior of language models, as they allow for identifying and collecting prompts that are critical for generating respective outputs. However, it is essential to note that relying solely on counterfactual examples can be misleading (Laugel et al., 2019; Slack et al., 2021). In this work, we focus on counterfactual examples that exhibit consistent and systematic performance divergence to better understand the failure modes and strengths of the model. We also analyze attention patterns to supplement our findings. We neither rely on the results that do not exhibit such characteristics, nor reject prompts that pose contradictory observations. We discuss additional limitations of our approach in Section 8.

**Spurious Correlations** While this approach has its advantages, there are limitations to consider. The counterfactual approach assumes that the model's behavior can be understood by analyzing its output given a specific input. However, there may be uncharted and baffling artifacts that the model could be exploiting (McCoy et al., 2019; Geirhos et al., 2020), leading to potentially misleading observations. For instance, there is a potential for spurious correlations between symbols, patterns, text, and the outcome, which can lead to false conclusions. Our exhaustive empirical study addresses some of these concerns by providing in-depth analysis and methodical measures to ground our hypotheses. Additionally, the discrete and multiplicative nature of language understanding tasks implies that no study can be completely thorough.

**Limited Task and Dataset Scope** This work is also limited to a subset of common tasks and datasets, including math (Cobbe et al., 2021), commonsense reasoning (BIG-bench Collaboration, 2022), and symbolic reasoning. Our conclusions may not apply to other reasoning tasks. Despite these limitations, we hope that this work sheds light on the ability of large language models to solve complex reasoning tasks.

**Model availability.** In our experiments, we use three different language models: PaLM, GPT-3 (text-davinci-002), and CODEX (code-davinci-002). While PaLM is not publicly available at the time of submission, the provided source code is compatible with OpenAI API v0.23.0 and can work with any OpenAI model. However, using closed models like PaLM may limit our results' reproducibility and hinder our findings' generalizability to other models. Additionally, our results may not be directly comparable to other studies that use different models, as the behavior of models may vary across architectures and training datasets. This limitation should be considered when interpreting the results of our study.

## Acknowledgements

We thanks the anonymous reviewers for their useful comments and suggestions. We would like to extend our gratitude towards Kathy Meier-Hellstern, Denny Zhou, Victor Veitch, Saleem Abdulrasool, Shruthi Sukumar, Milad Hashemi, Douglas Eck, Christian Szegedy, Cliff Young, Yiming Yang, and James Laudon for their invaluable feedback and support. We also thank the PaLM team and our extended team at Google Research, Brain Team (now Google DeepMind) who enabled this research and helped us conduct our experiments.

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

# Part I

# Appendix

## A Reproducing the Results with Publicly Available Models

We take the following steps to enable the reproducibility of our work.

**Controlling for randomness due to the order of examples.** We run each experiment with multiple random seeds to control for randomness because of the order of examples in the prompt. We report the average and standard deviation of the results across all the random seeds. Additionally, we conduct statistical significance tests (McNemar's test (McNemar, 1947)) to compare the results across different prompts. Finally, we evaluate the agreement in output generated by different models using Cohen's kappa ($\kappa$) metric.

**Experiments with publicly available models.** We experiment with three different language models: PaLM, GPT-3 (text-davinci-002), and CODEX (code-davinci-002). PaLM is not publicly available as of submission time, but the provided source code is compatible with OpenAI API v0.23.0, and can work with any OpenAI models. Finally, CODEX is free to use as of submission time that further helps with the reproducibility of the results.

All the prompts are included in the *prompt_lib/prompts/* directory in the code repository.

## B Details on Studied Reasoning Tasks

In this work, we evaluate counterfactual prompting on the following reasoning tasks:

1. ◄ MATHEMATICAL ► We experiment with GSM-8K (Cobbe et al., 2021) (1319 samples). The dataset contains math word problems geared toward an average middle-school curriculum.
2. ◄ COMMONSENSE ► We use date understanding (DATE, 349 samples) and sports understanding (SPORTS, 980 samples) as representative tasks for commonsense reasoning, both derived from BIG-bench Collaboration (2022).
3. ◄ SYMBOLIC ► We experiment with sorting (SORTING, 500 samples) a list of single-digit integers. We do not associate explicit instruction (e.g., *sort these numbers*) with the questions. Instead, we frame the questions as a challenging setup in which the model should figure out the task and the requisite information to solve it.

## C Computational Resources and Models

In this work, we neither train any of the PaLM models, nor performs finetuning. We solely perform inference on PaLM variants using TPU v4 (Jouppi et al., 2021, 2017). For PaLM-62B, we use $4\times4\times4$ TPU v4 configuration, whereas, for PaLM-540B we use $4\times4\times16$ mesh configuration. To account for the variation in results caused by the order of examples in the prompt, we conduct each experiment three times, each with different seeds, and report the average task solve rate. Following Wei et al. (2022b), we evaluate each task using accuracy i.e. fraction of examples where the output matched the expected result.

**Public large language models.** We use OpenAI API[3] to conduct experiments with GPT-3 (text-davinci-002) and CODEX (code-davinci-002).

## D FAQ

**Q: Are the definitions of symbols and patterns universal?** **A:** It is possible that there can be other ways to define symbols, patterns, and text in the context of Chain-of-thought prompting (CoTp). Our characterization of these components is not meant to be universal, and there may be additional properties and perspectives that future research can explore. However, the goal of our study is not to provide a universal definition but to make practical and reasonable distinctions that allow us to manipulate each component individually while keeping the others fixed. This approach allows us to better understand the underlying structure of prompts in different contexts. We acknowledge that the impact of symbols and patterns on CoTp success is task-dependent, and our results provide a diverse set of insights. Finally, while there is not a universal definition of symbols and patterns, for any given task we have aimed to provide a clear and reasonable characterization.

**Q: What is the TLDR?** **A:** The effectiveness of CoT in few-shot learning with large language models is due to two factors: it helps reinforce task understanding and fills in the missing information. The traditional notion of in-context learning may need to be reevaluated as the model may be using the few-shot examples to be reminded of the task (Reynolds and McDonell, 2021).

---

[3]https://openai.com/blog/openai-api/, v0.23.0

Table 6: Examples of "*what if*" questions that we seek to answer in this work.

**Q1.** *What if* we replace all the symbols in the prompt with abstract placeholders, can the required task still be discerned?

**Q2.** *What if* the examples in the prompt were incorrect, will it affect the correctness of the outputs?

**Q3.** *What if* we remove all patterns from the input, will CoT continue to be effective?

**Q4.** *What if* the linguistic style of the prompt was different than that of the questions, will it hamper the performance?

Table 7: Examples of tasks used in this work. The ➡ $\mathcal{Q}$ ↦ question, ◈ $\mathcal{T}$ ↦ thought, and ◆ $\mathcal{A}$ ↦ answer are separately highlighted. For the complete list of vanilla CoT for each category refer to Table 20, Table 22, Table 21, and Table 23, respectively.

| ◄ **MATHEMATICAL** ► |
| --- |
| ➡ $\mathcal{Q}$ ↦ Shawn has five toys. For Christmas, he got two toys each from his mom and dad. How many toys does he have now? |
| ◈ $\mathcal{T}$ ↦ Shawn started with 5 toys. If he got 2 toys each from his mom and dad, then that is 4 more toys. 5 + 4 = 9. |
| ◆ $\mathcal{A}$ ↦ The answer is 9. |
| ◄ **COMMONSENSE** ► (**SPORTS**) |
| ➡ $\mathcal{Q}$ ↦ Is the following sentence plausible? "Jamal Murray was perfect from the line." |
| ◈ $\mathcal{T}$ ↦ Jamal Murray is a basketball player. Being perfect from the line is part of basketball. |
| ◆ $\mathcal{A}$ ↦ The answer is yes. |
| ◄ **COMMONSENSE** ► (**DATE**) |
| ➡ $\mathcal{Q}$ ↦ It is 4/19/1969 today. What is the date 24 hours later in MM/DD/YYYY? |
| ◈ $\mathcal{T}$ ↦ Today is 04/19/1969. 24 hours later is one day after today, which would be 04/20/1969. |
| ◆ $\mathcal{A}$ ↦ The answer is 04/20/1969. |
| ◄ **SYMBOLIC** ► (**SORTING**) |
| ➡ $\mathcal{Q}$ ↦ 2, 4, 3, 8, 9, 6, 7, 1. |
| ◈ $\mathcal{T}$ ↦ 1 < 2 < 3 < 4 < 5 < 6 < 7 < 8 < 9. |
| ◆ $\mathcal{A}$ ↦ The answer is 1 , 2 , 3 , 4 , 5 , 6 , 7 , 8 , 9 |

**Q: What are some key findings? How can they help understand few-shot prompting better? A:** We summarize key takeaways in Section 6. One of the main conclusions is the shift in perspective on how prompts should be viewed - rather than being used as a method of teaching the model, they should be seen as a way to remind the model of the task at hand. A helpful analogy is that of search engines; just as query expansion is a useful tool for retrieving relevant results, CoT is helpful in extracting meaningful answers from a model.

**Q: Why the long appendix? A:** Our study includes experimentation with over 20 counterfactual prompts for 4 datasets and 3 models. Due to space limitations and for clarity purposes, only a subset of the results is included in the main paper. The Appendix includes all the results and additional insights, such as additional attention analysis. Every task reveals unique insights about CoT and prompting. However, the empirical results alone may not convey all necessary information, so a comprehensive qualitative analysis is included in the Appendix. Additionally, the prompts take up a significant portion of space, which are also included in the Appendix.

**Q: Where are additional details on Attention patterns? A:** Appendix G shows additional details, including details on the attention score calculation. Note that the attention analysis was only possible for PaLM, because that was the only model we had weight-level access to.

# E Extended Background

**Chain of thought prompting**. This work broadly investigates the premise of in-context few-shot prompting in large language models (LLM). In these methods, the input to the model is a prompt $p$ consisting of $k$ in-context examples in the form of ⟨input ↦ $x_i$, output ↦ $y_i$⟩ tuples [4]. Each ⟨$x_i$, $y_i$⟩ alludes to the target task. For example, in math solving problems (Cobbe et al., 2021), an input is math question (*If three apples were added to a basket which had two apples, how many apples are in the basket now?*), and the output supplies the answer (*5*). Wei et al. (2022b) additionally sup-

---

[4]The number of ⟨$x_i$, $y_i$⟩ tuples depend on the maximum input sequence length of the model, typically $k \leqslant 10$.

plement in-context few-shot prompting with *Chain Of Thought* (CoT) method, improving the performance of LLM in solving several reasoning tasks. In particular, CoT additionally prefixes each output with a *thought*, creating triplets $\langle x_i, t_i, y_i \rangle$. The "chain of thought" $t_i$ describes the intermediate steps and/or results required to derive the output $y_i$ from $x_i$. Therefore, the prompt is assembled in the form of $p \equiv \langle x_1 \cdot t_1 \cdot y_1 \rangle \| \langle x_2 \cdot t_2 \cdot y_2 \rangle \| \ldots \| \langle x_k \cdot t_k \cdot y_k \rangle$, where "·" and "‖" are indicator symbols. The role of · is to separate elements of an example, whereas ‖ indicates the boundary of an example. The intuition behind chain of thought prompting is that catering the outputs/answers with intermediate steps/results present additional in-context information to the model (Ling et al., 2017; Amini et al., 2019; Chen et al., 2021a; Cobbe et al., 2021; Nye et al., 2021). This additional in-context information presumably improves accuracy in solving various reasoning tasks.

At inference time, CoT appends an unseen question $\hat{x}$ to the prompt $p$ and supplies the extended prompt to a LLM. The model completes the prompt to generate a relevant thought $\hat{t}$ *and* an answer $\hat{y}$. To assess the performance of LLM, CoT *only* compares the post-processed generated answer with the ground truth. Gauging the correctness of the generated thought $\hat{t}$ is not straightforward because ground truth thoughts are unavailable. Nonetheless, the generated thought can be further analyzed to infer the possible mechanisms, allowing an analogy with the human thought process, with which the model attains the answer.

**Counterfactual explanation.** Counterfactual explanations seek to explain the behavior of a model by conducting "*what if*" analysis on examples for which the expected outputs of the model is known (Mothilal et al., 2020; Stepin et al., 2021; Verma et al., 2020; Poyiadzi et al., 2020; Goyal et al., 2019; Feder et al., 2021). Specifically, let $(x, y)$ be a tuple where $x$ is the input to a model $\mathcal{M}$ that estimates an output distribution $p(\cdot \mid x)$, and $y \sim p(\cdot \mid x)$. Counterfactual explanations utilize variants $C_f(x, b, a)$ of the inputs that differ from the original input $x$ in all except one feature $f$. Here, $b$ and $a$ denote the before and after values of the feature $f$ in $x$. For instance, consider an image $x$ of a camel with a brown background labeled correctly by a classifier. A counterfactual $C_{bg}(x, brown, green)$ example is an identical image with only a different background color, green,

in this example. By virtue of comparing $p(\cdot \mid x)$ with $p(\cdot \mid C_{bg}(x, brown, green))$ for a sufficiently large sample of images, one may infer certain facts about the classifier, for example its reliance on the background color.

# F    Extended Related Work and Discussion

Broadly, this paper intersects with a growing body of work on prompting and large language model reasoning (Brown et al., 2020; Chowdhery et al., 2022; Scao et al., 2022; Zhang et al., 2022). Below, we review the most relevant work in these directions.

**Least to most prompting.** Zhou et al. (2022) help the model generate a chain of thought by first asking the model to generate the sub-questions for the given problem. Next, the model is asked to answer the sub-questions, and finally, the sub-questions, along with sub-answers, are combined to generate the final result. This work is closely related to Kojima et al. (2022), the latter distinguished by generating the rationale from a large language model directly. We posit that Zhou et al. (2022) derives its key strengths from its ability to generate useful sub-steps. This resonates with our finding that the key contribution of CoT is the extraction of meaningful sub-steps.

**Prompt selection.** Several works have recently explored the design of the prompt—a process often called "prompt engineering" (Le Scao and Rush, 2021; Liu et al., 2021c). The methods include dynamically creating prompts based on the question (Liu et al., 2021a; Rubin et al., 2021; Poesia et al., 2021), formatting the prompt as a list or questions (Mishra et al., 2021; Rubin et al., 2021), improving order of examples in the prompt (Lu et al., 2022), and providing instructions in the task (Ouyang et al., 2022). Unlike these techniques, CoT is relatively robust to minor changes in the prompt design. Thus, the findings of our work might be more generally applicable.

**Explaining model behavior using counterfactual prompts and attention.** As noted by (Jacovi and Goldberg, 2020), an explanation of a deep learning system typically serves two different purposes: i) plausibility, which aims to provide an interpretation of system outputs that is convincing for humans, and ii) faithfulness, which aims to capture the actual reasoning process of a model. Our study requires both and uses different means to achieve them. We utilize counterfactual prompts

to interpret the system outputs to aid human under standing. This is similar to using posthoc analysis tools (Ribeiro et al., 2016; Lundberg and Lee, 2017; Liu et al., 2021b), which also focus on analyzing outputs without concern for the details of the model. To get a glimpse of the model's inner workings, we leverage attention (Vaswani et al., 2017), a ubiquitous mechanism in NLP. While the broader question on the utility of attention for posthoc analysis is still open (Jain and Wallace, 2019; Pruthi et al., 2020), there is some evidence to show that attention can act as an explanation (Wiegreffe and Pinter, 2019). Finally, the utility of any explanation mechanism is closely tied to the users and application domain (Kaur et al., 2020; Burkart and Huber, 2021). As our analysis shows, attention adds intuition and insights to the empirical findings.

Counterfactual explanations seek to explain the behavior of a model by performing a *what if* analysis on examples (Mothilal et al., 2020; Stepin et al., 2021; Verma et al., 2020; Poyiadzi et al., 2020; Goyal et al., 2019). While counterfactuals can be misleading due to artifacts (e.g., see (Laugel et al., 2019; Slack et al., 2021)), they offer a tractable solution for probing large models like PaLM and GPT-3. Notably, unlike fine-tuned methods, the most important examples for generating the model output are readily available. Thus, counterfactual inputs that show a consistent and systematic change in the model performance are more likely to reflect the model's behavior.

## G  Attention Analysis

While attention mechanisms have proven invaluable for enhancing the performance of deep neural networks, they should be used with caution when interpreting how a model works. The interpretations derived from attention weights are, at best, approximate indicators of the model's decision process and should not be over-interpreted as a precise description of the underlying mechanisms.

The broader question on the utility of attention for posthoc analysis is still open (Jain and Wallace, 2019; Pruthi et al., 2020), with some evidence to show that attention can act as an explanation (Wiegreffe and Pinter, 2019). Finally, the utility of any explanation mechanism is closely tied to the users and application domain (Kaur et al., 2020; Burkart and Huber, 2021). Our analysis shows that attention provides concurring evidence that adds intuition and insights to the empirical findings of this

work. Note that while we conduct empirical experiments with PaLM, GPT-3, and CODEX, we only conduct attention-related ablations with PaLM as the GPT-3 and CODEX were only available to us via API.

**Attention for autoregressive models.** Consider a sentence: *my dog loved the toy*. Modern NLP methods divide each sentence into tokens, a decision dictated by the underlying tokenization library. PaLM uses SentencePiece (Kudo and Richardson, 2018) for tokenization. For simplicity, we assume a tokenizer that divides the sentence into tokens based on the whitespace. This yields the following list of tokens: [my, dog, loves, treats].

Let BOS be a special beginning of sequence token present in all sentences, and $p_\theta$ be a language model with the parameters $\theta$. Decoder-only language models such as PaLM estimate the likelihood of a sequence such as *my dog loved the toy* using an *autoregressive factorization* or the chain-rule:

$$p_\theta(\text{BOS}, my, dog, loves, treats) =$$
$$p_\theta(my \mid \text{BOS})$$
$$* p_\theta(dog \mid \text{BOS}, my)$$
$$* p_\theta(loves \mid \text{BOS}, my, dog)$$
$$* p_\theta(treats \mid \text{BOS}, my, dog, loves)$$

Estimating these conditional probabilities (e.g., $p_\theta(my \mid \text{BOS})$) requires a stack of transformer layers, each containing an attention module. Thus, this factorization also implies that tokens attend to the left (Figure 4), with a token $w_i$ at location $i$ attending to all tokens $w_{<i}$.

Let $w_s$ be the source token (current input to the model). The set of target tokens, or tokens that $w_s$ will attend to, thus are: $w_0, w_1, \ldots, w_{s-1}$. PaLM-62B has 64 layers, each containing the self-attention mechanism with 32 heads. Focusing on a single layer and head, let $a_{st}$ be the attention score from $w_s$ to $w_t$, where $\sum_{t=0}^{s-1} a_{st} = 1$.

**Analyzing the important components of a CoT prompt.** We leverage attention scores as an additional signal to help uncover the important components of a prompt. To this end, we calculate the attention scores *from* the *source* tokens that are part of the **Q'**, **T'**, or **A'** *to* the *target* prompt question **Q$_i$**, thought **T$_i$**, and answer **A$_i$** (Figure 3). Note that the same prompt is used for all the questions in the test set. Thus across questions, the set of target tokens remains the same.

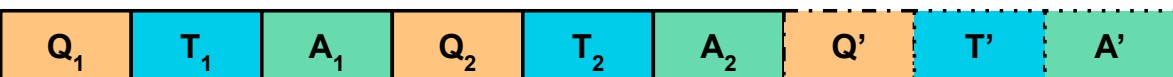

Figure 3: Structure of a typical chain of thought prompt. The prompt contains a handful of *QTA* examples, each containing three parts: 1) The question (**Q**), the thought (**T**) that spells out the reasoning process to derive the answer, and finally, 3) Answer (**A**) the final answer. In the Figure, the prompt contains two such QTA examples. During inference, a test question **Q'** is appended to the prompt, and the model is expected to complete it by generating a thought **T'** and the answer **A'**, presumably leveraging the two QTA examples in the input.

Our goal in attention analysis is to uncover important tokens and spans used by PaLM to solve a task. Since the distribution of attention scores $a_s$ is typically long-tailed, recording the attention score between every pair of source-target tokens might lead to noise and spurious patterns (Nan et al., 2021). To remedy this, we take inspiration from nucleus sampling (Holtzman et al., 2019) and set all values below the $k^{th}$ largest attention value to 0 (we use $k = 10$).

Let $Q'_j$ be the $j^{th}$ question in the test set $\mathcal{Q}$ of the questions to be evaluated. Recall that the same prompt is used for all the questions, and we calculate the attention scores from the source tokens (tokens in the inference question) to the target tokens (those in the prompt). Let $a_{st}$ be the attention from token $w_s$ to $w_t$.

We calculate the attention importance $\mathbf{I}_t$ of a token $w_t$ in the prompt as the average max attention it has received across the set $\mathcal{Q}$ of inference questions.

$$\mathbf{I}_t = \frac{\sum_{j=1}^{|\mathcal{Q}|} \max_{s=1}^{|Q'_j|+|A'_j|+|T'_j|} a_{st}}{|\mathcal{Q}|} \quad (1)$$

1. The spectrum plots show a comparison of $\mathbf{I}_t$ for all tokens in the prompt for two different prompts: vanilla CoT prompt and $C_{\mathsf{symb\_abs}}(p)$ prompt.

2. The pattern vs. text prompts group the target tokens by their type: the tokens that belong to a pattern vs. tokens belonging to the text. The attention importance values are then shown.

3. The bos by layer plots investigate the total attention importance for the BOS token.

## G.1 Per-layer Attention Analysis

The main draft provides spectrum plots averaged over heads and layers. Figure 11 shows the same question for three different datasets averaged across layers. Figure 5, Figure 6, and Figure 7 provide

the same plots, per layer. We find that the spectrum of $I_s$ values is identical between CoT(p) and $C_{\mathsf{symb\_abs}}(p)$ across layers, showing that averaging is not leading to spurious correlations.

## G.2 Specialized Attention Heads

Fine-tuned models can be expected to learn attention patterns that facilitate solving a task. *Does the same hold for few-shot models?* To our knowledge, the question of attention in a few-shot setup has not been explored. Surprisingly, we find that the model consistently uses certain heads and layers for attending over certain semantic parts of the inputs. We find such specialized head-layer pairs manually, and plot the average $I_t$ for 100 questions for them in Figure 9. The $I_t$ values show a clear tendency for the head to favor either past tense (*would, yesterday*) or future tense (*will*). Analyzing a large-language model's attention patterns in detail is an interesting future work.

**Symbiosis in attention scores** We have explored different semantic components of prompts, namely patterns (including symbols) and text. A logical next question is whether patterns or text confer differential importance. While importance can be measured via various approaches, we use attention scores as a reasonable proxy. For GSM-8K (where the distinction between patterns and text is clear), we calculate attention mass on patterns and text across several layers and average it over their attention heads. Figure 11 compares these average scores, normalized between patterns and text. Our findings show that the model pays approximately equal attention to both, indicating similar importance. These results concur with our findings that text and patterns contribute equally to the success of CoT.

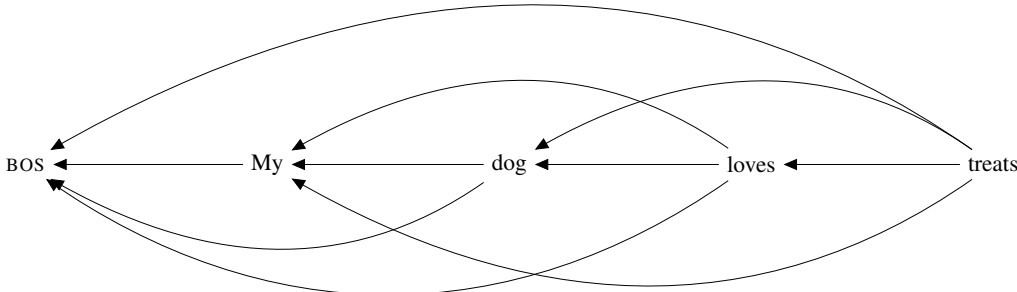

Figure 4: Auto-regressive language models: the tokens are generated as a sequence, with each token attending to the preceding tokens.

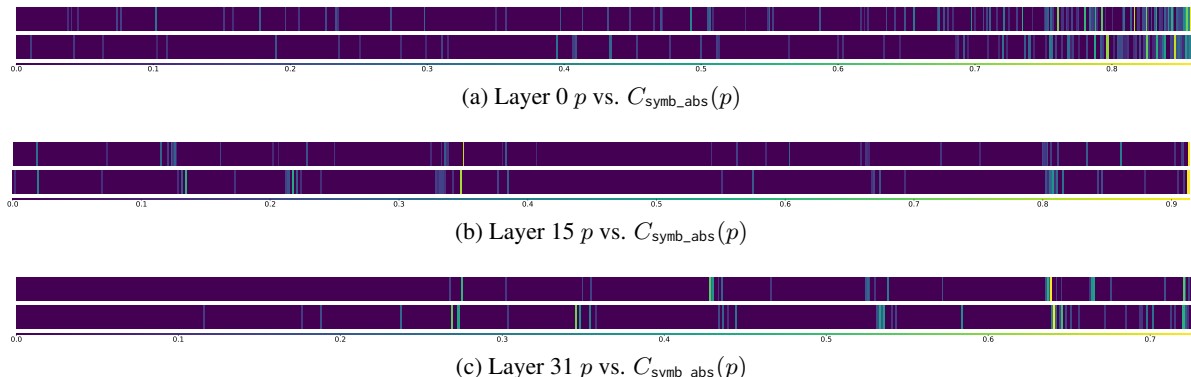

(a) Layer 0 $p$ vs. $C_{\text{symb\_abs}}(p)$

(b) Layer 15 $p$ vs. $C_{\text{symb\_abs}}(p)$

(c) Layer 31 $p$ vs. $C_{\text{symb\_abs}}(p)$

Figure 5: Average attention per token for a randomly sampled question using standard CoT prompt $p$ (above) and $C_{\text{symb\_abs}}(p)$ for GSM-8K across layers. Near identical attention pattern shows that few-shot models are relatively indifferent to the exact symbols, but are sensitive to patterns.

## H  Results on CODEX, GPT-3, PaLM-540B and Statistical Significance Test

We show results from four models: CODEX (Chen et al., 2021b), GPT-3 (Brown et al., 2020), and two variants of PaLM (Chowdhery et al., 2022) (PaLM-62B and PaLM-540B). Note that we could not get results on all variations of prompts for GPT-3 because of usage limits by OpenAI. Such cases are indicated with a hyphen (-). Similarly, due to the rate limitations, we experimented with two seeds for all variations on CODEX and had to use a single seed for some variations. The findings are shown in Table 8 (GSM-8K), Table 9 (DATE), Table 10 (SPORTS), and Table 11 (SORTING). We find that all the findings hold across models: correctness of patterns is immaterial, abstract and OOD symbols are still helpful, and the sensitivity to text is proportional to the degree of randomness. Finally, CCoT matches or outperforms CoT despite being 20% shorter.

### H.1  Significance tests for PaLM-62B

In this section, we present detailed results for experiments on PaLM-62B. Each experiment was repeated thrice using three different values of the random seed. We use McNemar's test (McNemar, 1947) to calculate the statistical significance of differences in the performance of a given Counterfactual prompt with CoT(p), and Cohen's kappa (Cohen, 1960) to measure the degree of agreement between the outputs generated by a counterfactual prompt and CoT(p).

## I  Additional Experiments

### I.1  Constructing Effective Intermediate Thoughts

Heeding our preceding findings, this section underscores few concrete venues in which the symbiosis of patterns and text contribute to the construction of effective thoughts, consequently leading to the success of CoT. To enable a systematic analysis, we first identify samples in which CoT(p) yields

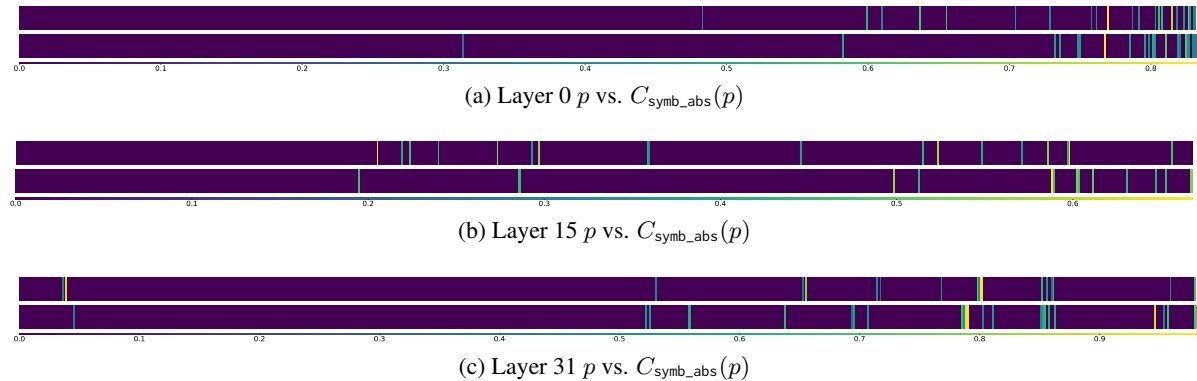

(a) Layer 0 $p$ vs. $C_{\mathtt{symb\_abs}}(p)$

(b) Layer 15 $p$ vs. $C_{\mathtt{symb\_abs}}(p)$

(c) Layer 31 $p$ vs. $C_{\mathtt{symb\_abs}}(p)$

Figure 6: Average attention per token for a randomly sampled question using standard CoT prompt $p$ (above) and $C_{\mathtt{symb\_abs}}(p)$ for DATE across layers. Near identical attention pattern shows that few-shot models are relatively indifferent to the exact symbols, but are sensitive to patterns.

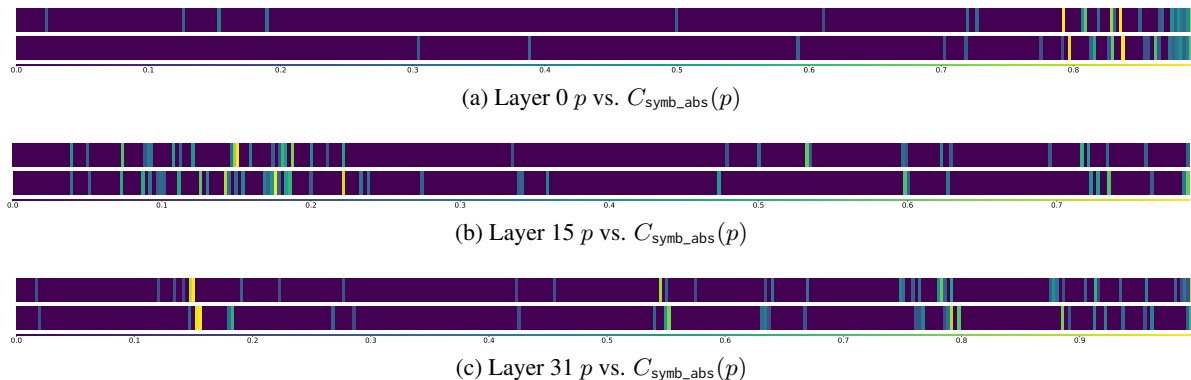

(a) Layer 0 $p$ vs. $C_{\mathtt{symb\_abs}}(p)$

(b) Layer 15 $p$ vs. $C_{\mathtt{symb\_abs}}(p)$

(c) Layer 31 $p$ vs. $C_{\mathtt{symb\_abs}}(p)$

Figure 7: Average attention per token for a randomly sampled question using standard CoT prompt $p$ (above) and $C_{\mathtt{symb\_abs}}(p)$ for SPORTS across layers. Near identical attention pattern shows that few-shot models are relatively indifferent to the exact symbols, but are sensitive to patterns.

| Prompt | CODEX | | GPT-3 | | PaLM-62B | | PaLM-540B | |
|---|---|---|---|---|---|---|---|---|
| | Avg. | SD | Avg. | SD | Avg. | SD | Avg. | SD |
| DIRECT | **20.8**% | 1.1 | 16.1% | 0 | 10.1% | 0.3 | 9.6% | 3.2 |
| CoT(p) (Table 20) | **65.6**% | 3 | 46.9% | 6.2 | 27.4% | 1.1 | 60.8% | 0.6 |
| CCoT | 62.5% | 1.4 | 52.2% | 1.4 | 29.1% | 0 | **63.2**% | 1.1 |
| $C_{\mathtt{symb\_ood}}(p)$ (Table 30) | **66.2**% | 0.5 | 55.3% | 1.7 | 25.7% | 0.5 | 60.7% | 0.2 |
| $C_{\mathtt{symb\_abs}}(p)$ (Table 25) | 56.5% | 6.4 | 49.4% | 0.1 | 28.2% | 0.2 | **59.0**% | 0.3 |
| $C_{\mathtt{pat\_wrong}}(p)$ (Table 37) | **65.5**% | 0.3 | 52.4% | 1.6 | 24.4% | 0.3 | 63.5% | 0.7 |
| $C_{\mathtt{pat\_inconsistent}}(p)$ (Table 39) | 33.3% | 0.8 | 37.8% | 2.9 | 21.5% | 0.6 | **53.9**% | 1.5 |
| $C_{\mathtt{text\_yodathoughts}}(p)$ (Table 52) | **60.8**% | 0 | - | - | 23.2% | 1 | 57.4% | 1.4 |
| $C_{\mathtt{text\_intra\_shuf}}(p)$ (Table 61) | 33.0% | 1.8 | - | - | 17.0% | 1.3 | **45.6**% | 3.8 |
| $C_{\mathtt{text\_inter\_shuf}}(p)$ (Table 64) | 29.7% | 6.3 | - | - | 10.8% | 1.3 | **37.2**% | 3.1 |
| $C_{\mathtt{text\_diff\_entities}}(p)$ (Table 50) | **59.0**% | 0 | 49.8% | 0 | 16.6% | 1 | 51.1% | 3.3 |

Table 8: All results for GSM-8K across four models: CODEX, GPT-3, PaLM-62B, and PaLM-540B.

correct answer, whereas **both** $C_{\mathtt{pat\_inconsistent}}(p)$ and $C_{\mathtt{pat\_only}}(p)$ are wrong. Analyzing these samples assist us in identifying probable systematic differences across these methods.

**CoT is more effective in solving questions with more patterns.** In general, questions with more patterns require more intermediate steps to arrive at correct answers. Thus, CoT is expected to help more for such cases. We test this hypothesis by glancing into the GSM-8K dataset. The num-

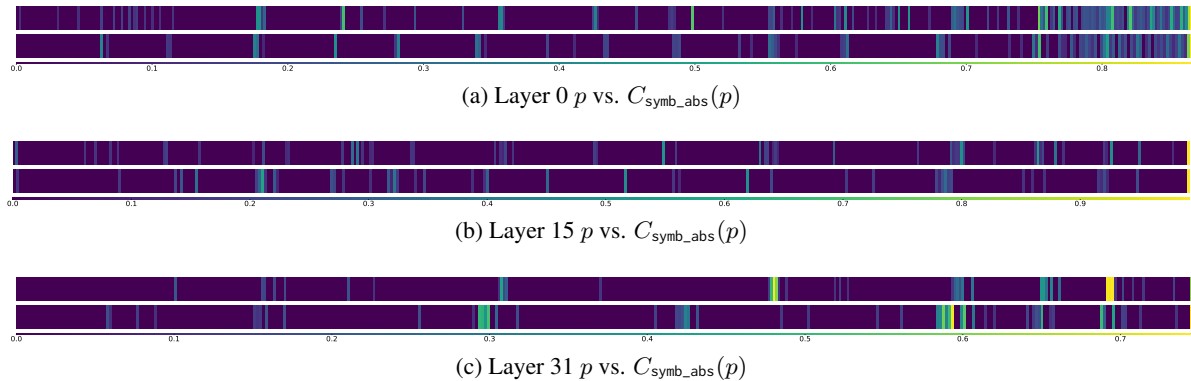

(a) Layer 0 $p$ vs. $C_{\texttt{symb\_abs}}(p)$

(b) Layer 15 $p$ vs. $C_{\texttt{symb\_abs}}(p)$

(c) Layer 31 $p$ vs. $C_{\texttt{symb\_abs}}(p)$

Figure 8: Average attention per token for a randomly sampled question using standard CoT prompt $p$ (above) and $C_{\texttt{symb\_abs}}(p)$ for SORTING across layers. Near identical attention pattern shows that few-shot models are relatively indifferent to the exact symbols, but are sensitive to patterns.

| | CODEX | | GPT-3 | | PaLM-62B | | PaLM-540B | |
|---|---|---|---|---|---|---|---|---|
| **Prompt** | **Avg.** | **SD** | **Avg.** | **SD** | **Avg.** | **SD** | **Avg.** | **SD** |
| DIRECT | **51.3%** | 1 | 44.2% | 0.5 | 31.6% | 1.6 | 49.0% | - |
| CoT(p) (Table 21) | **69.2%** | 2 | 56.7% | 3.7 | 45.2% | 0.5 | 65.3% | - |
| CCoT | **69.9%** | 5.7 | 61.2% | 1.4 | 51.3% | - | 69.6% | - |
| $C_{\texttt{symb\_ood}}(p)$ (Table 31) | **67.8%** | 1.8 | 58.2% | 0 | 44.5% | 1.4 | 59.6% | - |
| $C_{\texttt{symb\_abs}}(p)$ (Table 24) | 54.1% | 3.9 | 35% | 2 | 36.6% | 1 | **56.2%** | 0.8 |
| $C_{\texttt{pat\_wrong}}(p)$ (Table 36) | **69.0%** | 1.9 | 58.4% | 1.6 | 42.9% | 3 | 67.2% | 0.6 |
| $C_{\texttt{pat\_inconsistent}}(p)$ (Table 44) | **62.0%** | 1.4 | 50.9% | 3.1 | 36.1% | 1.2 | 59.6% | 2 |
| $C_{\texttt{text\_yodathoughts}}(p)$ (Table 54) | 55.0% | 2 | - | - | 30.8% | 1.7 | **62.2%** | 1.6 |
| $C_{\texttt{text\_intra\_shuf}}(p)$ (Table 63) | 44.3% | 3.9 | - | - | 25.5% | 0.7 | **54.7%** | 0.4 |
| $C_{\texttt{text\_inter\_shuf}}(p)$ (Table 66) | 39.0% | 2.4 | - | - | 24.2% | 0.8 | **44.4%** | 2 |

Table 9: Results for DATE across four models: CODEX, GPT-3, PaLM-62B, and PaLM-540B.

| | CODEX | | GPT-3 | | PaLM-62B | | PaLM-540B | |
|---|---|---|---|---|---|---|---|---|
| **Prompt** | **Avg.** | **SD** | **Avg.** | **SD** | **Avg.** | **SD** | **Avg.** | **SD** |
| DIRECT | 70.2% | 3.6 | 68.9% | 1 | 71.1% | 2 | **80.5%** | - |
| CoT(p) (Table 22) | **98.3%** | 0.1 | 82.1% | 2.2 | 93.6% | 0.3 | 95.4% | - |
| CCoT | **98.5%** | 0.1 | 85.3% | 0.5 | 94.6% | 0 | 97.4% | - |
| $C_{\texttt{symb\_ood}}(p)$ (Table 32) | **79.7%** | 1.8 | 69.9% | 0 | 79.3% | 0.3 | 79.2% | - |
| $C_{\texttt{symb\_abs\_per}}(p)$ (Table 27) | **86.7%** | 1.6 | 72.8% | 1.2 | 85.9% | 0.4 | - | - |
| $C_{\texttt{pat\_wrong}}(p)$ (Table 46) | 53.5% | 0.1 | **66.2%** | 2.9 | 53.7% | 0.6 | 53.7% | 0.6 |
| $C_{\texttt{pat\_inconsistent}}(p)$ (Table 45) | 78.3% | 1.8 | 71.2% | 0.6 | 79.2% | 6.6 | **85.8%** | 4.9 |
| $C_{\texttt{text\_diff\_entities}}(p)$ (Table 51) | **81.6%** | 13.9 | 66.9% | 0.1 | 66.9% | 2 | 54.5% | 0 |
| $C_{\texttt{text\_yodathoughts}}(p)$ (Table 53) | **93.7%** | 1.3 | - | - | 65.8% | 5.7 | 82.2% | 0.9 |
| $C_{\texttt{text\_intra\_shuf}}(p)$ (Table 62) | **70.1%** | 0 | - | - | 61.2% | 4.6 | 69.8% | 0 |
| $C_{\texttt{text\_inter\_shuf}}(p)$ (Table 65) | 66.8% | 2.1 | - | - | 61.7% | 2 | **70.0%** | 0 |

Table 10: All results for SPORTS across four models: CODEX, GPT-3, PaLM-62B, and PaLM-540B.

ber of GSM-8K questions that the model exclusively solve using CoT(p) is 140. Note that these are the questions that neither $C_{\texttt{pat\_inconsistent}}(p)$ nor $C_{\texttt{pat\_only}}(p)$ can solve. We observe that this is roughly twice and four-times the num-

ber of questions that are exclusively solved by $C_{\texttt{pat\_inconsistent}}(p)$ (71) and $C_{\texttt{pat\_only}}(p)$ (33), respectively. Further, the average number of numerical entities in these questions is 3.98 as compared to the overall average of 3.62, a statistically sig-

Q : 2 0 1 5 is coming in 3 6 hours . What is the date one week from today in MM / DD / YYYY ? A : If 2 0 1 5 is coming in 3 6 hours , then it is coming in 2 days . 2 days before 0 1 / 0 1 / 2 0 1 5 is 1 2 / 3 0 / 2 0 2 1 , so today is 1 2 / 3 0 / 2 0 2 1 . So one week from today will be 0 1 / 0 5 / 2 0 1 5 . So the answer is 0 1 / 0 5 / 2 0 1 5 . Q : The first day of 2 0 1 9 is a Tuesday , and today is the first Monday of 2 0 1 9 . What is the date today in MM / DD / YYYY ? A : If the first day of 2 0 1 9 was Tuesday , then 0 1 / 0 1 / 2 0 1 9 was a Tuesday . Today is the first monday , would be six days later . So today is 0 1 / 0 7 / 2 0 1 9 . So the answer is 0 1 / 0 7 / 2 0 1 9 . Q : The concert was scheduled to be on 0 6 / 0 1 / 1 9 4 3 , but was delayed by one day to today . What is the date 1 0 days ago in MM / DD / YYYY ? A : One day after 0 6 / 0 1 / 1 9 4 3 is 0 6 / 0 2 / 1 9 4 3 , so today is 0 6 / 0 2 / 1 9 4 3 . 1 0 days before today is 0 5 / 2 3 / 1 9 4 3 . So the answer is 0 5 / 2 3 / 1 9 4 3 . Q : It is 4 / 1 9 / 1 9 6 9 today . What is the date 2 4 hours later in MM / DD / YYYY ? A : Today is 0 4 / 1 9 / 1 9 6 9 . 2 4 hours later is one day after today , which would be 0 4 / 2 0 / 1 9 6 9 . So the answer is 0 4 / 2 0 / 1 9 6 9 . Q : Jane thought today is 3 / 1 1 / 2 0 0 2 , but today is in fact Mar 1 2 , which is 1 day later . What is the date 2 4 hours later in MM / DD / YYYY ? A : Today is 0 3 / 1 2 / 2 0 0 2 . So the date 2 4 hours later will be 0 3 / 1 3 / 2 0 0 2 . So the answer is 0 3 / 1 3 / 2 0 0 2 . Q : Jane was born on the last day of Feb ur ary in 2 0 0 1 . Today is her 1 6 - year - old birthday . What is the date yesterday in MM / DD / YYYY ? A : The last day of February is the 2 8 th , so Jane was born on 0 2 / 2 8 / 2 0 0 1 . Today is her 1 6 - year old birthday , so today is 0 2 / 2 8 / 2 0 1 7 . So yesterday was 0 2 / 2 7 / 2 0 1 7 . So the answer is 0 2 / 2 7 / 2 0 1 7 .

(a) Layer 45, Head 16 specializes in capturing tokens relevant to "past tense".

Q : 2 0 1 5 is coming in 3 6 hours . What is the date one week from today in MM / DD / YYYY ? A : If 2 0 1 5 is coming in 3 6 hours , then it is coming in 2 days . 2 days before 0 1 / 0 1 / 2 0 1 5 is 1 2 / 3 0 / 2 0 2 1 , so today is 1 2 / 3 0 / 2 0 2 1 . So one week from today will be 0 1 / 0 5 / 2 0 1 5 . So the answer is 0 1 / 0 5 / 2 0 1 5 . Q : The first day of 2 0 1 9 is a Tuesday , and today is the first Monday of 2 0 1 9 . What is the date today in MM / DD / YYYY ? A : If the first day of 2 0 1 9 was Tuesday , then 0 1 / 0 1 / 2 0 1 9 was a Tuesday . Today is the first monday , would be six days later . So today is 0 1 / 0 7 / 2 0 1 9 . So the answer is 0 1 / 0 7 / 2 0 1 9 . Q : The concert was scheduled to be on 0 6 / 0 1 / 1 9 4 3 , but was delayed by one day to today . What is the date 1 0 days ago in MM / DD / YYYY ? A : One day after 0 6 / 0 1 / 1 9 4 3 is 0 6 / 0 2 / 1 9 4 3 , so today is 0 6 / 0 2 / 1 9 4 3 . 1 0 days before today is 0 5 / 2 3 / 1 9 4 3 . So the answer is 0 5 / 2 3 / 1 9 4 3 . Q : It is 4 / 1 9 / 1 9 6 9 today . What is the date 2 4 hours later in MM / DD / YYYY ? A : Today is 0 4 / 1 9 / 1 9 6 9 . 2 4 hours later is one day after today , which would be 0 4 / 2 0 / 1 9 6 9 . So the answer is 0 4 / 2 0 / 1 9 6 9 . Q : Jane thought today is 3 / 1 1 / 2 0 0 2 , but today is in fact Mar 1 2 , which is 1 day later . What is the date 2 4 hours later in MM / DD / YYYY ? A : Today is 0 3 / 1 2 / 2 0 0 2 . So the date 2 4 hours later will be 0 3 / 1 3 / 2 0 0 2 . So the answer is 0 3 / 1 3 / 2 0 0 2 . Q : Jane was born on the last day of Feb ur ary in 2 0 0 1 . Today is her 1 6 - year - old birthday . What is the date yesterday in MM / DD / YYYY ? A : The last day of February is the 2 8 th , so Jane was born on 0 2 / 2 8 / 2 0 0 1 . Today is her 1 6 - year old birthday , so today is 0 2 / 2 8 / 2 0 1 7 . So yesterday was 0 2 / 2 7 / 2 0 1 7 . So the answer is 0 2 / 2 7 / 2 0 1 7 .

(b) Layer 45, Head 18 specializes in capturing tokens relevant to "future tense".

Figure 9: Specialized attention heads.

| Prompt | CODEX | | GPT-3 | | PaLM-62B | | PaLM-540B | |
|---|---|---|---|---|---|---|---|---|
| | Avg. | SD | Avg. | SD | Avg. | SD | Avg. | SD |
| DIRECT | **100**% | - | 50% | - | 46.87% | 2.288 | 99.8% | - |
| CoT(p) (Table 23) | **100**% | - | 50% | - | 61.87% | 3.151 | 99.8% | - |
| $C_{symb\_ood}(p)$ (Table 33) | **100**% | - | 50% | - | 57.2% | 3.441 | 99% | - |
| $C_{symb\_abs}(p)$ (Table 26) | **100**% | - | 50% | - | 81.47% | 10.804 | 91.4% | - |
| $C_{pat\_wrong}(p)$ (Table 47) | **99.9**% | 0.1 | 50% | - | 61.47% | 4.014 | 99.0% | - |
| $C_{pat\_inconsistent}(p)$ (Table 43) | 45.2% | 4 | 24.6% | - | 79.27% | 9.116 | **84.2**% | - |

Table 11: All results for SORTING across four models: CODEX, GPT-3, PaLM-62B, and PaLM-540B.

nificant difference (difference of means t-test $p = 0.04$). We attribute this significant gain to the ability of CoT(p) in forming longer and more detailed intermediate steps. These additional steps presumably steer the model to elicit meaningful structures/patterns from thoughts, arriving at a factual answer (➥ $\mathcal{Q}$1-◆ $\mathcal{T}_{\llcorner}$⌜CoT⌝ in Table 16). As we mentioned, text glues patterns and language structures together. Similarly, the presence of patterns not only leads to meaningful thoughts, but also assists the model not to neglect the symbols. Without patterns, the model has intrinsic tendency to prematurely arrive at a conclusion (➥ $\mathcal{Q}$2,➥ $\mathcal{Q}$6-◆ $\mathcal{T}_{\llcorner}$⌜$C_{pat\_inconsistent}(p)$⌝ in Table 16), which is more probable to be inaccurate.

**The importance of intermediate steps in reviving CoT.** One of the crucial intentions of thoughts is to resemble a practical intermediate solution process. In addition, thoughts reiterate patterns and text that presumably act as a beacon for the model to direct its attention (of various intensity) to particular tokens. To better understand the intricacy of repeated patterns and text in thought, we analyze the generated thoughts in the DATE dataset. We observe several scenarios in which the $C_{pat\_only}(p)$ prevents the model from generating requisite information (e.g., "*10 days ago*" ➥ $\mathcal{Q}$4 in Table 16) and complex patterns (e.g., "*day before yesterday*"

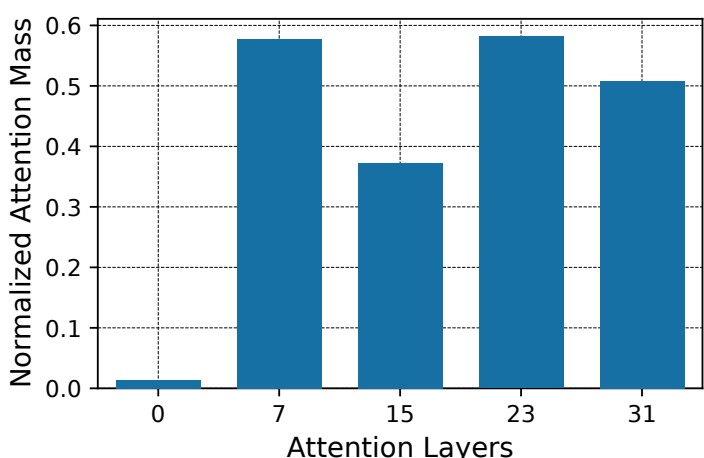

Figure 10: Attention on the beginning of sequence token across layers. Averaged across attention heads, higher attention mass goes to the bos token for the higher layers. Note that PaLM is auto-regressive, thus, higher attention mass on bos may be interpreted as the model paying equal attention to the rest of the sequence, indicating an evolving global representation.

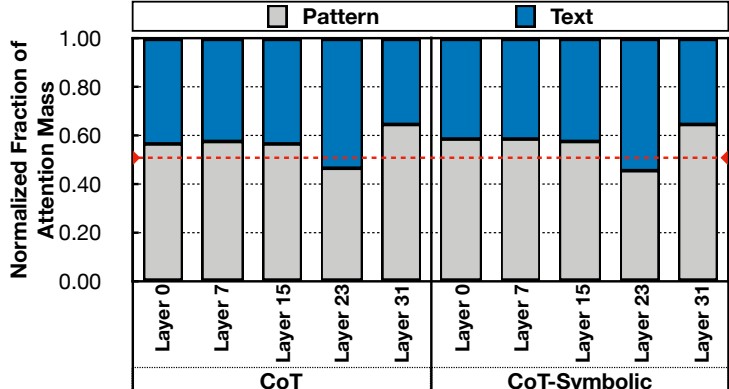

Figure 11: Visualizing the normalized fraction of attention mass between text and patterns across multiple layers of the PaLM-62B model layers with the vanilla CoT (left-side) and symbolic CoT (right-side). In general, patterns receive slightly higher attention across most of the layers, with the topmost layer paying the largest attention. Strikingly, the attention patterns closely match for vanilla and symbolic CoT, implying that few-shot models leverage patterns to a larger extent.

| | | | Solve Rate | | | | |
|---|---|---|---|---|---|---|---|
| **Prompt** | **S0** | **S1** | **S2** | **Avg.** | **SD** | **Mcnemar's $p$-value** | **Cohen's $\kappa$** |
| DIRECT | 10.08% | 10.54% | 9.70% | 10.11% | 0.341 | <0.00001 | 0.0956 |
| CoT(p) (Table 20) | 28.81% | 26.16% | 27.14% | 27.37% | 1.095 | — | — |
| $C_{\texttt{symb\_abs}}(p)$ (Table 25) | 25.70% | 25.09% | 26.31% | 25.70% | 0.495 | 0.320693 | 0.4846 |
| $C_{\texttt{symb\_ood}}(p)$ (Table 30) | 28.28% | 28.43% | 27.90% | 28.20% | 0.223 | 0.393705 | 0.6655 |
| $C_{\texttt{symb\_ood\_verbalized}}(p)$ (Table 69) | 24.03% | 22.74% | 24.11% | 23.63% | 0.626 | 0.005653 | 0.5193 |
| $C_{\texttt{symb\_ood\_neg}}(p)$ (Table 34) | 28.43% | 26.16% | 26.16% | 26.91% | 1.072 | 0.702915 | 0.6762 |
| $C_{\texttt{symb\_ood\_large}}(p)$ (Table 35) | 28.58% | 26.38% | 26.99% | 27.32% | 0.927 | 0.900833 | 0.6513 |
| $C_{\texttt{pat\_only}}(p)$ (Table 30) | 10.46% | 9.48% | 10.08% | 10.01% | 0.406 | <0.00001 | 0.1935 |
| $C_{\texttt{pat\_wrong}}(p)$ (Table 37) | 24.26% | 24.11% | 24.79% | 24.39% | 0.293 | 0.040501 | 0.5849 |
| $C_{\texttt{pat\_inconsistent}}(p)$ (Table 39) | 22.37% | 21.08% | 20.92% | 21.46% | 0.646 | 0.000018 | 0.4404 |
| $C_{\texttt{text\_diff\_entities}}(p)$ (Table 50) | 17.13% | 17.44% | 15.24% | 16.60% | 0.973 | <0.00001 | 0.3725 |
| $C_{\texttt{text\_rand}}(p)$ (Table 58) | 2.88% | 2.81% | 3.26% | 2.98% | 0.199 | <0.00001 | 0.0153 |
| $C_{\texttt{text\_yoda}}(p)$ (Table 55) | 24.03% | 21.08% | 21.46% | 22.19% | 1.314 | 0.000147 | 0.4056 |
| $C_{\texttt{text\_yodathoughts}}(p)$ (Table 52) | 24.18% | 21.68% | 23.81% | 23.22% | 1.101 | 0.005890 | 0.3859 |
| $C_{\texttt{text\_yodaquestions}}(p)$ (Table 67) | 28.35% | 26.08% | 26.84% | 27.09% | 0.946 | 0.847928 | 0.6394 |
| $C_{\texttt{text\_intra\_shuf}}(p)$ (Table 61) | 18.20% | 15.24% | 17.59% | 17.01% | 1.275 | <0.00001 | 0.3343 |
| $C_{\texttt{text\_inter\_shuf}}(p)$ (Table 64) | 12.13% | 9.10% | 11.30% | 10.84% | 1.279 | <0.00001 | 0.2308 |

Table 12: All results for GSM-8K: p-values corresponding to the results of McNemar's test comparing marginal frequencies of correct responses between various counterfactual prompts and CoT. Cohen's $\kappa$ measures agreement between the decisions of CoT and the counterfactual prompt. An agreement of over 0.4 is moderate, and over 0.6 is substantial.

| | | | Solve Rate | | | | |
|---|---|---|---|---|---|---|---|
| **Prompt** | **S0** | **S1** | **S2** | **Avg.** | **SD** | **Mcnemar's $p$-value** | **Cohen's $\kappa$** |
| DIRECT | 29.51% | 32.09% | 33.24% | 31.61% | 1.558 | 0.000002 | 0.4888 |
| CoT(p) (Table 21) | 44.70% | 44.99% | 45.85% | 45.18% | 0.487 | — | — |
| $C_{\texttt{symb\_abs}}(p)$ (Table 24) | 37.54% | 36.96% | 35.24% | 36.58% | 0.974 | 0.001335 | 0.5844 |
| $C_{\texttt{symb\_ood}}(p)$ (Table 31) | 42.69% | 46.13% | 44.70% | 44.51% | 1.410 | 0.520219 | 0.7895 |
| $C_{\texttt{pat\_only}}(p)$ (Table 31) | 33.52% | 32.38% | 33.52% | 33.14% | 0.540 | 0.000001 | 0.5931 |
| $C_{\texttt{pat\_wrong}}(p)$ (Table 36) | 38.68% | 45.56% | 44.41% | 42.88% | 3.008 | 0.433173 | 0.8060 |
| $C_{\texttt{pat\_inconsistent}}(p)$ (Table 44) | 37.54% | 36.10% | 34.67% | 36.10% | 1.170 | 0.000263 | 0.6212 |
| $C_{\texttt{text\_rand}}(p)$ (Table 60) | 21.78% | 28.37% | 18.05% | 22.73% | 4.265 | <0.00001 | 0.4094 |
| $C_{\texttt{text\_yodathoughts}}(p)$ (Table 54) | 28.94% | 32.95% | 30.37% | 30.75% | 1.660 | 0.000004 | 0.4426 |
| $C_{\texttt{text\_yoda}}(p)$ (Table 57) | 34.10% | 32.09% | 33.24% | 33.14% | 0.822 | 0.000010 | 0.5023 |
| $C_{\texttt{text\_yodaquestions}}(p)$ (Table 57) | 44.13% | 48.14% | 42.12% | 44.79% | 2.502 | 0.358008 | 0.7609 |
| $C_{\texttt{text\_intra\_shuf}}(p)$ (Table 63) | 26.36% | 25.50% | 24.64% | 25.50% | 0.702 | <0.00001 | 0.4428 |
| $C_{\texttt{text\_inter\_shuf}}(p)$ (Table 66) | 25.21% | 23.78% | 23.50% | 24.16% | 0.752 | <0.00001 | 0.4332 |

Table 13: All results for DATE: p-values corresponding to the results of McNemar's test comparing marginal frequencies of correct responses between various counterfactual prompts and CoT. Cohen's $\kappa$ measures agreement between the decisions of CoT and the counterfactual prompt. An agreement of over 0.4 is moderate, and over 0.6 is substantial.

➥ $\mathcal{Q}5$ in Table 16) out of questions.

$C_{\texttt{pat\_inconsistent}}(p)$ exhibit a similar trend, without interspersed patterns in prompts, the model often arrives at a wrong outcome via semantically correct steps. For example, ➥ $\mathcal{Q}1$-◆ $\mathcal{T}_{\llcorner}^{\ulcorner}C_{\texttt{pat\_inconsistent}}(p)_{\lrcorner}^{\urcorner}$ in Table 17 shows correct intermediate steps up until the very last calculation. The last calculation—*"44 times the weight of the Papillon, so 216 pounds"*—however, is incorrect. On the contrary, the model arrive at a factual and mathematically correct result when explicit pat-terns are employed (*"so the Mastiff weighed 44 × 5 = 220 pounds"*). Interestingly, we observe an identical phenomenon occurs in the DATE dataset. Recall that for DATE (Table 1), the patterns in the thoughts are (a) ⟨**calculation**⟩ ↦ a description of computation of the current state and (b) ⟨**output**⟩ ↦ a statement that repeats the requisite intermediate information from (a) to attain a correct answer. Breaking such explicit pattern disorients the model from generating intermediate information, as in ➥ $\mathcal{Q}5$-◆ $\mathcal{T}_{\llcorner}^{\ulcorner}C_{\texttt{pat\_inconsistent}}(p)_{\lrcorner}^{\urcorner}$ in Table 17.

| Prompt | Solve Rate | | | | | Mcnemar's $p$-value | Cohen's $\kappa$ |
|---|---|---|---|---|---|---|---|
| | S0 | S1 | S2 | Avg. | SD | | |
| DIRECT | 72.18% | 68.25% | 72.80% | 71.08% | 2.015 | <0.00001 | 0.0721 |
| CoT(p) (Table 22) | 93.59% | 94.00% | 93.17% | 93.59% | 0.338 | — | — |
| $C_{\text{symb\_abs\_per}}(p)$ (Table 27) | 86.25% | 85.32% | 86.14% | 85.90% | 0.417 | <0.00001 | 0.4420 |
| $C_{\text{symb\_abs\_act}}(p)$ (Table 28) | 90.69% | 92.45% | 93.17% | 92.11% | 1.042 | 0.349727 | 0.5878 |
| $C_{\text{symb\_ood}}(p)$ (Table 32) | 79.01% | 79.73% | 79.11% | 79.28% | 0.320 | <0.00001 | 0.2630 |
| $C_{\text{pat\_only}}(p)$ (Table 41) | 73.11% | 75.39% | 76.01% | 74.84% | 1.245 | <0.00001 | 0.2008 |
| $C_{\text{pat\_wrong}}(p)$ (Table 46) | 43.54% | 49.12% | 52.84% | 48.50% | 3.825 | <0.00001 | -0.0099 |
| $C_{\text{pat\_inconsistent}}(p)$ (Table 45) | 70.01% | 82.01% | 85.52% | 79.18% | 6.641 | <0.00001 | 0.1759 |
| $C_{\text{text\_diff\_entities}}(p)$ (Table 51) | 69.49% | 64.63% | 66.49% | 66.87% | 2.002 | <0.00001 | 0.0689 |
| $C_{\text{text\_rand}}(p)$ (Table 59) | 52.84% | 63.39% | 63.08% | 59.77% | 4.901 | <0.00001 | 0.0931 |
| $C_{\text{text\_yodathoughts}}(p)$ (Table 53) | 63.19% | 73.73% | 60.39% | 65.77% | 5.745 | <0.00001 | 0.1021 |
| $C_{\text{text\_yoda}}(p)$ (Table 56) | 67.43% | 69.29% | 55.12% | 63.94% | 6.286 | <0.00001 | 0.1114 |
| $C_{\text{text\_yodaquestions}}(p)$ (Table 56) | 91.73% | 89.56% | 90.49% | 90.59% | 0.890 | 0.000194 | 0.6772 |
| $C_{\text{text\_intra\_shuf}}(p)$ (Table 62) | 66.70% | 61.53% | 55.43% | 61.22% | 4.607 | <0.00001 | 0.1333 |
| $C_{\text{text\_inter\_shuf}}(p)$ (Table 65) | 58.84% | 63.19% | 63.19% | 61.74% | 2.047 | <0.00001 | 0.0595 |

Table 14: All results for SPORTS: p-values corresponding to the results of McNemar's test comparing marginal frequencies of correct responses between various counterfactual prompts and CoT. Cohen's $\kappa$ measures agreement between the decisions of CoT and the counterfactual prompt. An agreement of over 0.4 is moderate, and over 0.6 is substantial.

| Prompt | Solve Rate | | | | | Mcnemar's $p$-value | Cohen's $\kappa$ |
|---|---|---|---|---|---|---|---|
| | S0 | S1 | S2 | Avg. | SD | | |
| DIRECT | 46.0% | 50.0% | 44.6% | 46.87% | 2.288 | <0.00001 | 0.6981 |
| CoT(p) (Table 23) | 60.6% | 66.2% | 58.8% | 61.87% | 3.151 | — | — |
| $C_{\text{symb\_abs}}(p)$ (Table 26) | 52.8% | 57.6% | 61.2% | 57.20% | 3.441 | 0.140201 | 0.2333 |
| $C_{\text{symb\_ood}}(p)$ (Table 33) | 91.2% | 66.4% | 86.8% | 81.47% | 10.804 | 0.333333 | 0.4274 |
| $C_{\text{pat\_wrong}}(p)$ (Table 47) | 59.8% | 67.0% | 57.6% | 61.47% | 4.014 | 0.305524 | 0.9390 |
| $C_{\text{pat\_inconsistent}}(p)$ (Table 43) | 66.4% | 85.0% | 86.4% | 79.27% | 9.116 | 0.019411 | 0.1836 |

Table 15: Results for SORTING: p-values calculated using McNemar's test for various counterfactual prompts used in SORTING experiments. The p-value corresponds to the null hypothesis: *The marginal probability of a sample being correct with the counterfactual prompt and CoT is same*. Larger p-values indicates that likelihood that the null hypothesis is correct is large. Cohen's $\kappa$ measures the degree of agreement between the decisions of CoT and the counterfactual prompt. An agreement of over 0.4 is moderate, and over 0.6 is substantial.

## I.2 Commonsense Extraction

We conjecture that CoT unlocks a golden opportunity to bring forth supplementary commonsense knowledge from the question and generate correct intermediate steps. Consequently, the generated commonsense knowledge assists the model to attain a factual answer. For example, we observe that in ➥ $\mathcal{Q}$3 of Table 16 the model with CoT infuses commonsense knowledge about animals (e.g., "*5 dogs have 4 legs each . . . 2 cats have 4 legs each . . . 10 birds have 2 legs each*"). In addition, reviewing the generated thoughts in DATE underscores the ability of the model to recast commonsense knowledge about dates into a coherent format in thoughts (➥ $\mathcal{Q}$6-◆ $\mathcal{T}\lceil\text{CoT}\rceil$ in Table 16). In this example, the model articulates the exact date for "*Christ-*

*mas Eve*" in the generated thought. Evidently, the model conditions on "*Christmas Eve*" and "*Today is*" to form the exact date "*12/24/1937*". This is undeniably an arduous task for the DIRECT setup, as it warrants the creation of these two fragments of information in one step.

**Extracting rare commonsense knowledge.** To better understand the ability of the model to extract rare commonsense knowledge, we resort to the number of Google search results, which we refer to as "Popularity Metric", as a proxy to gauge the rarity of an entity. Employing this metric is germane to PaLM's training dataset, which is a web-based corpus (Chowdhery et al., 2022). We use this metric in the SPORTS dataset because the model is required to reason about factual commonsense knowledge to arrive at the correct conclu-

Table 16: Language generates better patterns: error analysis of cases where lack of language leads to misleading and incorrect thoughts for various datasets.

---

➡ $\mathcal{Q}1 \mapsto$ Skyler has 100 hats on his hand with the colors red, blue, and white. Half of the hats are red, 3/5 of the remaining hats are blue, and the rest are white. How many white hats does Skyler have?

◈ $\mathcal{T}^{\ulcorner}\text{CoT(p)}^{\urcorner} \mapsto$ Skyler has 100 hats. Half of the hats are red. So 100 / 2 = 50 hats are red. 3/5 of the remaining hats are blue. So 50 - 3/5 = 30 hats are blue. The rest are white. So 100 - 50 - 30 = 20 hats are white. The answer is 20.✓

◈ $\mathcal{T}^{\ulcorner}C_{\text{pat\_only}}(p)^{\urcorner} \mapsto$ 100 / 3 = 33. The answer is 33.✗

➡ $\mathcal{Q}2 \mapsto$ A crocodile grows 8 inches long in 4 years. At this rate, how many inches will it grow in 13 years?

◈ $\mathcal{T}^{\ulcorner}\text{CoT(p)}^{\urcorner} \mapsto$ The crocodile grows 8 inches long in 4 years. So it grows 8 / 4 = 2 inches long each year. 13 years is 13 times 2, so it will grow 13 * 2 = 26 inches long. The answer is 26.✓

◈ $\mathcal{T}^{\ulcorner}C_{\text{pat\_only}}(p)^{\urcorner} \mapsto$ 8 * 13 = 104. The answer is 104.✗

➡ $\mathcal{Q}3 \mapsto$ A pet store currently has 5 dogs, 2 cats, and 10 birds. How many legs in total do the pets in the store have?

◈ $\mathcal{T}^{\ulcorner}\text{CoT(p)}^{\urcorner} \mapsto$ There are 5 dogs, 2 cats, and 10 birds. 5 dogs have 4 legs each, so 5 * 4 = 20 legs. 2 cats have 4 legs each, so 2 * 4 = 8 legs. 10 birds have 2 legs each, so 10 * 2 = 20 legs. So the total number of legs is 20 + 8 + 20 = 48. The answer is 48.✓

◈ $\mathcal{T}^{\ulcorner}C_{\text{pat\_only}}(p)^{\urcorner} \mapsto$ 5 + 2 + 10 = 17. The answer is 17.✗

➡ $\mathcal{Q}4 \mapsto$ The current local time is 3:02 pm of 5/4/2004. What is the date 10 days ago in MM/DD/YYYY?

◈ $\mathcal{T}^{\ulcorner}\text{CoT(p)}^{\urcorner} \mapsto$ The current local time is 3:02 pm of 05/04/2004. 10 days ago would be 04/24/2004. So the answer is 04/24/2004.✓

◈ $\mathcal{T}^{\ulcorner}C_{\text{pat\_only}}(p)^{\urcorner} \mapsto$ 5/4/2004 + 10 days = 5/14/2004. So the answer is 05/14/2004.✗

➡ $\mathcal{Q}5 \mapsto$ The day before yesterday was 11/23/1933. What is the date yesterday in MM/DD/YYYY?

◈ $\mathcal{T}^{\ulcorner}\text{CoT(p)}^{\urcorner} \mapsto$ The day before yesterday was 11/23/1933. Yesterday was 11/24/1933. So the answer is 11/24/1933.✓

◈ $\mathcal{T}^{\ulcorner}C_{\text{pat\_only}}(p)^{\urcorner} \mapsto$ 11/23/1933 - 1 day = 11/22/1933. Today = 11/22/1933 - 1 day = 11/21/1933. So the answer is 11/21/1933.✗

➡ $\mathcal{Q}6 \mapsto$ Today is Christmas Eve of 1937. What is the date one week ago from today in MM/DD/YYYY?

◈ $\mathcal{T}^{\ulcorner}\text{CoT(p)}^{\urcorner} \mapsto$ Today is 12/24/1937. One week ago from today is 12/17/1937. So the answer is 12/17/1937.✓

◈ $\mathcal{T}^{\ulcorner}C_{\text{pat\_only}}(p)^{\urcorner} \mapsto$ 12/20/1937✗

➡ $\mathcal{Q}7 \mapsto$ The concert was scheduled to be on 06/01/1943, but was delayed by one day to today. What is the date one year ago from today in MM/DD/YYYY?

◈ $\mathcal{T}^{\ulcorner}\text{CoT(p)}^{\urcorner} \mapsto$ One day after 06/01/1943 is 06/02/1943, so today is 06/02/1943. One year ago from today is 06/02/1942. So the answer is 06/02/1942.✓

◈ $\mathcal{T}^{\ulcorner}C_{\text{pat\_only}}(p)^{\urcorner} \mapsto$ 1 year ago from today is 06/01/1942. So the answer is 06/01/1942.✗

---

sions. Table 18 shows a contingency table summarizing the average popularity metric of "⟨Activity⟩" across correct and incorrect conclusions by CoT and $C_{\text{pat\_only}}(p)$. We observe that the average popularity metric of activities for which CoT *exclusively* yields correct answers is lower ($\approx 52\times$) compared to the ones for which $C_{\text{pat\_only}}(p)$ exclusively lands correct answers. Compared to the average popularity metric across the entire SPORTS dataset (399k), this is still $\approx 6\times$ lower.

We conjecture that the well-formed structure of thoughts in SPORTS—intertwined patterns and text—empower the model to extract commonsense knowledge about "⟨Activity⟩", even when the activities are infrequently seen during training. On the contrary, the lack of explanatory thoughts in $C_{\text{pat\_only}}(p)$ (See Section 4) disorient the model to arrive at the correct conclusion. Therefore, in these cases, the model arrives at the correct conclusion only when the popularity of "⟨Activity⟩" is significantly high (3,575 k).

**Establishing factually meaningful connections.** While generating correct clauses in thoughts is pivotal to answer a question, establishing factually meaningful connections between these clauses is the final step to attain a correct answer. As mentioned before, the structure of patterns for SPORTS is " *person is a sport$_1$ player.* **activity is** part of **sport$_2$** " (Section 4). If **sport$_1$** and **sport$_2$** are concordant (discordant), the answer is yes (no). Breaking this pattern, as in $C_{\text{pat\_inconsistent}}(p)$ of Section 4, leads to a peculiar failure case. While the model successfully generate correct clauses—the sport that is associated with the player and the corresponding activity—it

Table 17: Patterns are important: error analysis of cases where lack of patterns leads to misleading and incorrect thoughts for various datasets.

---

➥ $\mathcal{Q}1 \mapsto$ Four dogs sat in a line within the veterinarian's waiting room. The first dog was the Affenpinscher, who weighed only 10 pounds. Next to him sat a Dachshund, who weighed twice as much as the Affenpinscher. Next to the Dachshund sat a Papillon, who weighed one-fourth as much as the Dachshund. And at the end of the line sat a Mastiff, who weighed 44 times the weight of the Papillon. How much did the Mastiff weigh, in pounds?

◆ $\mathcal{T}_{\llcorner}^{\ulcorner}\text{CoT(p)}_{\lrcorner}^{\urcorner} \mapsto$ The Affenpinscher weighed 10 pounds. The Dachshund weighed twice as much as the Affenpinscher, so the Dachshund weighed 20 pounds. The Papillon weighed one-fourth as much as the Dachshund, so the Papillon weighed 5 pounds. The Mastiff weighed 44 times as much as the Papillon, so the Mastiff weighed $44 \times 5 = 220$ pounds. The answer is 220 pounds.✓

◆ $\mathcal{T}_{\llcorner}^{\ulcorner}C_{\texttt{pat\_inconsistent}}(p)_{\lrcorner}^{\urcorner} \mapsto$ The Affenpinscher weighed 10 pounds. The Dachshund weighed twice as much as the Affenpinscher, so 20 pounds. The Papillon weighed one-fourth as much as the Dachshund, so 5 pounds. The Mastiff weighed 44 times the weight of the Papillon, so 216 pounds. The answer is 216.✗

➥ $\mathcal{Q}2 \mapsto$ John injured his back and needs to go to physical therapy. He went to physical therapy for 6 weeks. Each week he went twice for 2 hours at a time. If the sessions cost \$125 per hour how much did the physical therapy cost?

◆ $\mathcal{T}_{\llcorner}^{\ulcorner}\text{CoT(p)}_{\lrcorner}^{\urcorner} \mapsto$ John went to physical therapy for 6 weeks. Each week he went twice for 2 hours at a time. So he went 12 times. 12 times 2 hours is 24 hours. 24 hours at \$125 per hour is $24 \times 125 = 3000$. The answer is 3000.✓

◆ $\mathcal{T}_{\llcorner}^{\ulcorner}C_{\texttt{pat\_inconsistent}}(p)_{\lrcorner}^{\urcorner} \mapsto$ John went to physical therapy for 6 weeks. Each week he went twice for 2 hours at a time. So he went 12 times. The answer is 12.✗

➥ $\mathcal{Q}3 \mapsto$ A pet store currently has 5 dogs, 2 cats, and 10 birds. How many legs in total do the pets in the store have?

◆ $\mathcal{T}_{\llcorner}^{\ulcorner}\text{CoT(p)}_{\lrcorner}^{\urcorner} \mapsto$ There are 5 dogs, 2 cats, and 10 birds. 5 dogs have 4 legs each, so 5 * 4 = 20 legs. 2 cats have 4 legs each, so 2 * 4 = 8 legs. 10 birds have 2 legs each, so 10 * 2 = 20 legs. So the total number of legs is 20 + 8 + 20 = 48. The answer is 48.✓

◆ $\mathcal{T}_{\llcorner}^{\ulcorner}C_{\texttt{pat\_inconsistent}}(p)_{\lrcorner}^{\urcorner} \mapsto$ The pet store has 5 dogs, 2 cats, and 10 birds. So the total number of legs is 5 + 2 + 10 = 17. The answer is 17.✗

➥ $\mathcal{Q}4 \mapsto$ Yes or no: Is the following sentence plausible? "Robert Woods killed the powerplay".

◆ $\mathcal{T}_{\llcorner}^{\ulcorner}\text{CoT(p)}_{\lrcorner}^{\urcorner} \mapsto$ Robert Woods is a football player. Killing the powerplay is part of hockey, not football. So the answer is no.✓

◆ $\mathcal{T}_{\llcorner}^{\ulcorner}C_{\texttt{pat\_inconsistent}}(p)_{\lrcorner}^{\urcorner} \mapsto$ Robert Woods is a football player. Killing the powerplay is a part of hockey. So the answer is yes.✗

➥ $\mathcal{Q}5 \mapsto$ The concert was scheduled to be on 06/01/1943, but was delayed by one day to today. What is the date one year ago from today in MM/DD/YYYY?

◆ $\mathcal{T}_{\llcorner}^{\ulcorner}\text{CoT(p)}_{\lrcorner}^{\urcorner} \mapsto$ One day after 06/01/1943 is 06/02/1943, so today is 06/02/1943. One year ago from today is 06/02/1942. So the answer is 06/02/1942.✓

◆ $\mathcal{T}_{\llcorner}^{\ulcorner}C_{\texttt{pat\_inconsistent}}(p)_{\lrcorner}^{\urcorner} \mapsto$ 1 year ago from today is 06/01/1942. So the answer is 06/01/1942.✗

---

fails to connect these clauses in a meaningful way to attain a correct yes/no answer. For example: "*Robert Woods is a football player* (✓). *Killing the powerplay is a part of hockey* (✓). *So the answer is yes.* ✗" This finding underscores another pivotal role that patterns play in prompts. The patterns *reinforce the mechanism* of extracting relevant information followed by establishing factually meaningful connections between the generated clauses.

|  | $C_{\texttt{pat\_only}}(p)$ ✓ | $C_{\texttt{pat\_only}}(p)$ ✗ |
|---|---|---|
| CoT ✓ | 472k | **68k** |
| CoT ✗ | 3,575k | 40k |

Table 18: We use the number of Google search results to approximate the "popularity" of an activity or individual, for cases where CoT and $C_{\texttt{pat\_only}}(p)$ made correct and incorrect conclusions. For each activity, we perform Google search with double quotes. Across the entire SPORTS dataset the average popularity metric for corresponding activities is 399k. When CoT is exclusively right, the average popularity metric is 68k. CoT is able to arrive at factual conclusions for activities that are $4\times$ rarer on the web. We acknowledge that these results can be influenced by several factors, including current events, algorithmic changes, and regional variations. However, despite some limitations, our approach aligns with previous studies that have used Google data to gauge social interest or behavior (Ginsberg et al., 2009; Choi and Varian, 2012).

| Task | Avg. # of Input Tokens | | | Avg. # of Output Tokens | | |
|------|------|------|------|------|------|------|
| | CoT | CCoT | Reduction | CoT | CCoT | Reduction |
| GSM-8K (CoT ↦ Table 20, CCoT ↦Table 73) | 503.25 | 434.25 | 1.16× | 55.51 | 62.02 | 0.90× |
| DATE (CoT ↦ Table 21, CCoT ↦ Table 74) | 321.39 | 276.39 | 1.16× | 21.47 | 15.55 | 1.38× |
| SPORTS (CoT ↦ Table 22, CCoT ↦ Table 75) | 283.97 | 240.97 | 1.18× | 20.04 | 15.27 | 1.31× |
| SORTING (CoT ↦ Table 68, CCoT ↦ Table 23) | 553.01 | 266.01 | 2.08× | 99.34 | 36.16 | 2.75× |
| **Average Reduction** | — | — | **1.39×** | — | — | **1.58×** |

Table 19: Comparison of the average number of input and output tokens for PaLM-62B. The corresponding tables for each task refers to CoT and CCoT prompts, respectively.

# J Complete List of Counterfactual Prompts

This section includes all the counterfactual prompts used in our experiments. Please note that the prompts are added here for quick reference, and can also be located in the accompanying repository (https://github.com/reasoning-machines/prompt-lib).

```python
from itertools import combinations
from itertools import permutations
import numpy as np

def GenerateCombinations(A, r):
    """
    Returns subsequences of elements with length r from array A.
    """
    return list(combinations(A, r))

def main(_):
    base_array = np.arange(10)
    with open('sort_dataset.tsv', 'wt') as sort_ds:
        tsv_writer = csv.writer(sort_ds, delimiter='\t')
        tsv_writer.writerow(['Question', 'Answer', 'Thought'])
        for j in range(2, 10):
            current_array = GenerateCombinations(base_array, j)
            for i in current_array:
                current_permutations = list(permutations(i))
                for s in current_permutations:
                    question = ' , '.join([str(x) for x in s])
                    answer = ' , '.join([str(x) for x in np.sort(list(s))])
                    thought = ' < '.join([str(x) for x in np.sort(list(s))])
                    tsv_writer.writerow([question, answer, thought])
```

Figure 12: Python code to reproduce Sort dataset.

Table 20: GSM-8K prompts used by Wei et al. (2022b).

---

**❖ Prompt 1**

➥ $\mathcal{Q}$ ↦ Shawn has five toys. For Christmas, he got two toys each from his mom and dad. How many toys does he have now?

❖ $\mathcal{T}$ ↦ Shawn started with 5 toys. If he got 2 toys each from his mom and dad, then that is 4 more toys. 5 + 4 = 9.

◆ $\mathcal{A}$ ↦ 9

**❖ Prompt 2**

➥ $\mathcal{Q}$ ↦ If there are 3 cars in the parking lot and 2 more cars arrive, how many cars are in the parking lot?

❖ $\mathcal{T}$ ↦ There are originally 3 cars. 2 more cars arrive. 3 + 2 = 5.

◆ $\mathcal{A}$ ↦ 5

**❖ Prompt 3**

➥ $\mathcal{Q}$ ↦ Jason had 20 lollipops. He gave Denny some lollipops. Now Jason has 12 lollipops. How many lollipops did Jason give to Denny?

❖ $\mathcal{T}$ ↦ Jason started with 20 lollipops. Then he had 12 after giving some to Denny. So he gave Denny 20 - 12 = 8.

◆ $\mathcal{A}$ ↦ 8

**❖ Prompt 4**

➥ $\mathcal{Q}$ ↦ There were nine computers in the server room. Five more computers were installed each day, from Monday to Thursday. How many computers are now in the server room?

❖ $\mathcal{T}$ ↦ There were originally 9 computers. For each of 4 days, 5 more computers were added. So 5 * 4 = 20 computers were added. 9 + 20 is 29.

◆ $\mathcal{A}$ ↦ 29

**❖ Prompt 5**

➥ $\mathcal{Q}$ ↦ There are 15 trees in the grove. Grove workers will plant trees in the grove today. After they are done, there will be 21 trees. How many trees did the grove workers plant today?

❖ $\mathcal{T}$ ↦ There are 15 trees originally. Then there were 21 trees after some more were planted. So there must have been 21 - 15 = 6.

◆ $\mathcal{A}$ ↦ 6

**❖ Prompt 6**

➥ $\mathcal{Q}$ ↦ Leah had 32 chocolates and her sister had 42. If they ate 35, how many pieces do they have left in total?

❖ $\mathcal{T}$ ↦ Originally, Leah had 32 chocolates. Her sister had 42. So in total they had 32 + 42 = 74. After eating 35, they had 74 - 35 = 39.

◆ $\mathcal{A}$ ↦ 39

**❖ Prompt 7**

➥ $\mathcal{Q}$ ↦ Olivia has $23. She bought five bagels for $3 each. How much money does she have left?

❖ $\mathcal{T}$ ↦ Olivia had 23 dollars. 5 bagels for 3 dollars each will be 5 x 3 = 15 dollars. So she has 23 - 15 dollars left. 23 - 15 is 8.

◆ $\mathcal{A}$ ↦ 8

**❖ Prompt 8**

➥ $\mathcal{Q}$ ↦ Michael had 58 golf balls. On Tuesday, he lost 23 golf balls. On Wednesday, he lost 2 more. How many golf balls did he have at the end of Wednesday?

❖ $\mathcal{T}$ ↦ Michael started with 58 golf balls. After losing 23 on Tuesday, he had 58 - 23 = 35. After losing 2 more, he had 35 - 2 = 33 golf balls.

◆ $\mathcal{A}$ ↦ 33

Table 21: DATE prompts used by Wei et al. (2022b).

---

❖ **Prompt 1**

➥ $\mathcal{Q} \mapsto$ It is 4/19/1969 today. What is the date 24 hours later in MM/DD/YYYY?

◈ $\mathcal{T} \mapsto$ Today is 04/19/1969. 24 hours later is one day after today, which would be 04/20/1969.

◆ $\mathcal{A} \mapsto$ 04/20/1969

❖ **Prompt 2**

➥ $\mathcal{Q} \mapsto$ The concert was scheduled to be on 06/01/1943, but was delayed by one day to today. What is the date 10 days ago in MM/DD/YYYY?

◈ $\mathcal{T} \mapsto$ One day after 06/01/1943 is 06/02/1943, so today is 06/02/1943. 10 days before today is 05/23/1943.

◆ $\mathcal{A} \mapsto$ 05/23/1943

❖ **Prompt 3**

➥ $\mathcal{Q} \mapsto$ The first day of 2019 is a Tuesday, and today is the first Monday of 2019. What is the date today in MM/DD/YYYY?

◈ $\mathcal{T} \mapsto$ If the first day of 2019 was Tuesday, then 01/01/2019 was a Tuesday. Today is the first Monday, would be six days later. So today is 01/07/2019.

◆ $\mathcal{A} \mapsto$ 01/07/2019

❖ **Prompt 4**

➥ $\mathcal{Q} \mapsto$ Jane was born on the last day of February in 2001. Today is her 16-year-old birthday. What is the date yesterday in MM/DD/YYYY?

◈ $\mathcal{T} \mapsto$ The last day of February is the 28th, so Jane was born on 02/28/2001. Today is her 16-year old birthday, so today is 02/28/2017. So yesterday was 02/27/2017.

◆ $\mathcal{A} \mapsto$ 02/27/2017

❖ **Prompt 5**

➥ $\mathcal{Q} \mapsto$ 2015 is coming in 36 hours. What is the date one week from today in MM/DD/YYYY?

◈ $\mathcal{T} \mapsto$ If 2015 is coming in 36 hours, then it is coming in 2 days. 2 days before 01/01/2015 is 12/30/2014, so today is 12/30/2014. So one week from today will be 01/05/2015.

◆ $\mathcal{A} \mapsto$ 01/05/2015

❖ **Prompt 6**

➥ $\mathcal{Q} \mapsto$ Jane thought today is 3/11/2002, but today is in fact Mar 12, which is 1 day later. What is the date 24 hours later in MM/DD/YYYY?

◈ $\mathcal{T} \mapsto$ Today is 03/12/2002. So the date 24 hours later will be 03/13/2002.

◆ $\mathcal{A} \mapsto$ 03/13/2002

Table 22: SPORTS prompts used by Wei et al. (2022b).

**❖ Prompt 1**
➥ $\mathcal{Q}$ ↦ Is the following sentence plausible? "Jamal Murray was perfect from the line."
❖ $\mathcal{T}$ ↦ Jamal Murray is a basketball player. Being perfect from the line is part of basketball.
◆ $\mathcal{A}$ ↦ yes

**❖ Prompt 2**
➥ $\mathcal{Q}$ ↦ Is the following sentence plausible? "Joao Moutinho caught the screen pass in the NFC championship."
❖ $\mathcal{T}$ ↦ Joao Moutinho is a soccer player. The NFC championship is part of American football, not soccer.
◆ $\mathcal{A}$ ↦ no

**❖ Prompt 3**
➥ $\mathcal{Q}$ ↦ Is the following sentence plausible? "Jonas Valanciunas beat the buzzer."
❖ $\mathcal{T}$ ↦ Jonas Valanciunas is a basketball player. Beating the buzzer is part of basketball.
◆ $\mathcal{A}$ ↦ yes

**❖ Prompt 4**
➥ $\mathcal{Q}$ ↦ Is the following sentence plausible? "Sam Darnold passed the puck."
❖ $\mathcal{T}$ ↦ Sam Darnold is a American football player. Passing the puck is part of hockey, not American football.
◆ $\mathcal{A}$ ↦ no

**❖ Prompt 5**
➥ $\mathcal{Q}$ ↦ Is the following sentence plausible? "Kyle Palmieri was called for slashing."
❖ $\mathcal{T}$ ↦ Kyle Palmieri is a hockey player. Being called for slashing is part of hockey.
◆ $\mathcal{A}$ ↦ yes

**❖ Prompt 6**
➥ $\mathcal{Q}$ ↦ Is the following sentence plausible? "Carson Wentz set the pick and roll."
❖ $\mathcal{T}$ ↦ Carson Wentz is an American football player. Pick and roll is part of basketball, not football.
◆ $\mathcal{A}$ ↦ no

**❖ Prompt 7**
➥ $\mathcal{Q}$ ↦ Is the following sentence plausible? "Malcolm Brogdon banked the shot in."
❖ $\mathcal{T}$ ↦ Malcolm Brogdon is a basketball player. Banking the shot in is part of basketball.
◆ $\mathcal{A}$ ↦ yes

**❖ Prompt 8**
➥ $\mathcal{Q}$ ↦ Is the following sentence plausible? "Draymond Green threw a touchdown."
❖ $\mathcal{T}$ ↦ Draymond Green is an basketball player. Throwing a touchdown is part of football, not basketball.
◆ $\mathcal{A}$ ↦ no

Table 23: SORTING prompts using code snippet 12.

❖ **Prompt 1**
➥ $\mathcal{Q} \mapsto$ 7 , 8 , 4 , 1 , 2 , 9 , 3 , 6 , 5
◈ $\mathcal{T} \mapsto$ 1 < 2 < 3 < 4 < 5 < 6 < 7 < 8 < 9
◆ $\mathcal{A} \mapsto$ 1 , 2 , 3 , 4 , 5 , 6 , 7 , 8 , 9

❖ **Prompt 2**
➥ $\mathcal{Q} \mapsto$ 5 , 9 , 3 , 1 , 8 , 4 , 6 , 2
◈ $\mathcal{T} \mapsto$ 1 < 2 < 3 < 4 < 5 < 6 < 8 < 9
◆ $\mathcal{A} \mapsto$ 1 , 2 , 3 , 4 , 5 , 6 , 8 , 9

❖ **Prompt 3**
➥ $\mathcal{Q} \mapsto$ 6 , 5 , 7 , 4 , 3 , 2 , 8 , 1
◈ $\mathcal{T} \mapsto$ 1 < 2 < 3 < 4 < 5 < 6 < 7 < 8
◆ $\mathcal{A} \mapsto$ 1 , 2 , 3 , 4 , 5 , 6 , 7 , 8

❖ **Prompt 4**
➥ $\mathcal{Q} \mapsto$ 1 , 6 , 4 , 8 , 5 , 3 , 7 , 2
◈ $\mathcal{T} \mapsto$ 1 < 2 < 3 < 4 < 5 < 6 < 7 < 8
◆ $\mathcal{A} \mapsto$ 1 , 2 , 3 , 4 , 5 , 6 , 7 , 8

❖ **Prompt 5**
➥ $\mathcal{Q} \mapsto$ 5 , 2 , 1 , 4 , 3 , 7
◈ $\mathcal{T} \mapsto$ 1 < 2 < 3 < 4 < 5 < 7
◆ $\mathcal{A} \mapsto$ 1 , 2 , 3 , 4 , 5 , 7

❖ **Prompt 6**
➥ $\mathcal{Q} \mapsto$ 3 , 8 , 2 , 5 , 6 , 4 , 7 , 1
◈ $\mathcal{T} \mapsto$ 1 < 2 < 3 < 4 < 5 < 6 < 7 < 8
◆ $\mathcal{A} \mapsto$ 1 , 2 , 3 , 4 , 5 , 6 , 7 , 8

❖ **Prompt 7**
➥ $\mathcal{Q} \mapsto$ 8 , 6 , 1 , 2 , 9 , 7 , 4
◈ $\mathcal{T} \mapsto$ 1 < 2 < 4 < 6 < 7 < 8 < 9
◆ $\mathcal{A} \mapsto$ 1 , 2 , 4 , 6 , 7 , 8 , 9

❖ **Prompt 8**
➥ $\mathcal{Q} \mapsto$ 7 , 6 , 8 , 1
◈ $\mathcal{T} \mapsto$ 1 < 6 < 7 < 8
◆ $\mathcal{A} \mapsto$ 1 , 6 , 7 , 8

Table 24: DATE with abstract dates.

---

❖ **Prompt 1**

➥ $\mathcal{Q} \mapsto$ It is DATE today. What is the date 24 hours later in MM/DD/YYYY?

◈ $\mathcal{T} \mapsto$ Today is DATE. 24 hours later is one day after today, which would be DATE.

◆ $\mathcal{A} \mapsto$ 04/20/1969

❖ **Prompt 2**

➥ $\mathcal{Q} \mapsto$ The concert was scheduled to be on DATE, but was delayed by one day to today. What is the date 10 days ago in MM/DD/YYYY?

◈ $\mathcal{T} \mapsto$ One day after DATE is DATE, so today is DATE. 10 days before today is 05/23/1943.

◆ $\mathcal{A} \mapsto$ 05/23/1943

❖ **Prompt 3**

➥ $\mathcal{Q} \mapsto$ The first day of 2019 is a Tuesday, and today is the first Monday of 2019. What is the date today in MM/DD/YYYY?

◈ $\mathcal{T} \mapsto$ If the first day of 2019 was Tuesday, then DATE was a Tuesday. Today is the first Monday, would be six days later. So today is 01/07/2019.

◆ $\mathcal{A} \mapsto$ 01/07/2019

❖ **Prompt 4**

➥ $\mathcal{Q} \mapsto$ Jane was born on the last day of February in 2001. Today is her 16-year-old birthday. What is the date yesterday in MM/DD/YYYY?

◈ $\mathcal{T} \mapsto$ The last day of February is the 28th, so Jane was born on DATE. Today is her 16-year old birthday, so today is DATE. So yesterday was 02/27/2017.

◆ $\mathcal{A} \mapsto$ 02/27/2017

❖ **Prompt 5**

➥ $\mathcal{Q} \mapsto$ 2015 is coming in 36 hours. What is the date one week from today in MM/DD/YYYY?

◈ $\mathcal{T} \mapsto$ If 2015 is coming in 36 hours, then it is coming in 2 days. 2 days before DATE is DATE, so today is DATE. So one week from today will be 01/05/2015.

◆ $\mathcal{A} \mapsto$ 01/05/2015

❖ **Prompt 6**

➥ $\mathcal{Q} \mapsto$ Jane thought today is DATE, but today is in fact Mar 12, which is 1 day later. What is the date 24 hours later in MM/DD/YYYY?

◈ $\mathcal{T} \mapsto$ Today is DATE. So the date 24 hours later will be 03/13/2002.

◆ $\mathcal{A} \mapsto$ 03/13/2002

Table 25: GSM-8ᴋ after replacing numbers with Greek letters.

---

**❖ Prompt 1**

➥ $\mathcal{Q} \mapsto$ Shawn has $\alpha$ toys. For Christmas, he got $\beta$ toys each from his mom and dad. How many toys does he have now?

◈ $\mathcal{T} \mapsto$ Shawn started with $\alpha$ toys. If he got $\beta$ toys each from his mom and dad, then that is $\lambda$ more toys. $\alpha + \lambda = \pi$.

◆ $\mathcal{A} \mapsto \pi$

**❖ Prompt 2**

➥ $\mathcal{Q} \mapsto$ If there are $\alpha$ cars in the parking lot and $\beta$ more cars arrive, how many cars are in the parking lot?

◈ $\mathcal{T} \mapsto$ There are originally $\alpha$ cars. $\beta$ more cars arrive. $\alpha + \beta = \lambda$.

◆ $\mathcal{A} \mapsto \lambda$

**❖ Prompt 3**

➥ $\mathcal{Q} \mapsto$ Jason had $\alpha$ lollipops. He gave Denny some lollipops. Now Jason has $\beta$ lollipops. How many lollipops did Jason give to Denny?

◈ $\mathcal{T} \mapsto$ Jason started with $\alpha$ lollipops. Then he had $\beta$ after giving some to Denny. So he gave Denny $\alpha - \beta = \lambda$.

◆ $\mathcal{A} \mapsto \lambda$

**❖ Prompt 4**

➥ $\mathcal{Q} \mapsto$ There were $\alpha$ computers in the server room. $\beta$ more computers were installed each day, from Monday to Thursday. How many computers are now in the server room?

◈ $\mathcal{T} \mapsto$ There were originally $\alpha$ computers. For each of four days, $\beta$ more computers were added. So $\beta$ * four = $\lambda$ computers were added. $\alpha + \lambda$ is $\pi$.

◆ $\mathcal{A} \mapsto \pi$

**❖ Prompt 5**

➥ $\mathcal{Q} \mapsto$ There are $\alpha$ trees in the grove. Grove workers will plant trees in the grove today. After they are done, there will be $\beta$ trees. How many trees did the grove workers plant today?

◈ $\mathcal{T} \mapsto$ There are $\alpha$ trees originally. Then there were $\beta$ trees after some more were planted. So there must have been $\beta - \alpha = \lambda$.

◆ $\mathcal{A} \mapsto \lambda$

**❖ Prompt 6**

➥ $\mathcal{Q} \mapsto$ Leah had $\alpha$ chocolates and her sister had $\beta$. If they ate $\lambda$, how many pieces do they have left in total?

◈ $\mathcal{T} \mapsto$ Originally, Leah had $\alpha$ chocolates. Her sister had $\beta$. So in total they had $\alpha + \beta = \pi$. After eating $\lambda$, they had $\pi - \lambda = \mu$.

◆ $\mathcal{A} \mapsto \mu$

**❖ Prompt 7**

➥ $\mathcal{Q} \mapsto$ Olivia has $\alpha$. She bought five bagels for $\beta$ each. How much money does she have left?

◈ $\mathcal{T} \mapsto$ Olivia had $\alpha$ dollars. 5 bagels for $\beta$ dollars each will be 5 x $\beta = \lambda$ dollars. So she has $\alpha - \lambda$ dollars left. $\alpha - \lambda$ is $\pi$.

◆ $\mathcal{A} \mapsto \pi$

**❖ Prompt 8**

➥ $\mathcal{Q} \mapsto$ Michael had $\alpha$ golf balls. On Tuesday, he lost $\beta$ golf balls. On Wednesday, he lost $\lambda$ more. How many golf balls did he have at the end of Wednesday?

◈ $\mathcal{T} \mapsto$ Michael started with $\alpha$ golf balls. After losing $\beta$ on Tuesday, he had $\alpha - \beta = \pi$. After losing $\lambda$ more, he had $\pi - \lambda = \mu$ golf balls.

◆ $\mathcal{A} \mapsto \mu$

---

Table 26: SORTING after replacing numbers with Greek letters.

**✤ Prompt 1**

➥ $\mathcal{Q} \mapsto \upsilon, \delta, \zeta, \phi, \pi, \gamma, \varsigma, \epsilon, \chi$

◈ $\mathcal{T} \mapsto \varsigma < \phi < \gamma < \delta < \zeta < \chi < \epsilon < \pi < \upsilon$

◆ $\mathcal{A} \mapsto \varsigma, \phi, \gamma, \delta, \zeta, \chi, \epsilon, \pi, \upsilon$

**✤ Prompt 2**

➥ $\mathcal{Q} \mapsto \nu, \iota, \lambda, \zeta, \xi, \gamma, \psi, \pi$

◈ $\mathcal{T} \mapsto \xi < \zeta < \lambda < \pi < \psi < \gamma < \iota < \nu$

◆ $\mathcal{A} \mapsto \xi, \zeta, \lambda, \pi, \psi, \gamma, \iota, \nu$

**✤ Prompt 3**

➥ $\mathcal{Q} \mapsto \xi, \tau, \phi, \theta, \zeta, \eta, \iota, \delta$

◈ $\mathcal{T} \mapsto \xi < \eta < \theta < \phi < \zeta < \delta < \iota < \tau$

◆ $\mathcal{A} \mapsto \xi, \eta, \theta, \phi, \zeta, \delta, \iota, \tau$

**✤ Prompt 4**

➥ $\mathcal{Q} \mapsto \zeta, \pi, \beta, \sigma, \phi, \epsilon, \lambda, o$

◈ $\mathcal{T} \mapsto \epsilon < \zeta < \pi < \lambda < \phi < \beta < o < \sigma$

◆ $\mathcal{A} \mapsto \epsilon, \zeta, \pi, \lambda, \phi, \beta, o, \sigma$

**✤ Prompt 5**

➥ $\mathcal{Q} \mapsto \omega, o, \delta, \iota, \upsilon, \nu$

◈ $\mathcal{T} \mapsto \omega < o < \nu < \delta < \iota < \upsilon$

◆ $\mathcal{A} \mapsto \omega, o, \nu, \delta, \iota, \upsilon$

**✤ Prompt 6**

➥ $\mathcal{Q} \mapsto \mu, \zeta, \pi, \chi, \lambda, \varsigma, \alpha, \gamma$

◈ $\mathcal{T} \mapsto \varsigma < \pi < \gamma < \lambda < \zeta < \chi < \mu < \alpha$

◆ $\mathcal{A} \mapsto \varsigma, \pi, \gamma, \lambda, \zeta, \chi, \mu, \alpha$

**✤ Prompt 7**

➥ $\mathcal{Q} \mapsto \tau, \gamma, \lambda, \mu, \alpha, \varsigma, \kappa$

◈ $\mathcal{T} \mapsto \mu < \lambda < \alpha < \tau < \kappa < \gamma < \varsigma$

◆ $\mathcal{A} \mapsto \mu, \lambda, \alpha, \tau, \kappa, \gamma, \varsigma$

**✤ Prompt 8**

➥ $\mathcal{Q} \mapsto \kappa, \xi, \nu, \delta$

◈ $\mathcal{T} \mapsto \kappa < \xi < \delta < \nu$

◆ $\mathcal{A} \mapsto \kappa, \xi, \delta, \nu$

Table 27: SPORTS with abstract sports person.

❖ **Prompt 1**

➡ $\mathcal{Q} \mapsto$ Is the following sentence plausible? "PERSON was perfect from the line."

❖ $\mathcal{T} \mapsto$ PERSON is a basketball player. Being perfect from the line is part of basketball.

◆ $\mathcal{A} \mapsto$ yes

❖ **Prompt 2**

➡ $\mathcal{Q} \mapsto$ Is the following sentence plausible? "PERSON caught the screen pass in the NFC championship."

❖ $\mathcal{T} \mapsto$ PERSON is a soccer player. The NFC championship is part of American football, not soccer.

◆ $\mathcal{A} \mapsto$ no

❖ **Prompt 3**

➡ $\mathcal{Q} \mapsto$ Is the following sentence plausible? "PERSON beat the buzzer."

❖ $\mathcal{T} \mapsto$ PERSON is a basketball player. Beating the buzzer is part of basketball.

◆ $\mathcal{A} \mapsto$ yes

❖ **Prompt 4**

➡ $\mathcal{Q} \mapsto$ Is the following sentence plausible? "PERSON passed the puck."

❖ $\mathcal{T} \mapsto$ PERSON is a American football player. Passing the puck is part of hockey, not American football.

◆ $\mathcal{A} \mapsto$ no

❖ **Prompt 5**

➡ $\mathcal{Q} \mapsto$ Is the following sentence plausible? "PERSON was called for slashing."

❖ $\mathcal{T} \mapsto$ PERSON is a hockey player. Being called for slashing is part of hockey.

◆ $\mathcal{A} \mapsto$ yes

❖ **Prompt 6**

➡ $\mathcal{Q} \mapsto$ Is the following sentence plausible? "PERSON set the pick and roll."

❖ $\mathcal{T} \mapsto$ PERSON is an American football player. Pick and roll is part of basketball, not football.

◆ $\mathcal{A} \mapsto$ no

❖ **Prompt 7**

➡ $\mathcal{Q} \mapsto$ Is the following sentence plausible? "PERSON banked the shot in."

❖ $\mathcal{T} \mapsto$ PERSON is a basketball player. Banking the shot in is part of basketball.

◆ $\mathcal{A} \mapsto$ yes

❖ **Prompt 8**

➡ $\mathcal{Q} \mapsto$ Is the following sentence plausible? "PERSON threw a touchdown."

❖ $\mathcal{T} \mapsto$ PERSON is an basketball player. Throwing a touchdown is part of football, not basketball.

◆ $\mathcal{A} \mapsto$ no

Table 28: SPORTS with abstract sports activity.

---

**♦ Prompt 1**

➥ $\mathcal{Q}$ ↦ Is the following sentence plausible? "Jamel Murray was ACTIVITY."

♦ $\mathcal{T}$ ↦ Jamal Murray is a basketball player. Being ACTIVITY is part of basketball.

♦ $\mathcal{A}$ ↦ yes

**♦ Prompt 2**

➥ $\mathcal{Q}$ ↦ Is the following sentence plausible? "Joao Moutinho caught ACTIVITY."

♦ $\mathcal{T}$ ↦ Joao Moutinho is a soccer player. The ACTIVITY is part of American football, not soccer.

♦ $\mathcal{A}$ ↦ no

**♦ Prompt 3**

➥ $\mathcal{Q}$ ↦ Is the following sentence plausible? "Jonas Valanciunas ACTIVITY."

♦ $\mathcal{T}$ ↦ Jonas Valanciunas is a basketball player. ACTIVITY is part of basketball.

♦ $\mathcal{A}$ ↦ yes

**♦ Prompt 4**

➥ $\mathcal{Q}$ ↦ Is the following sentence plausible? "Sam Darnold ACTIVITY."

♦ $\mathcal{T}$ ↦ Sam Darnold is a American football player. ACTIVITY is part of hockey, not American football.

♦ $\mathcal{A}$ ↦ no

**♦ Prompt 5**

➥ $\mathcal{Q}$ ↦ Is the following sentence plausible? "Kyle Palmieri was ACTIVITY."

♦ $\mathcal{T}$ ↦ Kyle Palmieri is a hockey player. Being ACTIVITY is part of hockey.

♦ $\mathcal{A}$ ↦ yes

**♦ Prompt 6**

➥ $\mathcal{Q}$ ↦ Is the following sentence plausible? "Carson Wentz set the ACTIVITY."

♦ $\mathcal{T}$ ↦ Carson Wentz is an American football player. ACTIVITY is part of basketball, not football.

♦ $\mathcal{A}$ ↦ no

**♦ Prompt 7**

➥ $\mathcal{Q}$ ↦ Is the following sentence plausible? "Malcolm Brogdon ACTIVITY."

♦ $\mathcal{T}$ ↦ Malcolm Brogdon is a basketball player. ACTIVITY is part of basketball.

♦ $\mathcal{A}$ ↦ yes

**♦ Prompt 8**

➥ $\mathcal{Q}$ ↦ Is the following sentence plausible? "Draymond Green ACTIVITY."

♦ $\mathcal{T}$ ↦ Draymond Green is an basketball player. ACTIVITY is part of football, not basketball.

♦ $\mathcal{A}$ ↦ no

Table 29: SPORTS with abstract sport.

---

❖**Prompt 1**

➥ $\mathcal{Q} \mapsto$ Is the following sentence plausible? "Jamel Murray was perfect from the line."

❖ $\mathcal{T} \mapsto$ Jamal Murray is a SPORT1 player. Being perfect from the line is part of SPORT1.

◆ $\mathcal{A} \mapsto$ yes

---

❖**Prompt 2**

➥ $\mathcal{Q} \mapsto$ Is the following sentence plausible? "Joao Moutinho caught the screen pass in the NFC championship."

❖ $\mathcal{T} \mapsto$ Joao Moutinho is a SPORT2 player. The NFC championship is part of SPORT3, not SPORT2.

◆ $\mathcal{A} \mapsto$ no

---

❖**Prompt 3**

➥ $\mathcal{Q} \mapsto$ Is the following sentence plausible? "Jonas Valanciunas beat the buzzer."

❖ $\mathcal{T} \mapsto$ Jonas Valanciunas is a SPORT1 player. Beating the buzzer is part of SPORT1.

◆ $\mathcal{A} \mapsto$ yes

---

❖**Prompt 4**

➥ $\mathcal{Q} \mapsto$ Is the following sentence plausible? "Sam Darnold passed the puck."

❖ $\mathcal{T} \mapsto$ Sam Darnold is a SPORT3 player. Passing the puck is part of SPORT4, not SPORT3.

◆ $\mathcal{A} \mapsto$ no

---

❖**Prompt 5**

➥ $\mathcal{Q} \mapsto$ Is the following sentence plausible? "Kyle Palmieri was called for slashing."

❖ $\mathcal{T} \mapsto$ Kyle Palmieri is a SPORT4 player. Being called for slashing is part of SPORT4.

◆ $\mathcal{A} \mapsto$ yes

---

❖**Prompt 6**

➥ $\mathcal{Q} \mapsto$ Is the following sentence plausible? "Carson Wentz set the pick and roll."

❖ $\mathcal{T} \mapsto$ Carson Wentz is an SPORT3 player. Pick and roll is part of SPORT1, not SPORT3.

◆ $\mathcal{A} \mapsto$ no

---

❖**Prompt 7**

➥ $\mathcal{Q} \mapsto$ Is the following sentence plausible? "Malcolm Brogdon banked the shot in."

❖ $\mathcal{T} \mapsto$ Malcolm Brogdon is a SPORT1 player. Banking the shot in is part of SPORT1.

◆ $\mathcal{A} \mapsto$ yes

---

❖**Prompt 8**

➥ $\mathcal{Q} \mapsto$ Is the following sentence plausible? "Draymond Green threw a touchdown."

❖ $\mathcal{T} \mapsto$ Draymond Green is an SPORT1 player. Throwing a touchdown is part of SPORT3, not SPORT1.

◆ $\mathcal{A} \mapsto$ no

---

Table 30: GSM-8ᴋ with decimal numbers.

❖ **Prompt 1**

➥ $\mathcal{Q} \mapsto$ Shawn has five and a half toys. For Christmas, he got two and a half toys each from his mom and dad. How many toys does he have now?

❖ $\mathcal{T} \mapsto$ Shawn started with 5.5 toys. If he got 2.5 toys each from his mom and dad, then that is 5 more toys. 5.5 + 5 = 10.5.

◆ $\mathcal{A} \mapsto$ 10.5

❖ **Prompt 2**

➥ $\mathcal{Q} \mapsto$ If there are 3.3 cars in the parking lot and 2.8 more cars arrive, how many cars are in the parking lot?

❖ $\mathcal{T} \mapsto$ There are originally 3.3 cars. 2.8 more cars arrive. 3.3 + 2.8 = 6.1.

◆ $\mathcal{A} \mapsto$ 6.1

❖ **Prompt 3**

➥ $\mathcal{Q} \mapsto$ Jason had 20.2 lollipops. He gave Denny some lollipops. Now Jason has 15.5 lollipops. How many lollipops did Jason give to Denny?

❖ $\mathcal{T} \mapsto$ Jason started with 20.2 lollipops. Then he had 15.5 after giving some to Denny. So he gave Denny 20.2 - 15.5 = 4.7.

◆ $\mathcal{A} \mapsto$ 4.7

❖ **Prompt 4**

➥ $\mathcal{Q} \mapsto$ There were nine and a quarter computers in the server room. Five and three quarters more computers were installed each day, from Monday to Thursday. How many computers are now in the server room?

❖ $\mathcal{T} \mapsto$ There were originally 9.25 computers. For each of 4 days, 5.75 more computers were added. So 5.75 * 4 = 23 computers were added. 9.25 + 23 is 32.25.

◆ $\mathcal{A} \mapsto$ 32.25

❖ **Prompt 5**

➥ $\mathcal{Q} \mapsto$ There are 15.3 trees in the grove. Grove workers will plant trees in the grove today. After they are done, there will be 16.5 trees. How many trees did the grove workers plant today?

❖ $\mathcal{T} \mapsto$ There are 15.3 trees originally. Then there were 16.5 trees after some more were planted. So there must have been 16.5 - 15.3 = 1.2.

◆ $\mathcal{A} \mapsto$ 1.2

❖ **Prompt 6**

➥ $\mathcal{Q} \mapsto$ Leah had 3.2 chocolates and her sister had 4.2. If they ate 3.5, how many pieces do they have left in total?

❖ $\mathcal{T} \mapsto$ Originally, Leah had 3.2 chocolates. Her sister had 4.2. So in total they had 3.2 + 4.2 = 7.4. After eating 3.5, they had 7.4 - 3.5 = 3.9.

◆ $\mathcal{A} \mapsto$ 3.9

❖ **Prompt 7**

➥ $\mathcal{Q} \mapsto$ Olivia has \$2.3. She bought five bagels for \$0.3 each. How much money does she have left?

❖ $\mathcal{T} \mapsto$ Olivia had 2.3 dollars. 5 bagels for .3 dollars each will be 5 x .3 = 1.5 dollars. So she has 2.3 - 1.5 dollars left. 2.3 - 1.5 is 0.8.

◆ $\mathcal{A} \mapsto$ 0.8

❖ **Prompt 8**

➥ $\mathcal{Q} \mapsto$ Michael had 5.8 golf balls. On Tuesday, he lost 2.3 golf balls. On Wednesday, he lost 0.2 more. How many golf balls did he have at the end of Wednesday?

❖ $\mathcal{T} \mapsto$ Michael started with 5.8 golf balls. After losing 2.3 on Tuesday, he had 5.8 - 2.3 = 3.5. After losing 0.2 more, he had 3.5 - 0.2 = 3.3 golf balls.

◆ $\mathcal{A} \mapsto$ 3.3

Table 31: DATE with future dates.

---

❖ **Prompt 1**

➥ $\mathcal{Q}$ ↦ It is 4/30/3069 today. What is the date 24 hours later in MM/DD/YYYY?

◆ $\mathcal{T}$ ↦ Today is 04/30/3069. 24 hours later is one day after today, which would be 04/31/3069.

◆ $\mathcal{A}$ ↦ 04/31/3069

❖ **Prompt 2**

➥ $\mathcal{Q}$ ↦ The concert was scheduled to be on 06/01/3043, but was delayed by one day to today. What is the date 10 days ago in MM/DD/YYYY?

◆ $\mathcal{T}$ ↦ One day after 06/01/3043 is 06/02/3043, so today is 06/02/3043. 10 days before today is 05/23/3043.

◆ $\mathcal{A}$ ↦ 05/23/3043

❖ **Prompt 3**

➥ $\mathcal{Q}$ ↦ The first day of 3130 is a Tuesday, and today is the first Monday of 3130. What is the date today in MM/DD/YYYY?

◆ $\mathcal{T}$ ↦ If the first day of 3130 was Tuesday, then 01/01/3130 was a Tuesday. Today is the first Monday, would be six days later. So today is 01/07/3130.

◆ $\mathcal{A}$ ↦ 01/07/3130

❖ **Prompt 4**

➥ $\mathcal{Q}$ ↦ Jane was born on the last day of February in 3101. Today is her 16-year-old birthday. What is the date yesterday in MM/DD/YYYY?

◆ $\mathcal{T}$ ↦ The last day of February is the 28th, so Jane was born on 02/28/3101. Today is her 16-year old birthday, so today is 02/28/3117. So yesterday was 02/27/3117.

◆ $\mathcal{A}$ ↦ 02/27/3117

❖ **Prompt 5**

➥ $\mathcal{Q}$ ↦ 3115 is coming in 36 hours. What is the date one week from today in MM/DD/YYYY?

◆ $\mathcal{T}$ ↦ If 3115 is coming in 36 hours, then it is coming in 2 days. 2 days before 01/01/3115 is 12/30/3114, so today is 12/30/3114. So one week from today will be 01/05/3115.

◆ $\mathcal{A}$ ↦ 01/05/3115

❖ **Prompt 6**

➥ $\mathcal{Q}$ ↦ Jane thought today is 3/11/3102, but today is in fact Mar 12, which is 1 day later. What is the date 24 hours later in MM/DD/YYYY?

◆ $\mathcal{T}$ ↦ Today is 03/12/3102. So the date 24 hours later will be 03/13/3102.

◆ $\mathcal{A}$ ↦ 03/13/3102

Table 32: SPORTS with non-sport names and activities.

---

❖ **Prompt 1**

➥ $\mathcal{Q} \mapsto$ Is the following sentence plausible? "Adair Foster was juggling the paper cups."

❖ $\mathcal{T} \mapsto$ Adair Foster is a basketball player. Juggling the paper cups is part of basketball.

◆ $\mathcal{A} \mapsto$ yes

---

❖ **Prompt 2**

➥ $\mathcal{Q} \mapsto$ Is the following sentence plausible? "Scottie Colby caught the hot potato in the NFC championship."

❖ $\mathcal{T} \mapsto$ Scottie Colby is a soccer player. The NFC championship is part of American football, not soccer.

◆ $\mathcal{A} \mapsto$ no

---

❖ **Prompt 3**

➥ $\mathcal{Q} \mapsto$ Is the following sentence plausible? "Chas Jase beat the buzzer."

❖ $\mathcal{T} \mapsto$ Chas Jase is a basketball player. Beating the pillow is part of basketball.

◆ $\mathcal{A} \mapsto$ yes

---

❖ **Prompt 4**

➥ $\mathcal{Q} \mapsto$ Is the following sentence plausible? "Denny Dillan passed the soda."

❖ $\mathcal{T} \mapsto$ Denny Dillan is a American football player. Passing the soda is part of hockey, not American football.

◆ $\mathcal{A} \mapsto$ no

---

❖ **Prompt 5**

➥ $\mathcal{Q} \mapsto$ Is the following sentence plausible? "Myron Storm was called for trashing."

❖ $\mathcal{T} \mapsto$ Myron Storm is a hockey player. Being called for trashing is part of hockey.

◆ $\mathcal{A} \mapsto$ yes

---

❖ **Prompt 6**

➥ $\mathcal{Q} \mapsto$ Is the following sentence plausible? "Cliff Kristopher set the tick and floor."

❖ $\mathcal{T} \mapsto$ Cliff Kristopher is an American football player. Tick and floor is part of basketball, not football.

◆ $\mathcal{A} \mapsto$ no

---

❖ **Prompt 7**

➥ $\mathcal{Q} \mapsto$ Is the following sentence plausible? "Celestine Holden banked the pot in."

❖ $\mathcal{T} \mapsto$ Celestine Holden is a basketball player. Banking the pot in is part of basketball.

◆ $\mathcal{A} \mapsto$ yes

---

❖ **Prompt 8**

➥ $\mathcal{Q} \mapsto$ Is the following sentence plausible? "Peter Abraham threw a tantrum."

❖ $\mathcal{T} \mapsto$ Peter Abraham is an basketball player. Throwing a tantrum is part of football, not basketball.

◆ $\mathcal{A} \mapsto$ no

---

Table 33: SORTING with out-of-distribution numbers.

❖ **Prompt 1**
➥ $\mathcal{Q} \mapsto$ 72 , 85 , 48 , 11 , 23 , 95 , 34 , 63 , 56
◈ $\mathcal{T} \mapsto$ 11 < 23 < 34 < 48 < 56 < 63 < 72 < 85 < 95
◆ $\mathcal{A} \mapsto$ 11 , 23 , 34 , 48 , 56 , 63 , 72 , 85 , 95

❖ **Prompt 2**
➥ $\mathcal{Q} \mapsto$ 56 , 95 , 34 , 11 , 85 , 48 , 63 , 23
◈ $\mathcal{T} \mapsto$ 11 < 23 < 34 < 48 < 56 < 63 < 85 < 95
◆ $\mathcal{A} \mapsto$ 11 , 23 , 34 , 48 , 56 , 63 , 85 , 95

❖ **Prompt 3**
➥ $\mathcal{Q} \mapsto$ 63 , 56 , 72 , 48 , 34 , 23 , 85 , 11
◈ $\mathcal{T} \mapsto$ 11 < 23 < 34 < 48 < 56 < 63 < 72 < 85
◆ $\mathcal{A} \mapsto$ 11 , 23 , 34 , 48 , 56 , 63 , 72 , 85

❖ **Prompt 4**
➥ $\mathcal{Q} \mapsto$ 11 , 63 , 48 , 85 , 56 , 34 , 72 , 23
◈ $\mathcal{T} \mapsto$ 11 < 23 < 34 < 48 < 56 < 63 < 72 < 85
◆ $\mathcal{A} \mapsto$ 11 , 23 , 34 , 48 , 56 , 63 , 72 , 85

❖ **Prompt 5**
➥ $\mathcal{Q} \mapsto$ 56 , 23 , 11 , 48 , 34 , 72
◈ $\mathcal{T} \mapsto$ 11 < 23 < 34 < 48 < 56 < 72
◆ $\mathcal{A} \mapsto$ 11 , 23 , 34 , 48 , 56 , 72

❖ **Prompt 6**
➥ $\mathcal{Q} \mapsto$ 34 , 85 , 23 , 56 , 63 , 48 , 72 , 11
◈ $\mathcal{T} \mapsto$ 11 < 23 < 34 < 48 < 56 < 63 < 72 < 85
◆ $\mathcal{A} \mapsto$ 11 , 23 , 34 , 48 , 56 , 63 , 72 , 85

❖ **Prompt 7**
➥ $\mathcal{Q} \mapsto$ 85 , 63 , 11 , 23 , 95 , 72 , 48
◈ $\mathcal{T} \mapsto$ 11 < 23 < 48 < 63 < 72 < 85 < 95
◆ $\mathcal{A} \mapsto$ 11 , 23 , 48 , 63 , 72 , 85 , 95

❖ **Prompt 8**
➥ $\mathcal{Q} \mapsto$ 72 , 63 , 85 , 11
◈ $\mathcal{T} \mapsto$ 11 < 63 < 72 < 85
◆ $\mathcal{A} \mapsto$ 11 , 63 , 72 , 85

Table 34: GSM-8K with negative numbers.

---

❖ **Prompt 1**

➥ $\mathcal{Q}$ ↦ Shawn has five toys. For Christmas, he got minus two toys each from his mom and dad. How many toys does he have now?

◈ $\mathcal{T}$ ↦ Shawn started with 5 toys. If he got -2 toys each from his mom and dad, then that is -4 more toys. 5 + -4 = 1.

◆ $\mathcal{A}$ ↦ 1

❖ **Prompt 2**

➥ $\mathcal{Q}$ ↦ If there are 3 cars in the parking lot and minus 2 more cars arrive, how many cars are in the parking lot?

◈ $\mathcal{T}$ ↦ There are originally 3 cars. -2 more cars arrive. 3 - 2 = 1.

◆ $\mathcal{A}$ ↦ 1

❖ **Prompt 3**

➥ $\mathcal{Q}$ ↦ Jason had 20 lollipops. He gave Denny some lollipops. Now Jason has -2 lollipops. How many lollipops did Jason give to Denny?

◈ $\mathcal{T}$ ↦ Jason started with 20 lollipops. Then he had -2 after giving some to Denny. So he gave Denny 20 - -2 = 22.

◆ $\mathcal{A}$ ↦ 22

❖ **Prompt 4**

➥ $\mathcal{Q}$ ↦ There were fifteen computers in the server room. Two computers were uninstalled each day, from Monday to Thursday. How many computers are now in the server room?

◈ $\mathcal{T}$ ↦ There were originally 15 computers. For each of 4 days, 2 computers were removed. So 2 * 4 = 8 computers were removed. 15 - 8 is 7.

◆ $\mathcal{A}$ ↦ 7

❖ **Prompt 5**

➥ $\mathcal{Q}$ ↦ There are 15 trees in the grove. Grove workers will uproot trees in the grove today. After they are done, there will be 10 trees. How many trees did the grove workers uproot today?

◈ $\mathcal{T}$ ↦ There are 15 trees originally. Then there were 10 trees after some were uprooted. So the number that is uprooted must have been 10, as 15 - 5 = 10.

◆ $\mathcal{A}$ ↦ 5

❖ **Prompt 6**

➥ $\mathcal{Q}$ ↦ Leah had 32 chocolates and her sister had 42. If they ate 75, how many pieces of chocolate did they have to borrow in total?

◈ $\mathcal{T}$ ↦ Originally, Leah had 32 chocolates. Her sister had 42. So in total they had 32 + 42 = 74. Since they ate 75, they had to borrow 75 - 74 = 1.

◆ $\mathcal{A}$ ↦ 1

❖ **Prompt 7**

➥ $\mathcal{Q}$ ↦ Olivia has $23. She wants to buy 10 bagels for $3 each. How much more money does she need?

◈ $\mathcal{T}$ ↦ Olivia has 23 dollars. 10 bagels for 3 dollars each will be 10 x 3 = 30 dollars. So she is short of 23 - 30 = -7 dollars. 30 - 23 is 7.

◆ $\mathcal{A}$ ↦ 7

❖ **Prompt 8**

➥ $\mathcal{Q}$ ↦ Michael reserved 58 golf balls. On Tuesday, he lost 57 golf balls. On Wednesday, he lost 2 more. How many more golf balls did he lose than his reserved quota at the end of Wednesday?

◈ $\mathcal{T}$ ↦ Michael started with 58 golf balls. After losing 57 on Tuesday, he had 58 - 57 = 1. After losing 2 more, he had 1 - 2 = -1 golf balls. So he lost 2 - 1 = 1 more balls.

◆ $\mathcal{A}$ ↦ 1

---

Table 35: GSM-8K with larger numbers.

---

**❖ Prompt 1**

➥ 𝒬 ↦ Shawn has three fifty nine toys. For Christmas, he got seventy one toys each from his mom and dad. How many toys does he have now?

◈ 𝒯 ↦ Shawn started with 359 toys. If he got 71 toys each from his mom and dad, then that is 142 more toys. 359 + 142 = 501.

◆ 𝒜 ↦ 501

**❖ Prompt 2**

➥ 𝒬 ↦ If there are 987 cars in the parking lot and 659 more cars arrive, how many cars are in the parking lot?

◈ 𝒯 ↦ There are originally 987 cars. 659 more cars arrive. 987 + 659 = 1646.

◆ 𝒜 ↦ 1646

**❖ Prompt 3**

➥ 𝒬 ↦ Jason had 813 lollipops. He gave Denny some lollipops. Now Jason has 582 lollipops. How many lollipops did Jason give to Denny?

◈ 𝒯 ↦ Jason started with 813 lollipops. Then he had 582 after giving some to Denny. So he gave Denny 813 - 582 = 231.

◆ 𝒜 ↦ 231

**❖ Prompt 4**

➥ 𝒬 ↦ There were six hundred seventy seven computers in the server room. Five hundred twenty two more computers were installed each day, from Monday to Thursday. How many computers are now in the server room?

◈ 𝒯 ↦ There were originally 677 computers. For each of 4 days, 522 more computers were added. So 522 * 4 = 2088 computers were added. 677 + 2088 is 29.

◆ 𝒜 ↦ 2765

**❖ Prompt 5**

➥ 𝒬 ↦ There are 715 trees in the grove. Grove workers will plant trees in the grove today. After they are done, there will be 803 trees. How many trees did the grove workers plant today?

◈ 𝒯 ↦ There are 715 trees originally. Then there were 803 trees after some more were planted. So there must have been 803 - 715 = 88.

◆ 𝒜 ↦ 88

**❖ Prompt 6**

➥ 𝒬 ↦ Leah had 732 chocolates and her sister had 642. If they ate 35, how many pieces do they have left in total?

◈ 𝒯 ↦ Originally, Leah had 732 chocolates. Her sister had 642. So in total they had 32 + 42 = 1374. After eating 35, they had 1374 - 35 = 1339.

◆ 𝒜 ↦ 1339

**❖ Prompt 7**

➥ 𝒬 ↦ Olivia has $323. She bought five bagels for $39 each. How much money does she have left?

◈ 𝒯 ↦ Olivia had 323 dollars. 5 bagels for 39 dollars each will be 5 x 3 = 195 dollars. So she has 323 - 195 dollars left. 323 - 195 is 128.

◆ 𝒜 ↦ 128

**❖ Prompt 8**

➥ 𝒬 ↦ Michael had 958 golf balls. On Tuesday, he lost 323 golf balls. On Wednesday, he lost 259 more. How many golf balls did he have at the end of Wednesday?

◈ 𝒯 ↦ Michael started with 958 golf balls. After losing 323 on Tuesday, he had 958 - 323 = 665. After losing 259 more, he had 665 - 259 = 406 golf balls.

◆ 𝒜 ↦ 406

Table 36: DATE with wrong answers.

---

❖ **Prompt 1**

➥ $\mathcal{Q}$ ↦ It is 4/19/1969 today. What is the date 24 hours later in MM/DD/YYYY?

◆ $\mathcal{T}$ ↦ Today is 04/19/1969. 24 hours later is one day after today, which would be 03/20/1969.

◆ $\mathcal{A}$ ↦ 04/20/1969

❖ **Prompt 2**

➥ $\mathcal{Q}$ ↦ The concert was scheduled to be on 06/01/1943, but was delayed by one day to today. What is the date 10 days ago in MM/DD/YYYY?

◆ $\mathcal{T}$ ↦ One day after 06/01/1943 is 06/02/1943, so today is 06/02/1943. 10 days before today is 05/12/1943.

◆ $\mathcal{A}$ ↦ 05/23/1943

❖ **Prompt 3**

➥ $\mathcal{Q}$ ↦ The first day of 2019 is a Tuesday, and today is the first Monday of 2019. What is the date today in MM/DD/YYYY?

◆ $\mathcal{T}$ ↦ If the first day of 2019 was Tuesday, then 01/01/2019 was a Tuesday. Today is the first Monday, would be six days later. So today is 01/07/2009.

◆ $\mathcal{A}$ ↦ 01/07/2019

❖ **Prompt 4**

➥ $\mathcal{Q}$ ↦ Jane was born on the last day of February in 2001. Today is her 16-year-old birthday. What is the date yesterday in MM/DD/YYYY?

◆ $\mathcal{T}$ ↦ The last day of February is the 28th, so Jane was born on 02/28/2001. Today is her 16-year old birthday, so today is 02/28/2017. So yesterday was 03/27/2017.

◆ $\mathcal{A}$ ↦ 02/27/2017

❖ **Prompt 5**

➥ $\mathcal{Q}$ ↦ 2015 is coming in 36 hours. What is the date one week from today in MM/DD/YYYY?

◆ $\mathcal{T}$ ↦ If 2015 is coming in 36 hours, then it is coming in 2 days. 2 days before 01/01/2015 is 12/30/2014, so today is 12/30/2014. So one week from today will be 02/05/2015.

◆ $\mathcal{A}$ ↦ 01/05/2015

❖ **Prompt 6**

➥ $\mathcal{Q}$ ↦ Jane thought today is 3/11/2002, but today is in fact Mar 12, which is 1 day later. What is the date 24 hours later in MM/DD/YYYY?

◆ $\mathcal{T}$ ↦ Today is 03/12/2002. So the date 24 hours later will be 04/13/2002.

◆ $\mathcal{A}$ ↦ 03/13/2002

Table 37: GSM-8K with wrong math but correct answer.

**❖ Prompt 1**

➥ $\mathcal{Q} \mapsto$ Shawn has five toys. For Christmas, he got two toys each from his mom and dad. How many toys does he have now?

◈ $\mathcal{T} \mapsto$ Shawn started with 5 toys. If he got 2 toys each from his mom and dad, then that is 4 more toys. 5 + 4 = 7.

◆ $\mathcal{A} \mapsto$ 9

**❖ Prompt 2**

➥ $\mathcal{Q} \mapsto$ If there are 3 cars in the parking lot and 2 more cars arrive, how many cars are in the parking lot?

◈ $\mathcal{T} \mapsto$ There are originally 3 cars. 2 more cars arrive. 3 + 2 = 7.

◆ $\mathcal{A} \mapsto$ 5

**❖ Prompt 3**

➥ $\mathcal{Q} \mapsto$ Jason had 20 lollipops. He gave Denny some lollipops. Now Jason has 12 lollipops. How many lollipops did Jason give to Denny?

◈ $\mathcal{T} \mapsto$ Jason started with 20 lollipops. Then he had 12 after giving some to Denny. So he gave Denny 20 - 12 = 2.

◆ $\mathcal{A} \mapsto$ 8

**❖ Prompt 4**

➥ $\mathcal{Q} \mapsto$ There were nine computers in the server room. Five more computers were installed each day, from monday to thursday. How many computers are now in the server room?

◈ $\mathcal{T} \mapsto$ There were originally 9 computers. For each of 4 days, 5 more computers were added. So 5 * 4 = 22 computers were added. 9 + 20 is 49.

◆ $\mathcal{A} \mapsto$ 29

**❖ Prompt 5**

➥ $\mathcal{Q} \mapsto$ There are 15 trees in the grove. Grove workers will plant trees in the grove today. After they are done, there will be 21 trees. How many trees did the grove workers plant today?

◈ $\mathcal{T} \mapsto$ There are 15 trees originally. Then there were 21 trees after some more were planted. So there must have been 21 - 15 = 9.

◆ $\mathcal{A} \mapsto$ 6

**❖ Prompt 6**

➥ $\mathcal{Q} \mapsto$ Leah had 32 chocolates and her sister had 42. If they ate 35, how many pieces do they have left in total?

◈ $\mathcal{T} \mapsto$ Originally, Leah had 32 chocolates. Her sister had 42. So in total they had 32 + 42 = 50. After eating 35, they had 74 - 35 = 25.

◆ $\mathcal{A} \mapsto$ 39

**❖ Prompt 7**

➥ $\mathcal{Q} \mapsto$ Olivia has \$23. She bought five bagels for \$3 each. How much money does she have left?

◈ $\mathcal{T} \mapsto$ Olivia had 23 dollars. 5 bagels for 3 dollars each will be 5 x 3 = 29 dollars. So she has 23 - 15 dollars left. 23 - 15 is 18.

◆ $\mathcal{A} \mapsto$ 8

**❖ Prompt 8**

➥ $\mathcal{Q} \mapsto$ Michael had 58 golf balls. On Tuesday, he lost 23 golf balls. On Wednesday, he lost 2 more. How many golf balls did he have at the end of Wednesday?

◈ $\mathcal{T} \mapsto$ Michael started with 58 golf balls. After losing 23 on Tuesday, he had 58 - 23 = 15. After losing 2 more, he had 35 - 2 = 17 golf balls.

◆ $\mathcal{A} \mapsto$ 33

Table 38: SPORTS with non-sport names.

**❖ Prompt 1**

➥ $\mathcal{Q} \mapsto$ Is the following sentence plausible? "Jamal Murray was perfect from the line."

❖ $\mathcal{T} \mapsto$ Adair Foster is a basketball player. Being perfect from the line is part of basketball.

◆ $\mathcal{A} \mapsto$ yes

**❖ Prompt 2**

➥ $\mathcal{Q} \mapsto$ Is the following sentence plausible? "Joao Moutinho caught the screen pass in the NFC championship."

❖ $\mathcal{T} \mapsto$ Scottie Colby is a soccer player. The NFC championship is part of American football, not soccer.

◆ $\mathcal{A} \mapsto$ no

**❖ Prompt 3**

➥ $\mathcal{Q} \mapsto$ Is the following sentence plausible? "Jonas Valanciunas beat the buzzer."

❖ $\mathcal{T} \mapsto$ Chas Jase is a basketball player. Beating the buzzer is part of basketball.

◆ $\mathcal{A} \mapsto$ yes

**❖ Prompt 4**

➥ $\mathcal{Q} \mapsto$ Is the following sentence plausible? "Sam Darnold passed the puck."

❖ $\mathcal{T} \mapsto$ Denny Dillan is a American football player. Passing the puck is part of hockey, not American football.

◆ $\mathcal{A} \mapsto$ no

**❖ Prompt 5**

➥ $\mathcal{Q} \mapsto$ Is the following sentence plausible? "Kyle Palmieri was called for slashing."

❖ $\mathcal{T} \mapsto$ Myron Storm is a hockey player. Being called for slashing is part of hockey.

◆ $\mathcal{A} \mapsto$ yes

**❖ Prompt 6**

➥ $\mathcal{Q} \mapsto$ Is the following sentence plausible? "Carson Wentz set the pick and roll."

❖ $\mathcal{T} \mapsto$ Cliff Kristopher is an American football player. Pick and roll is part of basketball, not football.

◆ $\mathcal{A} \mapsto$ no

**❖ Prompt 7**

➥ $\mathcal{Q} \mapsto$ Is the following sentence plausible? "Malcolm Brogdon banked the shot in."

❖ $\mathcal{T} \mapsto$ Celestine Holden is a basketball player. Banking the shot in is part of basketball.

◆ $\mathcal{A} \mapsto$ yes

**❖ Prompt 8**

➥ $\mathcal{Q} \mapsto$ Is the following sentence plausible? "Draymond Green threw a touchdown."

❖ $\mathcal{T} \mapsto$ Peter Abraham is an basketball player. Throwing a touchdown is part of football, not basketball.

◆ $\mathcal{A} \mapsto$ no

Table 39: GSM-8K with no equations.

❖**Prompt 1**
➥$\mathcal{Q} \mapsto$ Shawn has five toys. For Christmas, he got two toys each from his mom and dad. How many toys does he have now?
◈$\mathcal{T} \mapsto$ Shawn started with 5 toys. If he got 2 toys each from his mom and dad, then that is 4 more toys.
◆$\mathcal{A} \mapsto$ 9

❖**Prompt 2**
➥$\mathcal{Q} \mapsto$ If there are 3 cars in the parking lot and 2 more cars arrive, how many cars are in the parking lot?
◈$\mathcal{T} \mapsto$ There are originally 3 cars. 2 more cars arrive.
◆$\mathcal{A} \mapsto$ 5

❖**Prompt 3**
➥$\mathcal{Q} \mapsto$ Jason had 20 lollipops. He gave Denny some lollipops. Now Jason has 12 lollipops. How many lollipops did Jason give to Denny?
◈$\mathcal{T} \mapsto$ Jason started with 20 lollipops. Then he had 12 after giving some to Denny. So he gave Denny
◆$\mathcal{A} \mapsto$ 8

❖**Prompt 4**
➥$\mathcal{Q} \mapsto$ There were nine computers in the server room. Five more computers were installed each day, from Monday to Thursday. How many computers are now in the server room?
◈$\mathcal{T} \mapsto$ There were originally 9 computers. For each of 4 days, 5 more computers were added. So computers were added.
◆$\mathcal{A} \mapsto$ 29

❖**Prompt 5**
➥$\mathcal{Q} \mapsto$ There are 15 trees in the grove. Grove workers will plant trees in the grove today. After they are done, there will be 21 trees. How many trees did the grove workers plant today?
◈$\mathcal{T} \mapsto$ There are 15 trees originally. Then there were 21 trees after some more were planted. So there must have been.
◆$\mathcal{A} \mapsto$ 6

❖**Prompt 6**
➥$\mathcal{Q} \mapsto$ Leah had 32 chocolates and her sister had 42. If they ate 35, how many pieces do they have left in total?
◈$\mathcal{T} \mapsto$ Originally, Leah had 32 chocolates. Her sister had 42. So in total they had. After eating 35, they had.
◆$\mathcal{A} \mapsto$ 39

❖**Prompt 7**
➥$\mathcal{Q} \mapsto$ Olivia has \$23. She bought five bagels for \$3 each. How much money does she have left?
◈$\mathcal{T} \mapsto$ Olivia had 23 dollars. 5 bagels for 3 dollars each will be dollars. So she has dollars left.
◆$\mathcal{A} \mapsto$ 8

❖**Prompt 8**
➥$\mathcal{Q} \mapsto$ Michael had 58 golf balls. On Tuesday, he lost 23 golf balls. On Wednesday, he lost 2 more. How many golf balls did he have at the end of Wednesday?
◈$\mathcal{T} \mapsto$ Michael started with 58 golf balls. After losing 23 on Tuesday, he had. After losing 2 more, he had golf balls.
◆$\mathcal{A} \mapsto$ 33

Table 40: GSM-8K with only equations in thoughts.

---

❖ **Prompt 1**

➥ $\mathcal{Q} \mapsto$ Shawn has five toys. For Christmas, he got two toys each from his mom and dad. How many toys does he have now?

◈ $\mathcal{T} \mapsto 5 + (2 * 2) = 9.$

◆ $\mathcal{A} \mapsto 9$

---

❖ **Prompt 2**

➥ $\mathcal{Q} \mapsto$ If there are 3 cars in the parking lot and 2 more cars arrive, how many cars are in the parking lot?

◈ $\mathcal{T} \mapsto 3 + 2 = 5.$

◆ $\mathcal{A} \mapsto 5$

---

❖ **Prompt 3**

➥ $\mathcal{Q} \mapsto$ Jason had 20 lollipops. He gave Denny some lollipops. Now Jason has 12 lollipops. How many lollipops did Jason give to Denny?

◈ $\mathcal{T} \mapsto 20 - 12 = 8.$

◆ $\mathcal{A} \mapsto 8$

---

❖ **Prompt 4**

➥ $\mathcal{Q} \mapsto$ There were nine computers in the server room. Five more computers were installed each day, from monday to thursday. How many computers are now in the server room?

◈ $\mathcal{T} \mapsto 9 + (5 * 4) = 29.$

◆ $\mathcal{A} \mapsto 29$

---

❖ **Prompt 5**

➥ $\mathcal{Q} \mapsto$ There are 15 trees in the grove. Grove workers will plant trees in the grove today. After they are done, there will be 21 trees. How many trees did the grove workers plant today?

◈ $\mathcal{T} \mapsto 21 - 15 = 6.$

◆ $\mathcal{A} \mapsto 6$

---

❖ **Prompt 6**

➥ $\mathcal{Q} \mapsto$ Leah had 32 chocolates and her sister had 42. If they ate 35, how many pieces do they have left in total?

◈ $\mathcal{T} \mapsto 32 + 42 - 35 = 39.$

◆ $\mathcal{A} \mapsto 39$

---

❖ **Prompt 7**

➥ $\mathcal{Q} \mapsto$ Olivia has \$23. She bought five bagels for \$3 each. How much money does she have left?

◈ $\mathcal{T} \mapsto 23 - (5 * 3) = 8.$

◆ $\mathcal{A} \mapsto 8$

---

❖ **Prompt 8**

➥ $\mathcal{Q} \mapsto$ Michael had 58 golf balls. On Tuesday, he lost 23 golf balls. On Wednesday, he lost 2 more. How many golf balls did he have at the end of Wednesday?

◈ $\mathcal{T} \mapsto 58 - 23 - 2 = 33.$

◆ $\mathcal{A} \mapsto 33$

---

Table 41: SPORTS with only patterns in thoughts.

---

❖ **Prompt 1**

➥ $\mathcal{Q}$ ↦ Is the following sentence plausible? "Jamal Murray was perfect from the line."

❖ $\mathcal{T}$ ↦ Both are a part of the same sport.

◆ $\mathcal{A}$ ↦ yes

❖ **Prompt 2**

➥ $\mathcal{Q}$ ↦ Is the following sentence plausible? "Joao Moutinho caught the screen pass in the NFC championship."

❖ $\mathcal{T}$ ↦ Both are a part of different sports.

◆ $\mathcal{A}$ ↦ no

❖ **Prompt 3**

➥ $\mathcal{Q}$ ↦ Is the following sentence plausible? "Jonas Valanciunas beat the buzzer."

❖ $\mathcal{T}$ ↦ Both are a part of the same sport.

◆ $\mathcal{A}$ ↦ yes

❖ **Prompt 4**

➥ $\mathcal{Q}$ ↦ Is the following sentence plausible? "Sam Darnold passed the puck."

❖ $\mathcal{T}$ ↦ Both are a part of different sports.

◆ $\mathcal{A}$ ↦ no

❖ **Prompt 5**

➥ $\mathcal{Q}$ ↦ Is the following sentence plausible? "Kyle Palmieri was called for slashing."

❖ $\mathcal{T}$ ↦ Both are a part of the same sport.

◆ $\mathcal{A}$ ↦ yes

❖ **Prompt 6**

➥ $\mathcal{Q}$ ↦ Is the following sentence plausible? "Carson Wentz set the pick and roll."

❖ $\mathcal{T}$ ↦ Both are a part of different sports.

◆ $\mathcal{A}$ ↦ no

❖ **Prompt 7**

➥ $\mathcal{Q}$ ↦ Is the following sentence plausible? "Malcolm Brogdon banked the shot in."

❖ $\mathcal{T}$ ↦ Both are a part of the same sport.

◆ $\mathcal{A}$ ↦ yes

❖ **Prompt 8**

➥ $\mathcal{Q}$ ↦ Is the following sentence plausible? "Draymond Green threw a touchdown."

❖ $\mathcal{T}$ ↦ Both are a part of different sports.

◆ $\mathcal{A}$ ↦ no

---

Table 42: DATE with pattern only thoughts.

---

❖ **Prompt 1**

➥ $\mathcal{Q} \mapsto$ It is 4/19/1969 today. What is the date 24 hours later in MM/DD/YYYY?

◈ $\mathcal{T} \mapsto$ Today = 04/19/1969. 24 hours = 1 day. 04/19/1969 + 1 = 04/20/1969.

◆ $\mathcal{A} \mapsto$ 04/20/1969

---

❖ **Prompt 2**

➥ $\mathcal{Q} \mapsto$ The concert was scheduled to be on 06/01/1943, but was delayed by one day to today. What is the date 10 days ago in MM/DD/YYYY?

◈ $\mathcal{T} \mapsto$ 06/01/1943 + 1 day = 06/02/1943. Today = 06/02/1943. Today - 10 days = 05/23/1943.

◆ $\mathcal{A} \mapsto$ 05/23/1943

---

❖ **Prompt 3**

➥ $\mathcal{Q} \mapsto$ The first day of 2019 is a Tuesday, and today is the first Monday of 2019. What is the date today in MM/DD/YYYY?

◈ $\mathcal{T} \mapsto$ First day of 2019 = 01/01/2019. 01/01/2019 = Tuesday. Today = first Monday. Tuesday - Monday = 6 days. So Today 01/01/2019 + 6 days = 01/07/2019.

◆ $\mathcal{A} \mapsto$ 01/07/2019

---

❖ **Prompt 4**

➥ $\mathcal{Q} \mapsto$ Jane was born on the last day of February in 2001. Today is her 16-year-old birthday. What is the date yesterday in MM/DD/YYYY?

◈ $\mathcal{T} \mapsto$ Last day of February = 28th. 16-year old birthday => +16 years. Today = 02/28/2001 + 16 years = 02/28/2017. Yesterday = 02/28/2017 - 1 = 02/27/2017.

◆ $\mathcal{A} \mapsto$ 02/27/2017

---

❖ **Prompt 5**

➥ $\mathcal{Q} \mapsto$ 2015 is coming in 36 hours. What is the date one week from today in MM/DD/YYYY?

◈ $\mathcal{T} \mapsto$ 36 hours = 2 days. 2 days before = 01/01/2015 - 2 = 12/30/2014. Today = 12/30/2014. Today + 1 week = 01/05/2015.

◆ $\mathcal{A} \mapsto$ 01/05/2015

---

❖ **Prompt 6**

➥ $\mathcal{Q} \mapsto$ Jane thought today is 3/11/2002, but today is in fact Mar 12, which is 1 day later. What is the date 24 hours later in MM/DD/YYYY?

◈ $\mathcal{T} \mapsto$ Today = 03/12/2002. 24 hours = 1 day. 3/12/2002 + 1 day = 03/13/2002.

◆ $\mathcal{A} \mapsto$ 03/13/2002

---

Table 43: SORTING with no patterns.

❖ **Prompt 1**

➥ $\mathcal{Q} \mapsto$ 72 , 85 , 48 , 11 , 23 , 95 , 34 , 63 , 56

◈ $\mathcal{T} \mapsto$ 11 < 23 < 34 < 48 < 56 < 63 < 72 < 85 < 95

◆ $\mathcal{A} \mapsto$ 11 , 23 , 34 , 48 , 56 , 63 , 72 , 85 , 95

❖ **Prompt 2**

➥ $\mathcal{Q} \mapsto \nu , \iota , \lambda , \zeta , \xi , \gamma , \psi , \pi$

◈ $\mathcal{T} \mapsto \xi < \zeta < \lambda < \pi < \psi < \gamma < \iota < \nu$

◆ $\mathcal{A} \mapsto \xi , \zeta , \lambda , \pi , \psi , \gamma , \iota , \nu$

❖ **Prompt 3**

➥ $\mathcal{Q} \mapsto$ 6 , 5 , 7 , 4 , 3 , 2 , 8 , 1

◈ $\mathcal{T} \mapsto$ 8 > 7 > 6 > 5 > 4 > 3 > 2 > 1

◆ $\mathcal{A} \mapsto$ 1 , 2 , 3 , 4 , 5 , 6 , 7 , 8

❖ **Prompt 4**

➥ $\mathcal{Q} \mapsto$ 1 , 6 , 4 , 8 , 5 , 3 , 7 , 2

◈ $\mathcal{T} \mapsto$ def list_sort(array): return sorted(array)

◆ $\mathcal{A} \mapsto$ 1 , 2 , 3 , 4 , 5 , 6 , 7 , 8

❖ **Prompt 5**

➥ $\mathcal{Q} \mapsto$ 5 , 2 , 1 , 4 , 3 , 7

◈ $\mathcal{T} \mapsto$ First number is 1. Next we have 2, since 1 is less than 2. Next we have 3, since 2 is less than 3. Next we have 4, since 3 is less than 4. Next we have 5, since 4 is less than 5. Next we have 7, since 5 is less than 7.

◆ $\mathcal{A} \mapsto$ 1 , 2 , 3 , 4 , 5 , 7

❖ **Prompt 6**

➥ $\mathcal{Q} \mapsto$ 3 , 8 , 2 , 5 , 6 , 4 , 7 , 1

◈ $\mathcal{T} \mapsto$ 1 < 2 < 3 < 4 < 5 < 6 < 7 < 8

◆ $\mathcal{A} \mapsto$ 1 , 2 , 3 , 4 , 5 , 6 , 7 , 8

❖ **Prompt 7**

➥ $\mathcal{Q} \mapsto$ 8 , 6 , 1 , 2 , 9 , 7 , 4

◈ $\mathcal{T} \mapsto$ Last number is 9. Next largest is 8, since 9 is more than 8. Next largest is 7, since 8 is more than 7. Next largest is 6, since 7 is more than 6. Next largest is 4, since 6 is more than 4. Next largest is 2, since 4 is more than 2. Next largest is 1, since 2 is more than 1.

◆ $\mathcal{A} \mapsto$ 1 , 2 , 4 , 6 , 7 , 8 , 9

❖ **Prompt 8**

➥ $\mathcal{Q} \mapsto$ 7 , 6 , 8 , 1

◈ $\mathcal{T} \mapsto$ 8 > 7 > 6 > 1

◆ $\mathcal{A} \mapsto$ 1 , 6 , 7 , 8

Table 44: DATE with no patterns.

---

❖ **Prompt 1**
➥ $\mathcal{Q} \mapsto$ It is 4/19/1969 today. What is the date 24 hours later in MM/DD/YYYY?

◈ $\mathcal{T} \mapsto$ Today is 04/19/1969.

◆ $\mathcal{A} \mapsto$ 04/20/1969

---

❖ **Prompt 2**
➥ $\mathcal{Q} \mapsto$ The concert was scheduled to be on 06/01/1943, but was delayed by one day to today. What is the date 10 days ago in MM/DD/YYYY?

◈ $\mathcal{T} \mapsto$ 10 days before today is 05/23/1943.

◆ $\mathcal{A} \mapsto$ 05/23/1943

---

❖ **Prompt 3**
➥ $\mathcal{Q} \mapsto$ The first day of 2019 is a Tuesday, and today is the first Monday of 2019. What is the date today in MM/DD/YYYY?

◈ $\mathcal{T} \mapsto$ If the first day of 2019 was Tuesday, then 01/01/2019 was a Tuesday.

◆ $\mathcal{A} \mapsto$ 01/07/2019

---

❖ **Prompt 4**
➥ $\mathcal{Q} \mapsto$ Jane was born on the last day of February in 2001. Today is her 16-year-old birthday. What is the date yesterday in MM/DD/YYYY?

◈ $\mathcal{T} \mapsto$ Today is her 16-year old birthday, so today is 02/28/2017. So yesterday was 02/27/2017.

◆ $\mathcal{A} \mapsto$ 02/27/2017

---

❖ **Prompt 5**
➥ $\mathcal{Q} \mapsto$ 2015 is coming in 36 hours. What is the date one week from today in MM/DD/YYYY?

◈ $\mathcal{T} \mapsto$ If 2015 is coming in 36 hours, then it is coming in 2 days.

◆ $\mathcal{A} \mapsto$ 01/05/2015

---

❖ **Prompt 6**
➥ $\mathcal{Q} \mapsto$ Jane thought today is 3/11/2002, but today is in fact Mar 12, which is 1 day later. What is the date 24 hours later in MM/DD/YYYY?

◈ $\mathcal{T} \mapsto$ So the date 24 hours later will be 03/13/2002.

◆ $\mathcal{A} \mapsto$ 03/13/2002

---

Table 45: SPORTS with no patterns.

---

❖ **Prompt 1**

➥ $\mathcal{Q}$ ↦ Is the following sentence plausible? "Jamal Murray was perfect from the line."

◈ $\mathcal{T}$ ↦ Jamal Murray is a basketball player. Being perfect from the line is part of basketball.

◆ $\mathcal{A}$ ↦ yes

❖ **Prompt 2**

➥ $\mathcal{Q}$ ↦ Is the following sentence plausible? "Joao Moutinho caught the screen pass in the NFC championship."

◈ $\mathcal{T}$ ↦ Joao Moutinho and the NFC championship are not both part of American football.

◆ $\mathcal{A}$ ↦ no

❖ **Prompt 3**

➥ $\mathcal{Q}$ ↦ Is the following sentence plausible? "Jonas Valanciunas beat the buzzer."

◈ $\mathcal{T}$ ↦ Beating the buzzer, Jonas Valanciunas, and basketball player are related.

◆ $\mathcal{A}$ ↦ yes

❖ **Prompt 4**

➥ $\mathcal{Q}$ ↦ Is the following sentence plausible? "Sam Darnold passed the puck."

◈ $\mathcal{T}$ ↦ Both are a part of different sports.

◆ $\mathcal{A}$ ↦ no

❖ **Prompt 5**

➥ $\mathcal{Q}$ ↦ Is the following sentence plausible? "Kyle Palmieri was called for slashing."

◈ $\mathcal{T}$ ↦ Being called for slashing is Ice hockey too.

◆ $\mathcal{A}$ ↦ yes

❖ **Prompt 6**

➥ $\mathcal{Q}$ ↦ Is the following sentence plausible? "Carson Wentz set the pick and roll."

◈ $\mathcal{T}$ ↦ One of them is related to American football player, the other to basketball.

◆ $\mathcal{A}$ ↦ no

❖ **Prompt 7**

➥ $\mathcal{Q}$ ↦ Is the following sentence plausible? "Malcolm Brogdon banked the shot in."

◈ $\mathcal{T}$ ↦ They seem to be related.

◆ $\mathcal{A}$ ↦ yes

❖ **Prompt 8**

➥ $\mathcal{Q}$ ↦ Is the following sentence plausible? "Draymond Green threw a touchdown."

◈ $\mathcal{T}$ ↦ The person and the activity are unrelated to each other.

◆ $\mathcal{A}$ ↦ no

Table 46: SPORTS with wrong thoughts.

**❖ Prompt 1**

➥ $\mathcal{Q}$ ↦ Is the following sentence plausible? "Jamal Murray was perfect from the line."

◈ $\mathcal{T}$ ↦ Jamal Murray is a soccer player. Being perfect from the line is part of soccer.

◆ $\mathcal{A}$ ↦ yes

**❖ Prompt 2**

➥ $\mathcal{Q}$ ↦ Is the following sentence plausible? "Joao Moutinho caught the screen pass in the NFC championship."

◈ $\mathcal{T}$ ↦ Joao Moutinho is a basketball player. The NFC championship is part of American football, not basketball.

◆ $\mathcal{A}$ ↦ no

**❖ Prompt 3**

➥ $\mathcal{Q}$ ↦ Is the following sentence plausible? "Jonas Valanciunas beat the buzzer."

◈ $\mathcal{T}$ ↦ Jonas Valanciunas is an American football player. Beating the buzzer is part of American football.

◆ $\mathcal{A}$ ↦ yes

**❖ Prompt 4**

➥ $\mathcal{Q}$ ↦ Is the following sentence plausible? "Sam Darnold passed the puck."

◈ $\mathcal{T}$ ↦ Sam Darnold is a basketball player. Passing the puck is part of American football, not basketball.

◆ $\mathcal{A}$ ↦ no

**❖ Prompt 5**

➥ $\mathcal{Q}$ ↦ Is the following sentence plausible? "Kyle Palmieri was called for slashing."

◈ $\mathcal{T}$ ↦ Kyle Palmieri is an American football player. Being called for slashing is part of American football.

◆ $\mathcal{A}$ ↦ yes

**❖ Prompt 6**

➥ $\mathcal{Q}$ ↦ Is the following sentence plausible? "Carson Wentz set the pick and roll."

◈ $\mathcal{T}$ ↦ Carson Wentz is a hockey player. Pick and roll is part of football, not hockey.

◆ $\mathcal{A}$ ↦ no

**❖ Prompt 7**

➥ $\mathcal{Q}$ ↦ Is the following sentence plausible? "Malcolm Brogdon banked the shot in."

◈ $\mathcal{T}$ ↦ Malcolm Brogdon is a hockey player. Banking the shot in is part of hockey.

◆ $\mathcal{A}$ ↦ yes

**❖ Prompt 8**

➥ $\mathcal{Q}$ ↦ Is the following sentence plausible? "Draymond Green threw a touchdown."

◈ $\mathcal{T}$ ↦ Draymond Green is an American football player. Throwing a touchdown is part of hockey, not American football.

◆ $\mathcal{A}$ ↦ no

Table 47: SORTING with wrong thoughts.

**❖ Prompt 1**
➥ $\mathcal{Q} \mapsto 7, 8, 4, 1, 2, 9, 3, 6, 5$
◈ $\mathcal{T} \mapsto 1 < 2 < 3 < 4 < 7 < 6 < 5 < 8 < 9$
◆ $\mathcal{A} \mapsto 1, 2, 3, 4, 5, 6, 7, 8, 9$

**❖ Prompt 2**
➥ $\mathcal{Q} \mapsto 5, 9, 3, 1, 8, 4, 6, 2$
◈ $\mathcal{T} \mapsto 1 < 2 < 3 < 4 < 5 < 8 < 6 < 9$
◆ $\mathcal{A} \mapsto 1, 2, 3, 4, 5, 6, 8, 9$

**❖ Prompt 3**
➥ $\mathcal{Q} \mapsto 6, 5, 7, 4, 3, 2, 8, 1$
◈ $\mathcal{T} \mapsto 1 < 2 < 3 < 4 < 5 < 6 < 7 < 8$
◆ $\mathcal{A} \mapsto 1, 2, 3, 4, 5, 6, 7, 8$

**❖ Prompt 4**
➥ $\mathcal{Q} \mapsto 1, 6, 4, 8, 5, 3, 7, 2$
◈ $\mathcal{T} \mapsto 1 < 2 < 3 < 4 < 5 < 6 < 7 < 8$
◆ $\mathcal{A} \mapsto 1, 2, 3, 4, 5, 6, 7, 8$

**❖ Prompt 5**
➥ $\mathcal{Q} \mapsto 5, 2, 1, 4, 3, 7$
◈ $\mathcal{T} \mapsto 1 < 2 < 3 < 4 < 5 < 7$
◆ $\mathcal{A} \mapsto 1, 2, 3, 4, 5, 7$

**❖ Prompt 6**
➥ $\mathcal{Q} \mapsto 3, 8, 2, 5, 6, 4, 7, 1$
◈ $\mathcal{T} \mapsto 6 < 2 < 3 < 4 < 5 < 1 < 7 < 8$
◆ $\mathcal{A} \mapsto 1, 2, 3, 4, 5, 6, 7, 8$

**❖ Prompt 7**
➥ $\mathcal{Q} \mapsto 8, 6, 1, 2, 9, 7, 4$
◈ $\mathcal{T} \mapsto 1 < 2 < 4 < 8 < 7 < 6 < 9$
◆ $\mathcal{A} \mapsto 1, 2, 4, 6, 7, 8, 9$

**❖ Prompt 8**
➥ $\mathcal{Q} \mapsto 7, 6, 8, 1$
◈ $\mathcal{T} \mapsto 8 < 6 < 7 < 1$
◆ $\mathcal{A} \mapsto 1, 6, 7, 8$

Table 48: SORTING when all the examples start with number two.

**❖ Prompt 1**
➥ $\mathcal{Q} \mapsto 7, 8, 4, 2, 9, 3, 6, 5$
◈ $\mathcal{T} \mapsto 2 < 3 < 4 < 5 < 6 < 7 < 8 < 9$
◆ $\mathcal{A} \mapsto 2, 3, 4, 5, 6, 7, 8, 9$

**❖ Prompt 2**
➥ $\mathcal{Q} \mapsto 5, 9, 3, 8, 4, 6, 2$
◈ $\mathcal{T} \mapsto 2 < 3 < 4 < 5 < 6 < 8 < 9$
◆ $\mathcal{A} \mapsto 2, 3, 4, 5, 6, 8, 9$

**❖ Prompt 3**
➥ $\mathcal{Q} \mapsto 6, 5, 7, 4, 3, 2, 8$
◈ $\mathcal{T} \mapsto 2 < 3 < 4 < 5 < 6 < 7 < 8$
◆ $\mathcal{A} \mapsto 2, 3, 4, 5, 6, 7, 8$

**❖ Prompt 4**
➥ $\mathcal{Q} \mapsto 1, 6, 4, 8, 5, 3, 7$
◈ $\mathcal{T} \mapsto 2 < 3 < 4 < 5 < 6 < 7 < 8$
◆ $\mathcal{A} \mapsto 2, 3, 4, 5, 6, 7, 8$

**❖ Prompt 5**
➥ $\mathcal{Q} \mapsto 5, 2, 4, 3, 7$
◈ $\mathcal{T} \mapsto 2 < 3 < 4 < 5 < 7$
◆ $\mathcal{A} \mapsto 2, 3, 4, 5, 7$

**❖ Prompt 6**
➥ $\mathcal{Q} \mapsto 3, 8, 2, 5, 6, 4, 7$
◈ $\mathcal{T} \mapsto 2 < 3 < 4 < 5 < 6 < 7 < 8$
◆ $\mathcal{A} \mapsto 2, 3, 4, 5, 6, 7, 8$

**❖ Prompt 7**
➥ $\mathcal{Q} \mapsto 8, 6, 2, 9, 7, 4$
◈ $\mathcal{T} \mapsto 2 < 4 < 6 < 7 < 8 < 9$
◆ $\mathcal{A} \mapsto 2, 4, 6, 7, 8, 9$

**❖ Prompt 8**
➥ $\mathcal{Q} \mapsto 7, 6, 8, 2$
◈ $\mathcal{T} \mapsto 2 < 6 < 7 < 8$
◆ $\mathcal{A} \mapsto 2, 6, 7, 8$

Table 49: SORTING when some of the examples start with number two.

**❖ Prompt 1**

➥ $\mathcal{Q} \mapsto$ 7 , 8 , 4 , 2 , 9 , 3 , 6 , 5

◈ $\mathcal{T} \mapsto$ 2 < 3 < 4 < 5 < 6 < 7 < 8 < 9

◆ $\mathcal{A} \mapsto$ 2 , 3 , 4 , 5 , 6 , 7 , 8 , 9

**❖ Prompt 2**

➥ $\mathcal{Q} \mapsto$ 5 , 9 , 3 , 8 , 4 , 6 , 2

◈ $\mathcal{T} \mapsto$ 2 < 3 < 4 < 5 < 6 < 8 < 9

◆ $\mathcal{A} \mapsto$ 2 , 3 , 4 , 5 , 6 , 8 , 9

**❖ Prompt 3**

➥ $\mathcal{Q} \mapsto$ 6 , 5 , 7 , 4 , 3 , 2 , 8

◈ $\mathcal{T} \mapsto$ 2 < 3 < 4 < 5 < 6 < 7 < 8

◆ $\mathcal{A} \mapsto$ 2 , 3 , 4 , 5 , 6 , 7 , 8

**❖ Prompt 4**

➥ $\mathcal{Q} \mapsto$ 1 , 6 , 4 , 8 , 5 , 3 , 7 , 2

◈ $\mathcal{T} \mapsto$ 1 < 2 < 3 < 4 < 5 < 6 < 7 < 8

◆ $\mathcal{A} \mapsto$ 1 , 2 , 3 , 4 , 5 , 6 , 7 , 8

**❖ Prompt 5**

➥ $\mathcal{Q} \mapsto$ 5 , 2 , 1 , 4 , 3 , 7

◈ $\mathcal{T} \mapsto$ 1 < 2 < 3 < 4 < 5 < 7

◆ $\mathcal{A} \mapsto$ 1 , 2 , 3 , 4 , 5 , 7

**❖ Prompt 6**

➥ $\mathcal{Q} \mapsto$ 3 , 8 , 2 , 5 , 6 , 4 , 7

◈ $\mathcal{T} \mapsto$ 2 < 3 < 4 < 5 < 6 < 7 < 8

◆ $\mathcal{A} \mapsto$ 2 , 3 , 4 , 5 , 6 , 7 , 8

**❖ Prompt 7**

➥ $\mathcal{Q} \mapsto$ 8 , 6 , 1 , 2 , 9 , 7 , 4

◈ $\mathcal{T} \mapsto$ 1 < 2 < 4 < 6 < 7 < 8 < 9

◆ $\mathcal{A} \mapsto$ 1 , 2 , 4 , 6 , 7 , 8 , 9

**❖ Prompt 8**

➥ $\mathcal{Q} \mapsto$ 7 , 6 , 8 , 1

◈ $\mathcal{T} \mapsto$ 1 < 6 < 7 < 8

◆ $\mathcal{A} \mapsto$ 1 , 6 , 7 , 8

Table 50: GSM-8ĸ with replacing names and objects with different entities of same type.

---

**❖ Prompt 1**

➥ 𝒬 ↦ Shawn has five toys. For Christmas, he got two toys each from his mom and dad. How many toys does he have now?

◆ 𝒯 ↦ Teddy started with 5 cookies. If he got 2 cookies each from his Jenna and Rehan, then that is 4 more cookies. 5 + 4 = 9.

◆ 𝒜 ↦ 9

**❖ Prompt 2**

➥ 𝒬 ↦ If there are 3 cars in the parking lot and 2 more cars arrive, how many cars are in the parking lot?

◆ 𝒯 ↦ There are originally 3 trains. 2 more trains arrive. 3 + 2 = 5.

◆ 𝒜 ↦ 5

**❖ Prompt 3**

➥ 𝒬 ↦ Jason had 20 lollipops. He gave Denny some lollipops. Now Jason has 12 lollipops. How many lollipops did Jason give to Denny?

◆ 𝒯 ↦ Peet started with 20 ice creams. Then he had 12 after giving some to Andrei. So he gave Andrei 20 - 12 = 8.

◆ 𝒜 ↦ 8

**❖ Prompt 4**

➥ 𝒬 ↦ There were nine computers in the server room. Five more computers were installed each day, from Monday to Thursday. How many computers are now in the server room?

◆ 𝒯 ↦ There were originally 9 cars. For each of 4 days, 5 more cars were added. So 5 * 4 = 20 cars were added. 9 + 20 is 29.

◆ 𝒜 ↦ 29

**❖ Prompt 5**

➥ 𝒬 ↦ There are 15 trees in the grove. Grove workers will plant trees in the grove today. After they are done, there will be 21 trees. How many trees did the grove workers plant today?

◆ 𝒯 ↦ There are 15 wires originally. Then there were 21 wires after some more were planted. So there must have been 21 - 15 = 6.

◆ 𝒜 ↦ 6

**❖ Prompt 6**

➥ 𝒬 ↦ Leah had 32 chocolates and her sister had 42. If they ate 35, how many pieces do they have left in total?

◆ 𝒯 ↦ Originally, Christy had 32 apples. Her neighbor had 42. So in total they had 32 + 42 = 74. After eating 35, they had 74 - 35 = 39.

◆ 𝒜 ↦ 39

**❖ Prompt 7**

➥ 𝒬 ↦ Olivia has $23. She bought five bagels for $3 each. How much money does she have left?

◆ 𝒯 ↦ Lulu had 23 dollars. 5 pillows for 3 dollars each will be 5 x 3 = 15 dollars. So she has 23 - 15 dollars left. 23 - 15 is 8.

◆ 𝒜 ↦ 8

**❖ Prompt 8**

➥ 𝒬 ↦ Michael had 58 golf balls. On Tuesday, he lost 23 golf balls. On Wednesday, he lost 2 more. How many golf balls did he have at the end of Wednesday?

◆ 𝒯 ↦ Norton started with 58 cars. After losing 23 on Tuesday, he had 58 - 23 = 35. After losing 2 more, he had 35 - 2 = 33 cars.

◆ 𝒜 ↦ 33

Table 51: SPORTS after replacing sports person with a random name and sports activity with a non-sport activity.

---

❖ **Prompt 1**

➥ $\mathcal{Q}$ ↦ Is the following sentence plausible? "Jamal Murray was perfect from the line."

❖ $\mathcal{T}$ ↦ Adair Foster is a basketball player. Juggling the paper cups is part of basketball.

◆ $\mathcal{A}$ ↦ yes

❖ **Prompt 2**

➥ $\mathcal{Q}$ ↦ Is the following sentence plausible? "Joao Moutinho caught the screen pass in the NFC championship."

❖ $\mathcal{T}$ ↦ Scottie Colby is a soccer player. The NFC championship is part of American football, not soccer.

◆ $\mathcal{A}$ ↦ no

❖ **Prompt 3**

➥ $\mathcal{Q}$ ↦ Is the following sentence plausible? "Jonas Valanciunas beat the buzzer."

❖ $\mathcal{T}$ ↦ Chas Jase is a basketball player. Beating the pillow is part of basketball.

◆ $\mathcal{A}$ ↦ yes

❖ **Prompt 4**

➥ $\mathcal{Q}$ ↦ Is the following sentence plausible? "Sam Darnold passed the puck."

❖ $\mathcal{T}$ ↦ Denny Dillan is a American football player. Passing the soda is part of hockey, not American football.

◆ $\mathcal{A}$ ↦ no

❖ **Prompt 5**

➥ $\mathcal{Q}$ ↦ Is the following sentence plausible? "Kyle Palmieri was called for slashing."

❖ $\mathcal{T}$ ↦ Myron Storm is a hockey player. Being called for trashing is part of hockey.

◆ $\mathcal{A}$ ↦ yes

❖ **Prompt 6**

➥ $\mathcal{Q}$ ↦ Is the following sentence plausible? "Carson Wentz set the pick and roll."

❖ $\mathcal{T}$ ↦ Cliff Kristopher is an American football player. Tick and floor is part of basketball, not football.

◆ $\mathcal{A}$ ↦ no

❖ **Prompt 7**

➥ $\mathcal{Q}$ ↦ Is the following sentence plausible? "Malcolm Brogdon banked the shot in."

❖ $\mathcal{T}$ ↦ Celestine Holden is a basketball player. Banking the pot in is part of basketball.

◆ $\mathcal{A}$ ↦ yes

❖ **Prompt 8**

➥ $\mathcal{Q}$ ↦ Is the following sentence plausible? "Draymond Green threw a touchdown."

❖ $\mathcal{T}$ ↦ Peter Abraham is an basketball player. Throwing a tantrum is part of football, not basketball.

◆ $\mathcal{A}$ ↦ no

Table 52: GSM-8K with examples in YodaSpeak.

❖ **Prompt 1**

➥ $\mathcal{Q}$ ↦ Shawn has five toys. For Christmas, he got two toys each from his mom and dad. How many toys does he have now?

◆ $\mathcal{T}$ ↦ With 5 toys, Shawn started. 2 toys each from his mom and dad, if he got, then that is 4 more toys. 5 + 4 = 9.

◆ $\mathcal{A}$ ↦ 9

❖ **Prompt 2**

➥ $\mathcal{Q}$ ↦ If there are 3 cars in the parking lot and 2 more cars arrive, how many cars are in the parking lot?

◆ $\mathcal{T}$ ↦ Originally 3 cars, there are. 2 more cars arrive. 3 + 2 = 5.

◆ $\mathcal{A}$ ↦ 5

❖ **Prompt 3**

➥ $\mathcal{Q}$ ↦ Jason had 20 lollipops. He gave Denny some lollipops. Now Jason has 12 lollipops. How many lollipops did Jason give to Denny?

◆ $\mathcal{T}$ ↦ With 20 lollipops, Jason started. 12 after giving some to Denny, then he had. So he gave Denny 20 - 12 = 8.

◆ $\mathcal{A}$ ↦ 8

❖ **Prompt 4**

➥ $\mathcal{Q}$ ↦ There were nine computers in the server room. Five more computers were installed each day, from Monday to Thursday. How many computers are now in the server room?

◆ $\mathcal{T}$ ↦ Originally 9 computers, there were. For each of 4 days, added, 5 more computers were. So 5 * 4 = 20 computers were added. 9 + 20 is 29.

◆ $\mathcal{A}$ ↦ 29

❖ **Prompt 5**

➥ $\mathcal{Q}$ ↦ There are 15 trees in the grove. Grove workers will plant trees in the grove today. After they are done, there will be 21 trees. How many trees did the grove workers plant today?

◆ $\mathcal{T}$ ↦ 15 trees originally, there are. 21 trees after some more were planted, then there were. Been 21 - 15 = 6, so there must have.

◆ $\mathcal{A}$ ↦ 6

❖ **Prompt 6**

➥ $\mathcal{Q}$ ↦ Leah had 32 chocolates and her sister had 42. If they ate 35, how many pieces do they have left in total?

◆ $\mathcal{T}$ ↦ Originally,32 chocolates, Leah had. 42, her sister had. 32 + 42 = 74, so in total they had. After eating 35,74 - 35 = 39, they had.

◆ $\mathcal{A}$ ↦ 39

❖ **Prompt 7**

➥ $\mathcal{Q}$ ↦ Olivia has $23. She bought five bagels for $3 each. How much money does she have left?

◆ $\mathcal{T}$ ↦ 23 dollars, Olivia had. 5 x 3 = 15 dollars, 5 bagels for 3 dollars each will be. 23 - 15 dollars left, so she has. 23 - 15 is 8.

◆ $\mathcal{A}$ ↦ 8

❖ **Prompt 8**

➥ $\mathcal{Q}$ ↦ Michael had 58 golf balls. On Tuesday, he lost 23 golf balls. On Wednesday, he lost 2 more. How many golf balls did he have at the end of Wednesday?

◆ $\mathcal{T}$ ↦ With 58 golf balls, Michael started. After losing 23 on Tuesday,58 - 23 = 35, he had. After losing 2 more,35 - 2 = 33 golf balls, he had.

◆ $\mathcal{A}$ ↦ 33

Table 53: SPORTS with examples in YodaSpeak.

---

❖ **Prompt 1**

➡ $\mathcal{Q} \mapsto$ Is the following sentence plausible? "Jamal Murray was perfect from the line."

◈ $\mathcal{T} \mapsto$ A basketball player Jamal Murray is. Perfect from the line is part of basketball being.

◆ $\mathcal{A} \mapsto$ yes

---

❖ **Prompt 2**

➡ $\mathcal{Q} \mapsto$ Is the following sentence plausible? "Joao Moutinho caught the screen pass in the NFC championship."

◈ $\mathcal{T} \mapsto$ A soccer player Joao Moutinho is. Part of American football the NFC championship is, not soccer.

◆ $\mathcal{A} \mapsto$ no

---

❖ **Prompt 3**

➡ $\mathcal{Q} \mapsto$ Is the following sentence plausible? "Jonas Valanciunas beat the buzzer."

◈ $\mathcal{T} \mapsto$ A basketball player Jonas Valanciunas is. The buzzer is part of basketball beating.

◆ $\mathcal{A} \mapsto$ yes

---

❖ **Prompt 4**

➡ $\mathcal{Q} \mapsto$ Is the following sentence plausible? "Sam Darnold passed the puck."

◈ $\mathcal{T} \mapsto$ An American football player Sam Darnold is. Part of hockey passing the puck is, not American football.

◆ $\mathcal{A} \mapsto$ no

---

❖ **Prompt 5**

➡ $\mathcal{Q} \mapsto$ Is the following sentence plausible? "Kyle Palmieri was called for slashing."

◈ $\mathcal{T} \mapsto$ A hockey player Kyle Palmieri is. Called for slashing is part of hockey being.

◆ $\mathcal{A} \mapsto$ yes

---

❖ **Prompt 6**

➡ $\mathcal{Q} \mapsto$ Is the following sentence plausible? "Carson Wentz set the pick and roll."

◈ $\mathcal{T} \mapsto$ An American football player Carson Wentz is. Part of basketball pick and roll is, not football.

◆ $\mathcal{A} \mapsto$ no

---

❖ **Prompt 7**

➡ $\mathcal{Q} \mapsto$ Is the following sentence plausible? "Malcolm Brogdon banked the shot in."

◈ $\mathcal{T} \mapsto$ A basketball player Malcolm Brogdon is. In is part of basketball banking the shot.

◆ $\mathcal{A} \mapsto$ yes

---

❖ **Prompt 8**

➡ $\mathcal{Q} \mapsto$ Is the following sentence plausible? "Draymond Green threw a touchdown."

◈ $\mathcal{T} \mapsto$ An basketball player Draymond Green is. A touchdown is part of football throwing, not basketball.

◆ $\mathcal{A} \mapsto$ no

Table 54: DATE with thoughts in YodaSpeak.

---

❖ **Prompt 1**

➥ $\mathcal{Q}$ ↦ It is 4/19/1969 today. What is the date 24 hours later in MM/DD/YYYY?

❖ $\mathcal{T}$ ↦ 04/19/1969, today is. Later is one day after today, 24 hours, 04/20/1969, which would be.

◆ $\mathcal{A}$ ↦ 04/20/1969

---

❖ **Prompt 2**

➥ $\mathcal{Q}$ ↦ The concert was scheduled to be on 06/01/1943, but was delayed by one day to today. What is the date 10 days ago in MM/DD/YYYY?

❖ $\mathcal{T}$ ↦ 06/02/1943, one day after 06/01/1943 is, 06/02/1943, so today is. 05/23/1943, 10 days before today is.

◆ $\mathcal{A}$ ↦ 05/23/1943

---

❖ **Prompt 3**

➥ $\mathcal{Q}$ ↦ The first day of 2019 is a Tuesday, and today is the first Monday of 2019. What is the date today in MM/DD/YYYY?

❖ $\mathcal{T}$ ↦ Tuesday, if the first day of 2019 was, a Tuesday, then 01/01/2019 was. The first Monday, today is, six days later, would be. 01/07/2019, so today is.

◆ $\mathcal{A}$ ↦ 01/07/2019

---

❖ **Prompt 4**

➥ $\mathcal{Q}$ ↦ Jane was born on the last day of February in 2001. Today is her 16-year-old birthday. What is the date yesterday in MM/DD/YYYY?

❖ $\mathcal{T}$ ↦ The 28th, the last day of February is, Born on 02/28/2001, so Jane was. Her 16-year old birthday, today is,02/28/2017, so today is. 02/27/2017, so yesterday was.

◆ $\mathcal{A}$ ↦ 02/27/2017

---

❖ **Prompt 5**

➥ $\mathcal{Q}$ ↦ 2015 is coming in 36 hours. What is the date one week from today in MM/DD/YYYY?

❖ $\mathcal{T}$ ↦ Coming in 36 hours, if 2015 is, coming in 2 days, then it is. 12/30/2014, 2 days before 01/01/2015 is, 12/30/2014, so today is. 01/05/2015, so one week from today will be.

◆ $\mathcal{A}$ ↦ 01/05/2015

---

❖ **Prompt 6**

➥ $\mathcal{Q}$ ↦ Jane thought today is 3/11/2002, but today is in fact Mar 12, which is 1 day later. What is the date 24 hours later in MM/DD/YYYY?

❖ $\mathcal{T}$ ↦ 03/12/2002, today is. Later will be 03/13/2002, so the date 24 hours.

◆ $\mathcal{A}$ ↦ 03/13/2002

---

Table 55: GSM-8ᴋ with examples in YodaSpeak.

---

**❖ Prompt 1**

➥ 𝒬 ↦ Five toys, shawn has. For Christmas, two toys each from his mom and dad, he got. He have now, how many toys does?

❖ 𝒯 ↦ With 5 toys, Shawn started. 2 toys each from his mom and dad, if he got, then that is 4 more toys. 5 + 4 = 9.

◆ 𝒜 ↦ 9

**❖ Prompt 2**

➥ 𝒬 ↦ 3 cars in the parking lot and 2 more cars arrive, if there are,In the parking lot, how many cars are?

❖ 𝒯 ↦ Originally 3 cars, there are. 2 more cars arrive. 3 + 2 = 5.

◆ 𝒜 ↦ 5

**❖ Prompt 3**

➥ 𝒬 ↦ 20 lollipops, Jason had. Denny some lollipops, he gave. 12 lollipops, now Jason has. Jason give to Denny, how many lollipops did?

❖ 𝒯 ↦ With 20 lollipops, Jason started. 12 after giving some to Denny, then he had. So he gave Denny 20 - 12 = 8.

◆ 𝒜 ↦ 8

**❖ Prompt 4**

➥ 𝒬 ↦ Nine computers in the server room, there were. Installed each day, five more computers were, from Monday to Thursday. Now in the server room, how many computers are?

❖ 𝒯 ↦ Originally 9 computers, there were. For each of 4 days, added, 5 more computers were. So 5 * 4 = 20 computers were added. 9 + 20 is 29.

◆ 𝒜 ↦ 29

**❖ Prompt 5**

➥ 𝒬 ↦ 15 trees in the grove, there are. Grove workers will plant trees in the grove today. Done, after they are,21 trees, there will be. The grove workers plant today, how many trees did?

❖ 𝒯 ↦ 15 trees originally, there are. 21 trees after some more were planted, then there were. Been 21 - 15 = 6, so there must have.

◆ 𝒜 ↦ 6

**❖ Prompt 6**

➥ 𝒬 ↦ 32 chocolates and her sister had 42, Leah had. 35, if they ate, they have left in total, how many pieces do?

❖ 𝒯 ↦ Originally, 32 chocolates, Leah had. 42, her sister had. 32 + 42 = 74, so in total they had. After eating 35,74 - 35 = 39, they had.

◆ 𝒜 ↦ 39

**❖ Prompt 7**

➥ 𝒬 ↦ $23, Olivia has. Five bagels for $3 each, she bought. She have left, how much money does?

❖ 𝒯 ↦ 23 dollars, Olivia had. 5 x 3 = 15 dollars, 5 bagels for 3 dollars each will be. 23 - 15 dollars left, so she has. 23 - 15 is 8.

◆ 𝒜 ↦ 8

**❖ Prompt 8**

➥ 𝒬 ↦ 58 golf balls, Michael had. On Tuesday,23 golf balls, he lost. On Wednesday,2 more, he lost. He have at the end of Wednesday, how many golf balls did?

❖ 𝒯 ↦ With 58 golf balls, Michael started. After losing 23 on Tuesday,58 - 23 = 35, he had. After losing 2 more,35 - 2 = 33 golf balls, he had.

◆ 𝒜 ↦ 33

Table 56: SPORTS with examples in YodaSpeak.

---

❖ **Prompt 1**

➥ $\mathcal{Q} \mapsto$ Is the following sentence plausible? "Perfect from the line Jamal Murray was."

❖ $\mathcal{T} \mapsto$ A basketball player Jamal Murray is. Perfect from the line is part of basketball being.

◆ $\mathcal{A} \mapsto$ yes

---

❖ **Prompt 2**

➥ $\mathcal{Q} \mapsto$ Is the following sentence plausible? "The screen pass in the NFC championship Joao Moutinho caught."

❖ $\mathcal{T} \mapsto$ A soccer player Joao Moutinho is. Part of American football the NFC championship is, not soccer.

◆ $\mathcal{A} \mapsto$ no

---

❖ **Prompt 3**

➥ $\mathcal{Q} \mapsto$ Is the following sentence plausible? "The buzzer Jonas Valanciunas beat."

❖ $\mathcal{T} \mapsto$ A basketball player Jonas Valanciunas is. The buzzer is part of basketball beating.

◆ $\mathcal{A} \mapsto$ yes

---

❖ **Prompt 4**

➥ $\mathcal{Q} \mapsto$ Is the following sentence plausible? "Passed the puck, Sam Darnold did."

❖ $\mathcal{T} \mapsto$ An American football player Sam Darnold is. Part of hockey passing the puck is, not American football.

◆ $\mathcal{A} \mapsto$ no

---

❖ **Prompt 5**

➥ $\mathcal{Q} \mapsto$ Is the following sentence plausible? "Called for slashing Kyle Palmieri was."

❖ $\mathcal{T} \mapsto$ A hockey player Kyle Palmieri is. Called for slashing is part of hockey being.

◆ $\mathcal{A} \mapsto$ yes

---

❖ **Prompt 6**

➥ $\mathcal{Q} \mapsto$ Is the following sentence plausible? "The pick and roll Carson Wentz set."

❖ $\mathcal{T} \mapsto$ An American football player Carson Wentz is. Part of basketball pick and roll is, not football.

◆ $\mathcal{A} \mapsto$ no

---

❖ **Prompt 7**

➥ $\mathcal{Q} \mapsto$ is the following sentence plausible? In "Malcolm Brogdon banked the shot."

❖ $\mathcal{T} \mapsto$ A basketball player Malcolm Brogdon is. In is part of basketball banking the shot.

◆ $\mathcal{A} \mapsto$ yes

---

❖ **Prompt 8**

➥ $\mathcal{Q} \mapsto$ is the following sentence plausible? A touchdown "Draymond Green threw."

❖ $\mathcal{T} \mapsto$ An basketball player Draymond Green is. A touchdown is part of football throwing, not basketball.

◆ $\mathcal{A} \mapsto$ no

---

Table 57: DATE with examples in YodaSpeak.

---

❖ **Prompt 1**

➥ 𝒬 ↦ 4/19/1969 today it is. The date 24 hours later in MM/DD/YYYY what is?

◈ 𝒯 ↦ 04/19/1969, today is. Later is one day after today, 24 hours, 04/20/1969, which would be.

◆ 𝒜 ↦ 04/20/1969

---

❖ **Prompt 2**

➥ 𝒬 ↦ Scheduled to be on 06/01/1943 the concert was, delayed by one day to today but was. The date 10 days ago in MM/DD/YYYY what is?

◈ 𝒯 ↦ 06/02/1943, one day after 06/01/1943 is, 06/02/1943, so today is. 05/23/1943, 10 days before today is.

◆ 𝒜 ↦ 05/23/1943

---

❖ **Prompt 3**

➥ 𝒬 ↦ A Tuesday the first day of 2019 is, the first Monday of 2019 and today is. The date today in MM/DD/YYYY what is?

◈ 𝒯 ↦ Tuesday, if the first day of 2019 was, a Tuesday, then 01/01/2019 was. The first Monday, today is, six days later, would be. 01/07/2019, so today is.

◆ 𝒜 ↦ 01/07/2019

---

❖ **Prompt 4**

➥ 𝒬 ↦ Born on the last day of February in 2001 Jane was. Her 16-year-old birthday today is. The date yesterday in MM/DD/YYYY what is?

◈ 𝒯 ↦ The 28th, the last day of February is, Born on 02/28/2001, so Jane was. Her 16-year old birthday, today is,02/28/2017, so today is. 02/27/2017, so yesterday was.

◆ 𝒜 ↦ 02/27/2017

---

❖ **Prompt 5**

➥ 𝒬 ↦ Coming in 36 hours 2015 is. The date one week from today in MM/DD/YYYY what is? Yes.

◈ 𝒯 ↦ Coming in 36 hours, if 2015 is, coming in 2 days, then it is. 12/30/2014, 2 days before 01/01/2015 is, 12/30/2014, so today is. 01/05/2015, so one week from today will be.

◆ 𝒜 ↦ 01/05/2015

---

❖ **Prompt 6**

➥ 𝒬 ↦ Today is 3/11/2002 Jane thought, in fact mar 12 but today is, 1 day later which is. The date 24 hours later in MM/DD/YYYY what is?

◈ 𝒯 ↦ 03/12/2002, today is. Later will be 03/13/2002, so the date 24 hours.

◆ 𝒜 ↦ 03/13/2002

Table 58: GSM-8ᴋ with random thoughts.

---

❖ **Prompt 1**

➥ $\mathcal{Q} \mapsto$ Shawn has five toys. For Christmas, he got two toys each from his mom and dad. How many toys does he have now?

◈ $\mathcal{T} \mapsto$ Initially, Steve is 66 inches tall. After growing 6 inches, Steve is 66 + 6 = 72 inches tall.

◆ $\mathcal{A} \mapsto$ 9

❖ **Prompt 2**

➥ $\mathcal{Q} \mapsto$ If there are 3 cars in the parking lot and 2 more cars arrive, how many cars are in the parking lot?

◈ $\mathcal{T} \mapsto$ Sandra eats 3 beignets every morning and there are 7 days in a week so she eats 3 * 7 = 21 beignets in a week.

◆ $\mathcal{A} \mapsto$ 5

❖ **Prompt 3**

➥ $\mathcal{Q} \mapsto$ Jason had 20 lollipops. He gave Denny some lollipops. Now Jason has 12 lollipops. How many lollipops did Jason give to Denny?

◈ $\mathcal{T} \mapsto$ The quarter of the number is 1, thus the number is 1 * 4 = 4.

◆ $\mathcal{A} \mapsto$ 8

❖ **Prompt 4**

➥ $\mathcal{Q} \mapsto$ There were nine computers in the server room. Five more computers were installed each day, from Monday to Thursday. How many computers are now in the server room?

◈ $\mathcal{T} \mapsto$ Half of 10 is 10 / 2 = 5. Five more than 5 is 5 + 5 = 10.

◆ $\mathcal{A} \mapsto$ 29

❖ **Prompt 5**

➥ $\mathcal{Q} \mapsto$ There are 15 trees in the grove. Grove workers will plant trees in the grove today. After they are done, there will be 21 trees. How many trees did the grove workers plant today?

◈ $\mathcal{T} \mapsto$ Angie's age minus 4 is 20 - 4 = 16. Thus, Angie's age is 16.

◆ $\mathcal{A} \mapsto$ 6

❖ **Prompt 6**

➥ $\mathcal{Q} \mapsto$ Leah had 32 chocolates and her sister had 42. If they ate 35, how many pieces do they have left in total?

◈ $\mathcal{T} \mapsto$ A total of 20 passengers are in two buses. Thus, one bus can fit 20 passengers / 2 buses = 10 passengers.

◆ $\mathcal{A} \mapsto$ 39

❖ **Prompt 7**

➥ $\mathcal{Q} \mapsto$ Olivia has $23. She bought five bagels for $3 each. How much money does she have left?

◈ $\mathcal{T} \mapsto$ 2 bags weight 2 pounds, so one bag weights 2 pounds / 2 = 1 pound. So 8 bags of oranges would weigh 8 * 1 pound = 8 pounds.

◆ $\mathcal{A} \mapsto$ 8

❖ **Prompt 8**

➥ $\mathcal{Q} \mapsto$ Michael had 58 golf balls. On Tuesday, he lost 23 golf balls. On Wednesday, he lost 2 more. How many golf balls did he have at the end of Wednesday?

◈ $\mathcal{T} \mapsto$ John takes 2 pills a day. In a week, John therefore takes 2 * 7 = 14 pills.

◆ $\mathcal{A} \mapsto$ 33

Table 59: SPORTS with random thoughts.

---

❖**Prompt 1**

➥ $\mathcal{Q}$ ↦ Is the following sentence plausible? "Jamal Murray was perfect from the line."

❖ $\mathcal{T}$ ↦ Sam Darnold is a American football player. Passing the puck is part of hockey, not American football.

◆ $\mathcal{A}$ ↦ yes

---

❖**Prompt 2**

➥ $\mathcal{Q}$ ↦ Is the following sentence plausible? "Joao Moutinho caught the screen pass in the NFC championship."

❖ $\mathcal{T}$ ↦ Draymond Green is an basketball player. Throwing a touchdown is part of football, not basketball.

◆ $\mathcal{A}$ ↦ no

---

❖**Prompt 3**

➥ $\mathcal{Q}$ ↦ Is the following sentence plausible? "Jonas Valanciunas beat the buzzer."

❖ $\mathcal{T}$ ↦ Carson Wentz is an American football player. Pick and roll is part of basketball, not football.

◆ $\mathcal{A}$ ↦ yes

---

❖**Prompt 4**

➥ $\mathcal{Q}$ ↦ Is the following sentence plausible? "Sam Darnold passed the puck."

❖ $\mathcal{T}$ ↦ Jamal Murray is a basketball player. Being perfect from the line is part of basketball.

◆ $\mathcal{A}$ ↦ no

---

❖**Prompt 5**

➥ $\mathcal{Q}$ ↦ Is the following sentence plausible? "Kyle Palmieri was called for slashing."

❖ $\mathcal{T}$ ↦ Joao Moutinho is a soccer player. The NFC championship is part of American football, not soccer.

◆ $\mathcal{A}$ ↦ yes

---

❖**Prompt 6**

➥ $\mathcal{Q}$ ↦ Is the following sentence plausible? "Carson Wentz set the pick and roll."

❖ $\mathcal{T}$ ↦ Malcolm Brogdon is a basketball player. Banking the shot in is part of basketball.

◆ $\mathcal{A}$ ↦ no

---

❖**Prompt 7**

➥ $\mathcal{Q}$ ↦ Is the following sentence plausible? "Malcolm Brogdon banked the shot in."

❖ $\mathcal{T}$ ↦ Jonas Valanciunas is a basketball player. Beating the buzzer is part of basketball.

◆ $\mathcal{A}$ ↦ yes

---

❖**Prompt 8**

➥ $\mathcal{Q}$ ↦ Is the following sentence plausible? "Draymond Green threw a touchdown."

❖ $\mathcal{T}$ ↦ Kyle Palmieri is a hockey player. Being called for slashing is part of hockey.

◆ $\mathcal{A}$ ↦ no

---

Table 60: DATE with random thoughts.

---

❖ **Prompt 1**

➥ $\mathcal{Q} \mapsto$ It is 4/19/1969 today. What is the date 24 hours later in MM/DD/YYYY?

◆ $\mathcal{T} \mapsto$ The last day of February is the 28th, so Jane was born on 02/28/2001. Today is her 16-year old birthday, so today is 02/28/2017. So yesterday was 02/27/2017.

◆ $\mathcal{A} \mapsto$ 04/20/1969

❖ **Prompt 2**

➥ $\mathcal{Q} \mapsto$ The concert was scheduled to be on 06/01/1943, but was delayed by one day to today. What is the date 10 days ago in MM/DD/YYYY?

◆ $\mathcal{T} \mapsto$ Today is 03/12/2002. So the date 24 hours later will be 03/13/2002.

◆ $\mathcal{A} \mapsto$ 05/23/1943

❖ **Prompt 3**

➥ $\mathcal{Q} \mapsto$ The first day of 2019 is a Tuesday, and today is the first Monday of 2019. What is the date today in MM/DD/YYYY?

◆ $\mathcal{T} \mapsto$ Today is 04/19/1969. 24 hours later is one day after today, which would be 04/20/1969.

◆ $\mathcal{A} \mapsto$ 01/07/2019

❖ **Prompt 4**

➥ $\mathcal{Q} \mapsto$ Jane was born on the last day of February in 2001. Today is her 16-year-old birthday. What is the date yesterday in MM/DD/YYYY?

◆ $\mathcal{T} \mapsto$ One day after 06/01/1943 is 06/02/1943, so today is 06/02/1943. 10 days before today is 05/23/1943.

◆ $\mathcal{A} \mapsto$ 02/27/2017

❖ **Prompt 5**

➥ $\mathcal{Q} \mapsto$ 2015 is coming in 36 hours. What is the date one week from today in MM/DD/YYYY?

◆ $\mathcal{T} \mapsto$ If the first day of 2019 was Tuesday, then 01/01/2019 was a Tuesday. Today is the first Monday, would be six days later. So today is 01/07/2019.

◆ $\mathcal{A} \mapsto$ 01/05/2015

❖ **Prompt 6**

➥ $\mathcal{Q} \mapsto$ Jane thought today is 3/11/2002, but today is in fact Mar 12, which is 1 day later. What is the date 24 hours later in MM/DD/YYYY?

◆ $\mathcal{T} \mapsto$ If 2015 is coming in 36 hours, then it is coming in 2 days. 2 days before 01/01/2015 is 12/30/2014, so today is 12/30/2014. So one week from today will be 01/05/2015.

◆ $\mathcal{A} \mapsto$ 03/13/2002

Table 61: GSM-8ᴋ with shuffled tokens within each sentence. The location of numbers is fixed.

---

❖ **Prompt 1**

➡ $\mathcal{Q}$ ↦ Shawn has five toys. For Christmas, he got two toys each from his mom and dad. How many toys does he have now?

◈ $\mathcal{T}$ ↦ with Shawn toys 5 started. dad, from more 2 his toys then is toys he mom got that each 4 and If. 5 + 4 = 9

◆ $\mathcal{A}$ ↦ 9

---

❖ **Prompt 2**

➡ $\mathcal{Q}$ ↦ If there are 3 cars in the parking lot and 2 more cars arrive, how many cars are in the parking lot?

◈ $\mathcal{T}$ ↦ originally cars There 3 are. 2 arrive more cars. 3 + 2 = 5

◆ $\mathcal{A}$ ↦ 5

---

❖ **Prompt 3**

➡ $\mathcal{Q}$ ↦ Jason had 20 lollipops. He gave Denny some lollipops. Now Jason has 12 lollipops. How many lollipops did Jason give to Denny?

◈ $\mathcal{T}$ ↦ lollipops with started 20 Jason. had after to 12 Denny Then some giving he. he gave Denny So 20 - 12 = 8

◆ $\mathcal{A}$ ↦ 8

---

❖ **Prompt 4**

➡ $\mathcal{Q}$ ↦ There were nine computers in the server room. Five more computers were installed each day, from monday to thursday. How many computers are now in the server room?

◈ $\mathcal{T}$ ↦ were There originally 9 computers. more For each 4 computers 5 of added days, were. computers 5 * 4 = 20 were added So. 9 + 20 is 29

◆ $\mathcal{A}$ ↦ 29

---

❖ **Prompt 5**

➡ $\mathcal{Q}$ ↦ There are 15 trees in the grove. Grove workers will plant trees in the grove today. After they are done, there will be 21 trees. How many trees did the grove workers plant today?

◈ $\mathcal{T}$ ↦ are There 15 originally trees. planted were some 21 more Then after there trees were. must So there been have 21 - 15 = 6

◆ $\mathcal{A}$ ↦ 6

---

❖ **Prompt 6**

➡ $\mathcal{Q}$ ↦ Leah had 32 chocolates and her sister had 42. If they ate 35, how many pieces do they have left in total?

◈ $\mathcal{T}$ ↦ Originally, had chocolates 32 Leah. Her sister had 42. total had they in So 32 + 42 = 74. eating had 35, After they 74 - 35 = 39

◆ $\mathcal{A}$ ↦ 39

---

❖ **Prompt 7**

➡ $\mathcal{Q}$ ↦ Olivia has $23. She bought five bagels for $3 each. How much money does she have left?

◈ $\mathcal{T}$ ↦ had Olivia 23 dollars. 5 dollars be 3 each dollars bagels for 5 x 3 = 15 will. dollars So she 23 - 15 has left. 23 - 15 is 8

◆ $\mathcal{A}$ ↦ 8

---

❖ **Prompt 8**

➡ $\mathcal{Q}$ ↦ Michael had 58 golf balls. On Tuesday, he lost 23 golf balls. On Wednesday, he lost 2 more. How many golf balls did he have at the end of Wednesday?

◈ $\mathcal{T}$ ↦ Michael started balls 58 with golf. had After 23 losing Tuesday, he on 58 - 23 = 35. golf losing 2 balls more, he 35 - 2 = 33 After had

◆ $\mathcal{A}$ ↦ 33

---

Table 62: SPORTS with shuffled tokens within each sentence.

**❖ Prompt 1**

➥ $\mathcal{Q}$ ↦ Is the following sentence plausible? "Jamal Murray was perfect from the line."

❖ $\mathcal{T}$ ↦ is a player Jamal basketball Murray. from line Being perfect part is the basketball of

◆ $\mathcal{A}$ ↦ yes

**❖ Prompt 2**

➥ $\mathcal{Q}$ ↦ Is the following sentence plausible? "Joao Moutinho caught the screen pass in the NFC championship."

❖ $\mathcal{T}$ ↦ Moutinho player is soccer a Joao. NFC American soccer of championship The not is part football,

◆ $\mathcal{A}$ ↦ no

**❖ Prompt 3**

➥ $\mathcal{Q}$ ↦ Is the following sentence plausible? "Jonas Valanciunas beat the buzzer."

❖ $\mathcal{T}$ ↦ is a Valanciunas Jonas basketball player. part buzzer basketball the Beating of is

◆ $\mathcal{A}$ ↦ yes

**❖ Prompt 4**

➥ $\mathcal{Q}$ ↦ Is the following sentence plausible? "Sam Darnold passed the puck."

❖ $\mathcal{T}$ ↦ a Sam player football is Darnold American. of hockey, puck the football American is Passing part not

◆ $\mathcal{A}$ ↦ no

**❖ Prompt 5**

➥ $\mathcal{Q}$ ↦ Is the following sentence plausible? "Kyle Palmieri was called for slashing."

❖ $\mathcal{T}$ ↦ Palmieri player hockey a is Kyle. called Being of part slashing hockey for is

◆ $\mathcal{A}$ ↦ yes

**❖ Prompt 6**

➥ $\mathcal{Q}$ ↦ Is the following sentence plausible? "Carson Wentz set the pick and roll."

❖ $\mathcal{T}$ ↦ football Carson is American player Wentz an. roll and not basketball, part is Pick football of

◆ $\mathcal{A}$ ↦ no

**❖ Prompt 7**

➥ $\mathcal{Q}$ ↦ Is the following sentence plausible? "Malcolm Brogdon banked the shot in."

❖ $\mathcal{T}$ ↦ player basketball Brogdon Malcolm a is. the Banking shot in of basketball part is

◆ $\mathcal{A}$ ↦ yes

**❖ Prompt 8**

➥ $\mathcal{Q}$ ↦ Is the following sentence plausible? "Draymond Green threw a touchdown."

❖ $\mathcal{T}$ ↦ an is Draymond player basketball Green. Throwing football, a of touchdown part not is basketball

◆ $\mathcal{A}$ ↦ no

Table 63: DATE with shuffled tokens of each sentence.

**❖ Prompt 1**

➥ $\mathcal{Q} \mapsto$ It is 4/19/1969 today. What is the date 24 hours later in MM/DD/YYYY?

◈ $\mathcal{T} \mapsto$ Today 04/19/1969 is. 24 after one which later be would hours 04/20/1969 day today, is

◆ $\mathcal{A} \mapsto$ 04/20/1969

**❖ Prompt 2**

➥ $\mathcal{Q} \mapsto$ The concert was scheduled to be on 06/01/1943, but was delayed by one day to today. What is the date 10 days ago in MM/DD/YYYY?

◈ $\mathcal{T} \mapsto$ is 06/02/1943, so today 06/02/1943 day 06/01/1943 One is after. 10 today days is before 05/23/1943

◆ $\mathcal{A} \mapsto$ 05/23/1943

**❖ Prompt 3**

➥ $\mathcal{Q} \mapsto$ The first day of 2019 is a Tuesday, and today is the first Monday of 2019. What is the date today in MM/DD/YYYY?

◈ $\mathcal{T} \mapsto$ was Tuesday the day was 2019 first If then 01/01/2019 of Tuesday, a. days Today six later is Monday, be would first the. today So 01/07/2019 is

◆ $\mathcal{A} \mapsto$ 01/07/2019

**❖ Prompt 4**

➥ $\mathcal{Q} \mapsto$ Jane was born on the last day of February in 2001. Today is her 16-year-old birthday. What is the date yesterday in MM/DD/YYYY?

◈ $\mathcal{T} \mapsto$ last day Jane February on was The of 28th, born is so the 02/28/2001. Today today is birthday, old so 02/28/2017 16-year is her. 02/27/2017 yesterday So was

◆ $\mathcal{A} \mapsto$ 02/27/2017

**❖ Prompt 5**

➥ $\mathcal{Q} \mapsto$ 2015 is coming in 36 hours. What is the date one week from today in MM/DD/YYYY?

◈ $\mathcal{T} \mapsto$ days 2015 is then coming 36 If it hours, coming is in 2 in. 2 is 12/30/2014, days 01/01/2015 so 12/30/2014 is today before. from week will So 01/05/2015 be today one

◆ $\mathcal{A} \mapsto$ 01/05/2015

**❖ Prompt 6**

➥ $\mathcal{Q} \mapsto$ Jane thought today is 3/11/2002, but today is in fact Mar 12, which is 1 day later. What is the date 24 hours later in MM/DD/YYYY?

◈ $\mathcal{T} \mapsto$ 03/12/2002 Today is. the hours later 24 be will So date 03/13/2002

◆ $\mathcal{A} \mapsto$ 03/13/2002

Table 64: GSM-8ᴋ with sentences within each thought are shuffled. The location of numbers is fixed.

---

❖ **Prompt 1**

➥ 𝒬 ↦ Shawn has five toys. For Christmas, he got two toys each from his mom and dad. How many toys does he have now?

◆ 𝒯 ↦ dad, he got 5 toys. then started mom 2 each is more that from If his and toys. toys 4 with Shawn 5 + 4 = 9.

◆ 𝒜 ↦ 9

❖ **Prompt 2**

➥ 𝒬 ↦ If there are 3 cars in the parking lot and 2 more cars arrive, how many cars are in the parking lot?

◆ 𝒯 ↦ arrive. more are 3 cars 2 originally cars. There 3 + 2 = 5.

◆ 𝒜 ↦ 5

❖ **Prompt 3**

➥ 𝒬 ↦ Jason had 20 lollipops. He gave Denny some lollipops. Now Jason has 12 lollipops. How many lollipops did Jason give to Denny?

◆ 𝒯 ↦ giving So started 20 some gave to Denny 12 Jason Denny. Then after had he with he lollipops. 20 - 12 = 8.

◆ 𝒜 ↦ 8

❖ **Prompt 4**

➥ 𝒬 ↦ There were nine computers in the server room. Five more computers were installed each day, from Monday to Thursday. How many computers are now in the server room?

◆ 𝒯 ↦ originally were were 9 computers added. is For 4 each 5 computers. added. of days, more 5 * 4 = 20 computers were There 9 + 20 So 29.

◆ 𝒜 ↦ 29

❖ **Prompt 5**

➥ 𝒬 ↦ There are 15 trees in the grove. Grove workers will plant trees in the grove today. After they are done, there will be 21 trees. How many trees did the grove workers plant today?

◆ 𝒯 ↦ So were 15 there after more have planted. 21 trees originally. must There are there were trees some Then been 21 - 15 = 6.

◆ 𝒜 ↦ 6

❖ **Prompt 6**

➥ 𝒬 ↦ Leah had 32 chocolates and her sister had 42. If they ate 35, how many pieces do they have left in total?

◆ 𝒯 ↦ After Originally, total 32 they sister had had 42. had eating in Her So 32 + 42 = 74. had they Leah 35, chocolates. 74 - 35 = 39.

◆ 𝒜 ↦ 39

❖ **Prompt 7**

➥ 𝒬 ↦ Olivia has $23. She bought five bagels for $3 each. How much money does she have left?

◆ 𝒯 ↦ bagels dollars 23 has 5 Olivia will 3 is dollars. left. for 5 had 3 = 15 dollars. So be x 23 - 15 dollars each 23 - 15 she 8.

◆ 𝒜 ↦ 8

❖ **Prompt 8**

➥ 𝒬 ↦ Michael had 58 golf balls. On Tuesday, he lost 23 golf balls. On Wednesday, he lost 2 more. How many golf balls did he have at the end of Wednesday?

◆ 𝒯 ↦ he golf Michael 58 golf After started with 23 more, losing had balls. 58 - 23 = 35. losing After 2 on Tuesday, he 35 - 2 = 33 balls. had

◆ 𝒜 ↦ 33

Table 65: SPORTS with sentences within each thought are shuffled.

---

❖ **Prompt 1**

➥ $\mathcal{Q} \mapsto$ Is the following sentence plausible? "Jamal Murray was perfect from the line."

❖ $\mathcal{T} \mapsto$ perfect player. basketball. a Murray part basketball is of Being Jamal is the from line

◆ $\mathcal{A} \mapsto$ yes

---

❖ **Prompt 2**

➥ $\mathcal{Q} \mapsto$ Is the following sentence plausible? "Joao Moutinho caught the screen pass in the NFC championship."

❖ $\mathcal{T} \mapsto$ American soccer. Moutinho not soccer Joao is part player. NFC The football, a championship is of

◆ $\mathcal{A} \mapsto$ no

---

❖ **Prompt 3**

➥ $\mathcal{Q} \mapsto$ Is the following sentence plausible? "Jonas Valanciunas beat the buzzer."

❖ $\mathcal{T} \mapsto$ Valanciunas of player. the basketball Jonas Beating is buzzer is part a basketball.

◆ $\mathcal{A} \mapsto$ yes

---

❖ **Prompt 4**

➥ $\mathcal{Q} \mapsto$ Is the following sentence plausible? "Sam Darnold passed the puck."

❖ $\mathcal{T} \mapsto$ football is puck the hockey, player. not football. Darnold part a American of American Passing Sam is

◆ $\mathcal{A} \mapsto$ no

---

❖ **Prompt 5**

➥ $\mathcal{Q} \mapsto$ Is the following sentence plausible? "Kyle Palmieri was called for slashing."

❖ $\mathcal{T} \mapsto$ hockey. of Palmieri slashing Kyle Being a player. called part hockey is is for

◆ $\mathcal{A} \mapsto$ yes

---

❖ **Prompt 6**

➥ $\mathcal{Q} \mapsto$ Is the following sentence plausible? "Carson Wentz set the pick and roll."

❖ $\mathcal{T} \mapsto$ is football not football. Carson Pick roll basketball, and part Wentz American an of player. is

◆ $\mathcal{A} \mapsto$ no

---

❖ **Prompt 7**

➥ $\mathcal{Q} \mapsto$ Is the following sentence plausible? "Malcolm Brogdon banked the shot in."

❖ $\mathcal{T} \mapsto$ player. is of a in Brogdon basketball Banking shot the basketball. part Malcolm is

◆ $\mathcal{A} \mapsto$ yes

---

❖ **Prompt 8**

➥ $\mathcal{Q} \mapsto$ Is the following sentence plausible? "Draymond Green threw a touchdown."

❖ $\mathcal{T} \mapsto$ of an a is touchdown football, Green basketball Draymond basketball. not player. is part Throwing

◆ $\mathcal{A} \mapsto$ no

---

Table 66: DATE with sentences within each thought are shuffled.

---

❖ **Prompt 1**

➡ $\mathcal{Q}$ ↦ It is 4/19/1969 today. What is the date 24 hours later in MM/DD/YYYY?

◈ $\mathcal{T}$ ↦ later is 04/19/1969. 24 day after which would be today, hours Today 04/20/1969. one is

◆ $\mathcal{A}$ ↦ 04/20/1969

---

❖ **Prompt 2**

➡ $\mathcal{Q}$ ↦ The concert was scheduled to be on 06/01/1943, but was delayed by one day to today. What is the date 10 days ago in MM/DD/YYYY?

◈ $\mathcal{T}$ ↦ today today days One 05/23/1943. 06/01/1943 06/02/1943, 06/02/1943. is is 10 is so day before after

◆ $\mathcal{A}$ ↦ 05/23/1943

---

❖ **Prompt 3**

➡ $\mathcal{Q}$ ↦ The first day of 2019 is a Tuesday, and today is the first Monday of 2019. What is the date today in MM/DD/YYYY?

◈ $\mathcal{T}$ ↦ 01/01/2019 today Monday, would Tuesday, 2019 first is Tuesday. day of later. then So was 01/07/2019. first be days Today the a six was is If the

◆ $\mathcal{A}$ ↦ 01/07/2019

---

❖ **Prompt 4**

➡ $\mathcal{Q}$ ↦ Jane was born on the last day of February in 2001. Today is her 16-year-old birthday. What is the date yesterday in MM/DD/YYYY?

◈ $\mathcal{T}$ ↦ last was her 16-year is 02/28/2001. So of so 02/27/2017. is birthday, is The February on yesterday was old today Today 02/28/2017. so day born 28th, the Jane

◆ $\mathcal{A}$ ↦ 02/27/2017

---

❖ **Prompt 5**

➡ $\mathcal{Q}$ ↦ 2015 is coming in 36 hours. What is the date one week from today in MM/DD/YYYY?

◈ $\mathcal{T}$ ↦ in 2015 one hours, 01/01/2015 36 so it days. today 12/30/2014, then 2 week 2 is 12/30/2014. is be days in is 01/05/2015. So from today coming coming before is will If

◆ $\mathcal{A}$ ↦ 01/05/2015

---

❖ **Prompt 6**

➡ $\mathcal{Q}$ ↦ Jane thought today is 3/11/2002, but today is in fact Mar 12, which is 1 day later. What is the date 24 hours later in MM/DD/YYYY?

◈ $\mathcal{T}$ ↦ later So date Today 03/13/2002. hours 24 is 03/12/2002. the be will

◆ $\mathcal{A}$ ↦ 03/13/2002

---

Table 67: GSM-8K with questions in YodaSpeak.

**❖ Prompt 1**

➥ $\mathcal{Q} \mapsto$ Five toys, Shawn has.For Christmas,Two toys each from his mom and dad, he got.He have now, how many toys does?

◆ $\mathcal{T} \mapsto$ Shawn started with 5 toys. If he got 2 toys each from his mom and dad, then that is 4 more toys. 5 + 4 = 9.

◆ $\mathcal{A} \mapsto$ 9

**❖ Prompt 2**

➥ $\mathcal{Q} \mapsto$ 3 cars in the parking lot and 2 more cars arrive, if there are,In the parking lot, how many cars are?

◆ $\mathcal{T} \mapsto$ There are originally 3 cars. 2 more cars arrive. 3 + 2 = 5.

◆ $\mathcal{A} \mapsto$ 5

**❖ Prompt 3**

➥ $\mathcal{Q} \mapsto$ 20 lollipops, Jason had. Denny some lollipops, he gave. 12 lollipops, now Jason has. Jason give to Denny, how many lollipops did?

◆ $\mathcal{T} \mapsto$ Jason started with 20 lollipops. Then he had 12 after giving some to Denny. So he gave Denny 20 - 12 = 8.

◆ $\mathcal{A} \mapsto$ 8

**❖ Prompt 4**

➥ $\mathcal{Q} \mapsto$ Nine computers in the server room, there were. Installed each day, five more computers were, from Monday to Thursday. Now in the server room, how many computers are?

◆ $\mathcal{T} \mapsto$ There were originally 9 computers. For each of 4 days, 5 more computers were added. So 5 * 4 = 20 computers were added. 9 + 20 is 29.

◆ $\mathcal{A} \mapsto$ 29

**❖ Prompt 5**

➥ $\mathcal{Q} \mapsto$ 15 trees in the grove, there are. Grove workers will plant trees in the grove today. Done, after they are,21 trees, there will be. The grove workers plant today, how many trees did?

◆ $\mathcal{T} \mapsto$ There are 15 trees originally. Then there were 21 trees after some more were planted. So there must have been 21 - 15 = 6.

◆ $\mathcal{A} \mapsto$ 6

**❖ Prompt 6**

➥ $\mathcal{Q} \mapsto$ 32 chocolates and her sister had 42, Leah had. 35, if they ate, they have left in total, how many pieces do?

◆ $\mathcal{T} \mapsto$ Originally, Leah had 32 chocolates. Her sister had 42. So in total they had 32 + 42 = 74. After eating 35, they had 74 - 35 = 39.

◆ $\mathcal{A} \mapsto$ 39

**❖ Prompt 7**

➥ $\mathcal{Q} \mapsto$ $23, Olivia has. Five bagels for $3 each, she bought. She have left, how much money does?

◆ $\mathcal{T} \mapsto$ Olivia had 23 dollars. 5 bagels for 3 dollars each will be 5 x 3 = 15 dollars. So she has 23 - 15 dollars left. 23 - 15 is 8.

◆ $\mathcal{A} \mapsto$ 8

**❖ Prompt 8**

➥ $\mathcal{Q} \mapsto$ 58 golf balls, Michael had. On Tuesday, 23 golf balls, he lost. On Wednesday, 2 more, he lost. He have at the end of Wednesday, how many golf balls did?

◆ $\mathcal{T} \mapsto$ Michael started with 58 golf balls. After losing 23 on Tuesday, he had 58 - 23 = 35. After losing 2 more, he had 35 - 2 = 33 golf balls.

◆ $\mathcal{A} \mapsto$ 33

Table 68: SORTING with verbose chain of thought.

---

❖ **Prompt 1**

➥ $\mathcal{Q} \mapsto$ 7 , 8 , 4 , 1 , 2 , 9 , 3 , 6 , 5

◈ $\mathcal{T} \mapsto$ First number is 1. Next we have 2, since 1 is less than 2. Next we have 3, since 2 is less than 3. Next we have 4, since 3 is less than 4. Next we have 5, since 4 is less than 5. Next we have 6, since 5 is less than 6. Next we have 7, since 6 is less than 7. Next we have 8, since 7 is less than 8. Next we have 9, since 8 is less than 9.

◆ $\mathcal{A} \mapsto$ 1 , 2 , 3 , 4 , 5 , 6 , 7 , 8 , 9

❖ **Prompt 2**

➥ $\mathcal{Q} \mapsto$ 5 , 9 , 3 , 1 , 8 , 4 , 6 , 2

◈ $\mathcal{T} \mapsto$ First number is 1. Next we have 2, since 1 is less than 2. Next we have 3, since 2 is less than 3. Next we have 4, since 3 is less than 4. Next we have 5, since 4 is less than 5. Next we have 6, since 5 is less than 6. Next we have 8, since 6 is less than 8. Next we have 9, since 8 is less than 9.

◆ $\mathcal{A} \mapsto$ 1 , 2 , 3 , 4 , 5 , 6 , 8 , 9

❖ **Prompt 3**

➥ $\mathcal{Q} \mapsto$ 6 , 5 , 7 , 4 , 3 , 2 , 8 , 1

◈ $\mathcal{T} \mapsto$ First number is 1. Next we have 2, since 1 is less than 2. Next we have 3, since 2 is less than 3. Next we have 4, since 3 is less than 4. Next we have 5, since 4 is less than 5. Next we have 6, since 5 is less than 6. Next we have 7, since 6 is less than 7. Next we have 8, since 7 is less than 8.

◆ $\mathcal{A} \mapsto$ 1 , 2 , 3 , 4 , 5 , 6 , 7 , 8

❖ **Prompt 4**

➥ $\mathcal{Q} \mapsto$ 1 , 6 , 4 , 8 , 5 , 3 , 7 , 2

◈ $\mathcal{T} \mapsto$ First number is 1. Next we have 2, since 1 is less than 2. Next we have 3, since 2 is less than 3. Next we have 4, since 3 is less than 4. Next we have 5, since 4 is less than 5. Next we have 6, since 5 is less than 6. Next we have 7, since 6 is less than 7. Next we have 8, since 7 is less than 8.

◆ $\mathcal{A} \mapsto$ 1 , 2 , 3 , 4 , 5 , 6 , 7 , 8

❖ **Prompt 5**

➥ $\mathcal{Q} \mapsto$ 5 , 2 , 1 , 4 , 3 , 7

◈ $\mathcal{T} \mapsto$ First number is 1. Next we have 2, since 1 is less than 2. Next we have 3, since 2 is less than 3. Next we have 4, since 3 is less than 4. Next we have 5, since 4 is less than 5. Next we have 7, since 5 is less than 7.

◆ $\mathcal{A} \mapsto$ 1 , 2 , 3 , 4 , 5 , 7

❖ **Prompt 6**

➥ $\mathcal{Q} \mapsto$ 3 , 8 , 2 , 5 , 6 , 4 , 7 , 1

◈ $\mathcal{T} \mapsto$ First number is 1. Next we have 2, since 1 is less than 2. Next we have 3, since 2 is less than 3. Next we have 4, since 3 is less than 4. Next we have 5, since 4 is less than 5. Next we have 6, since 5 is less than 6. Next we have 7, since 6 is less than 7. Next we have 8, since 7 is less than 8.

◆ $\mathcal{A} \mapsto$ 1 , 2 , 3 , 4 , 5 , 6 , 7 , 8

❖ **Prompt 7**

➥ $\mathcal{Q} \mapsto$ 8 , 6 , 1 , 2 , 9 , 7 , 4

◈ $\mathcal{T} \mapsto$ First number is 1. Next we have 2, since 1 is less than 2. Next we have 4, since 2 is less than 4. Next we have 6, since 4 is less than 6. Next we have 7, since 6 is less than 7. Next we have 8, since 7 is less than 8. Next we have 9, since 8 is less than 9.

◆ $\mathcal{A} \mapsto$ 1 , 2 , 4 , 6 , 7 , 8 , 9

❖ **Prompt 8**

➥ $\mathcal{Q} \mapsto$ 7 , 6 , 8 , 1

◈ $\mathcal{T} \mapsto$ First number is 1. Next we have 6, since 1 is less than 6. Next we have 7, since 6 is less than 7. Next we have 8, since 7 is less than 8.

◆ $\mathcal{A} \mapsto$ 1 , 6 , 7 , 8

Table 69: GSM-8ᴋ with verbalized numbers.

**❖ Prompt 1**

➥ $\mathcal{Q}$ ↦ Shawn has five toys. For Christmas, he got two toys each from his mom and dad. How many toys does he have now?

◈ $\mathcal{T}$ ↦ Shawn started with five toys. If he got two toys each from his mom and dad, then that is four more toys. five + four = 9.

◆ $\mathcal{A}$ ↦ 9

**❖ Prompt 2**

➥ $\mathcal{Q}$ ↦ If there are three cars in the parking lot and two more cars arrive, how many cars are in the parking lot?

◈ $\mathcal{T}$ ↦ There are originally three cars. two more cars arrive. three + two = five.

◆ $\mathcal{A}$ ↦ 5

**❖ Prompt 3**

➥ $\mathcal{Q}$ ↦ Jason had twenty lollipops. He gave Denny some lollipops. Now Jason has twelve lollipops. How many lollipops did Jason give to Denny?

◈ $\mathcal{T}$ ↦ Jason started with twenty lollipops. Then he had twelve after giving some to Denny. So he gave Denny twenty - twelve = 8.

◆ $\mathcal{A}$ ↦ 8

**❖ Prompt 4**

➥ $\mathcal{Q}$ ↦ There were nine computers in the server room. Five more computers were installed each day, from Monday to Thursday. How many computers are now in the server room?

◈ $\mathcal{T}$ ↦ There were originally nine computers. For each of four days, five more computers were added. So five * four = twenty computers were added. nine + twenty is 29.

◆ $\mathcal{A}$ ↦ 29

**❖ Prompt 5**

➥ $\mathcal{Q}$ ↦ There are fifteen trees in the grove. Grove workers will plant trees in the grove today. After they are done, there will be twenty one trees. How many trees did the grove workers plant today?

◈ $\mathcal{T}$ ↦ There are fifteen trees originally. Then there were twenty one trees after some more were planted. So there must have been twenty one - fifteen = 6.

◆ $\mathcal{A}$ ↦ 6

**❖ Prompt 6**

➥ $\mathcal{Q}$ ↦ Leah had thirty two chocolates and her sister had 42. If they ate thirty five,how many pieces do they have left in total?

◈ $\mathcal{T}$ ↦ Originally, Leah had thirty two chocolates. Her sister had 42. So in total they had thirty two + forty two = 74. After eating thirty five, they had seventy four - thirty five = 39.

◆ $\mathcal{A}$ ↦ 39

**❖ Prompt 7**

➥ $\mathcal{Q}$ ↦ Olivia has twenty three. She bought five bagels for three dollars each. How much money does she have left?

◈ $\mathcal{T}$ ↦ Olivia had twenty three dollars. five bagels for three dollars each will be five x three = fifteen dollars. So she has twenty three - fifteen dollars left. twenty three - fifteen is 8.

◆ $\mathcal{A}$ ↦ 8

**❖ Prompt 8**

➥ $\mathcal{Q}$ ↦ Michael had fifty eight golf balls. On Tuesday, he lost twenty three golf balls. On Wednesday, he lost two more. How many golf balls did he have at the end of Wednesday?

◈ $\mathcal{T}$ ↦ Michael started with fifty eight golf balls. After losing twenty three on Tuesday, he had fifty eight - twenty three = 3five. After losing two more, he had thirty five - two = thirty three golf balls.

◆ $\mathcal{A}$ ↦ 33

Table 70: SPORTS with abstract sports person, sport, and sport activity.

❖ **Prompt 1**

➥ $\mathcal{Q}$ ↦ Is the following sentence plausible? "PERSON was involved in ACTIVITY."

❖ $\mathcal{T}$ ↦ PERSON is a SPORT1 player. Being ACTIVITY is part of SPORT1.

◆ $\mathcal{A}$ ↦ yes

❖ **Prompt 2**

➥ $\mathcal{Q}$ ↦ Is the following sentence plausible? "PERSON does ACTIVITY."

❖ $\mathcal{T}$ ↦ PERSON is a SPORT2 player. The ACTIVITY is part of SPORT3, not SPORT2.

◆ $\mathcal{A}$ ↦ no

❖ **Prompt 3**

➥ $\mathcal{Q}$ ↦ Is the following sentence plausible? "PERSON finishes ACTIVITY."

❖ $\mathcal{T}$ ↦ PERSON is a SPORT1 player. ACTIVITY is part of SPORT1.

◆ $\mathcal{A}$ ↦ yes

❖ **Prompt 4**

➥ $\mathcal{Q}$ ↦ Is the following sentence plausible? "PERSON performs ACTIVITY."

❖ $\mathcal{T}$ ↦ PERSON is a SPORT3 player. ACTIVITY is part of SPORT4, not SPORT3.

◆ $\mathcal{A}$ ↦ no

❖ **Prompt 5**

➥ $\mathcal{Q}$ ↦ Is the following sentence plausible? "PERSON was seen doing ACTIVITY."

❖ $\mathcal{T}$ ↦ PERSON is a SPORT4 player. Being ACTIVITY is part of SPORT4.

◆ $\mathcal{A}$ ↦ yes

❖ **Prompt 6**

➥ $\mathcal{Q}$ ↦ Is the following sentence plausible? "PERSON complete ACTIVITY."

❖ $\mathcal{T}$ ↦ PERSON is an SPORT3 player. ACTIVITY is part of SPORT1, not SPORT3.

◆ $\mathcal{A}$ ↦ no

❖ **Prompt 7**

➥ $\mathcal{Q}$ ↦ Is the following sentence plausible? "PERSON executes ACTIVITY."

❖ $\mathcal{T}$ ↦ PERSON is a SPORT1 player. ACTIVITY is part of SPORT1.

◆ $\mathcal{A}$ ↦ yes

❖ **Prompt 8**

➥ $\mathcal{Q}$ ↦ Is the following sentence plausible? "PERSON carries out ACTIVITY."

❖ $\mathcal{T}$ ↦ PERSON is an SPORT1 player. ACTIVITY is part of SPORT3, not SPORT1.

◆ $\mathcal{A}$ ↦ no

Table 71: SPORTS with abstract sports person and sport.

---

❖ **Prompt 1**

➥ $\mathcal{Q} \mapsto$ Is the following sentence plausible? "PERSON was perfect from the line."

◈ $\mathcal{T} \mapsto$ PERSON is a SPORT1 player. Being perfect from the line is part of SPORT1.

◆ $\mathcal{A} \mapsto$ yes

---

❖ **Prompt 2**

➥ $\mathcal{Q} \mapsto$ Is the following sentence plausible? "PERSON caught the screen pass in the NFC championship."

◈ $\mathcal{T} \mapsto$ PERSON is a SPORT3 player. The NFC championship is part of SPORT2, not SPORT3.

◆ $\mathcal{A} \mapsto$ no

---

❖ **Prompt 3**

➥ $\mathcal{Q} \mapsto$ Is the following sentence plausible? "PERSON beat the buzzer."

◈ $\mathcal{T} \mapsto$ PERSON is a SPORT1 player. Beating the buzzer is part of SPORT1.

◆ $\mathcal{A} \mapsto$ yes

---

❖ **Prompt 4**

➥ $\mathcal{Q} \mapsto$ Is the following sentence plausible? "PERSON passed the puck."

◈ $\mathcal{T} \mapsto$ PERSON is a SPORT2 player. Passing the puck is part of SPORT4, not SPORT2.

◆ $\mathcal{A} \mapsto$ no

---

❖ **Prompt 5**

➥ $\mathcal{Q} \mapsto$ Is the following sentence plausible? "PERSON was called for slashing."

◈ $\mathcal{T} \mapsto$ PERSON is a SPORT4 player. Being called for slashing is part of SPORT4.

◆ $\mathcal{A} \mapsto$ yes

---

❖ **Prompt 6**

➥ $\mathcal{Q} \mapsto$ Is the following sentence plausible? "PERSON set the pick and roll."

◈ $\mathcal{T} \mapsto$ PERSON is an SPORT2 player. Pick and roll is part of SPORT1, not SPORT2.

◆ $\mathcal{A} \mapsto$ no

---

❖ **Prompt 7**

➥ $\mathcal{Q} \mapsto$ Is the following sentence plausible? "PERSON banked the shot in."

◈ $\mathcal{T} \mapsto$ PERSON is a SPORT1 player. Banking the shot in is part of SPORT1.

◆ $\mathcal{A} \mapsto$ yes

---

❖ **Prompt 8**

➥ $\mathcal{Q} \mapsto$ Is the following sentence plausible? "PERSON threw a touchdown."

◈ $\mathcal{T} \mapsto$ PERSON is an SPORT1 player. Throwing a touchdown is part of SPORT2, not SPORT1.

◆ $\mathcal{A} \mapsto$ no

---

Table 72: SPORTS with the order of clauses switched.

---

❖ **Prompt 1**

➥ $\mathcal{Q} \mapsto$ Is the following sentence plausible? "Jamel Murray was perfect from the line."

❖ $\mathcal{T} \mapsto$ Being perfect from the line is part of basketball. Jamal Murray is a basketball player.

◆ $\mathcal{A} \mapsto$ yes

❖ **Prompt 2**

➥ $\mathcal{Q} \mapsto$ Is the following sentence plausible? "Joao Moutinho caught the screen pass in the NFC championship."

❖ $\mathcal{T} \mapsto$ The NFC championship is part of American football, not soccer. Joao Moutinho is a soccer player.

◆ $\mathcal{A} \mapsto$ no

❖ **Prompt 3**

➥ $\mathcal{Q} \mapsto$ Is the following sentence plausible? "Jonas Valanciunas beat the buzzer."

❖ $\mathcal{T} \mapsto$ Beating the buzzer is part of basketball. Jonas Valanciunas is a basketball player.

◆ $\mathcal{A} \mapsto$ yes

❖ **Prompt 4**

➥ $\mathcal{Q} \mapsto$ Is the following sentence plausible? "Sam Darnold passed the puck."

❖ $\mathcal{T} \mapsto$ Passing the puck is part of hockey, not American football. Sam Darnold is a American football player.

◆ $\mathcal{A} \mapsto$ no

❖ **Prompt 5**

➥ $\mathcal{Q} \mapsto$ Is the following sentence plausible? "Kyle Palmieri was called for slashing."

❖ $\mathcal{T} \mapsto$ Being called for slashing is part of hockey. Kyle Palmieri is a hockey player.

◆ $\mathcal{A} \mapsto$ yes

❖ **Prompt 6**

➥ $\mathcal{Q} \mapsto$ Is the following sentence plausible? "Carson Wentz set the pick and roll."

❖ $\mathcal{T} \mapsto$ Pick and roll is part of basketball, not football. Carson Wentz is an American football player.

◆ $\mathcal{A} \mapsto$ no

❖ **Prompt 7**

➥ $\mathcal{Q} \mapsto$ Is the following sentence plausible? "Malcolm Brogdon banked the shot in."

❖ $\mathcal{T} \mapsto$ Banking the shot in is part of basketball. Malcolm Brogdon is a basketball player.

◆ $\mathcal{A} \mapsto$ yes

❖ **Prompt 8**

➥ $\mathcal{Q} \mapsto$ Is the following sentence plausible? "Draymond Green threw a touchdown."

❖ $\mathcal{T} \mapsto$ Throwing a touchdown is part of football, not basketball. Draymond Green is an basketball player.

◆ $\mathcal{A} \mapsto$ no

Table 73: CCoT prompt for GSM-8K.

---

❖ **Prompt 1**

➥ $\mathcal{Q} \mapsto$ What is fifteen more than a quarter of 48?

◈ $\mathcal{T} \mapsto$ A quarter of 48 is 48 / 4 = 12. 15 more than 12 is 12 + 15 = 27.

◆ $\mathcal{A} \mapsto$ 27

---

❖ **Prompt 2**

➥ $\mathcal{Q} \mapsto$ Twice Angie's age, plus 4, is 20. How old is Angie?

◈ $\mathcal{T} \mapsto$ Angie's age minus 4 is 20 - 4 = 16. 16 is twice Angie's age. Thus, Angie's age is 16 / 2 = 8.

◆ $\mathcal{A} \mapsto$ 8

---

❖ **Prompt 3**

➥ $\mathcal{Q} \mapsto$ Steve is 5'6". He grows 6 inches. How tall is he in inches?

◈ $\mathcal{T} \mapsto$ One feet has 12 inches. Initially, Steve is 5 * 12 + 6 = 66 inches tall. After growing 6 inches, Steve is 66 + 6 = 72 inches tall.

◆ $\mathcal{A} \mapsto$ 72

---

❖ **Prompt 4**

➥ $\mathcal{Q} \mapsto$ 198 passengers fit into 9 buses. How many passengers fit in 5 buses?

◈ $\mathcal{T} \mapsto$ Capacity of one bus is 198 passengers / 9 buses = 22 passengers in one bus. Thus, 5 buses can fit 22 * 5 = 110 passengers.

◆ $\mathcal{A} \mapsto$ 110

---

❖ **Prompt 5**

➥ $\mathcal{Q} \mapsto$ Fifteen more than a quarter of a number is 27. What is the number?

◈ $\mathcal{T} \mapsto$ Fifteen less than 27 is 27 - 15 = 12. The quarter of the number is thus 12, and the number is 12 * 4 = 48.

◆ $\mathcal{A} \mapsto$ 48

---

❖ **Prompt 6**

➥ $\mathcal{Q} \mapsto$ If 12 bags of oranges weigh 24 pounds, how much do 8 bags weigh?

◈ $\mathcal{T} \mapsto$ 12 bags weight 24 pounds, so one bag weights 24 pounds / 12 = 2 pounds. So 8 bags of oranges would weigh 8 * 2 pounds = 16 pounds.

◆ $\mathcal{A} \mapsto$ 16

---

❖ **Prompt 7**

➥ $\mathcal{Q} \mapsto$ Sandra eats 3 beignets every morning. How many beignets will she eat in 16 weeks?

◈ $\mathcal{T} \mapsto$ Sandra eats 3 beignets every morning and there are 7 days in a week so she eats 3 * 7 = 21 beignets in a week. Sandra eats 21 beignets in a week, so in 16 weeks she will eat 21 * 16 = 336 beignets.

◆ $\mathcal{A} \mapsto$ 336

---

❖ **Prompt 8**

➥ $\mathcal{Q} \mapsto$ John takes a pill every 6 hours. How many pills does he take a week?

◈ $\mathcal{T} \mapsto$ There are 24 hours in a day. So John takes 24 / 6 = 4 pills a day. In a week, John therefore takes 4 * 7 = 28 pills.

◆ $\mathcal{A} \mapsto$ 28

---

Table 74: CCoT prompt for DATE.

---

❖**Prompt 1**

➥ $\mathcal{Q}$ ↦ It is 4/19/1969 today. What is the date 24 hours later in MM/DD/YYYY?

◆ $\mathcal{T}$ ↦ Today is 04/19/1969. 24 hours (one day) later is 04/20/1969.

◆ $\mathcal{A}$ ↦ 04/20/1969

❖**Prompt 2**

➥ $\mathcal{Q}$ ↦ The concert was scheduled to be on 06/01/1943, but was delayed by one day to today. What is the date 10 days ago in MM/DD/YYYY?

◆ $\mathcal{T}$ ↦ Today is 06/02/1943 (one day after 06/01/1943). 10 days before today is 05/23/1943.

◆ $\mathcal{A}$ ↦ 05/23/1943

❖**Prompt 3**

➥ $\mathcal{Q}$ ↦ The first day of 2019 is a Tuesday, and today is the first Monday of 2019. What is the date today in MM/DD/YYYY?

◆ $\mathcal{T}$ ↦ 01/01/2019 was a Tuesday (first day of 2019). Today is the first Monday, 01/07/2019. (six days later).

◆ $\mathcal{A}$ ↦ 01/07/2019

❖**Prompt 4**

➥ $\mathcal{Q}$ ↦ Jane was born on the last day of February in 2001. Today is her 16-year-old birthday. What is the date yesterday in MM/DD/YYYY?

◆ $\mathcal{T}$ ↦ Jane was born on 02/28/2001. So today is 02/28/2017 and yesterday was 02/27/2017.

◆ $\mathcal{A}$ ↦ 02/27/2017

❖**Prompt 5**

➥ $\mathcal{Q}$ ↦ 2015 is coming in 36 hours. What is the date one week from today in MM/DD/YYYY?

◆ $\mathcal{T}$ ↦ 2015 is coming in 2 days (36 hours). So today is 12/30/2021, and one week from today will be 01/05/2015.

◆ $\mathcal{A}$ ↦ 01/05/2015

❖**Prompt 6**

➥ $\mathcal{Q}$ ↦ Jane thought today is 3/11/2002, but today is in fact Mar 12, which is 1 day later. What is the date 24 hours later in MM/DD/YYYY?

◆ $\mathcal{T}$ ↦ Today is 03/12/2002. So the date 24 hours later will be 03/13/2002.

◆ $\mathcal{A}$ ↦ 03/13/2002

---

Table 75: CCoT prompt for SPORTS.

---

❖**Prompt 1**

➥ $\mathcal{Q} \mapsto$ Is the following sentence plausible? "Jamel Murray was perfect from the line."

◈ $\mathcal{T} \mapsto$ Jamal Murray -> basketball. perfect from the line -> basketball.

◆ $\mathcal{A} \mapsto$ yes

❖**Prompt 2**

➥ $\mathcal{Q} \mapsto$ Is the following sentence plausible? "Joao Moutinho caught the screen pass in the NFC championship."

◈ $\mathcal{T} \mapsto$ Joao Moutinho -> soccer. NFC championship -> American football.

◆ $\mathcal{A} \mapsto$ no

❖**Prompt 3**

➥ $\mathcal{Q} \mapsto$ Is the following sentence plausible? "Jonas Valanciunas beat the buzzer."

◈ $\mathcal{T} \mapsto$ Jonas Valanciunas -> basketball. beating the buzzer -> basketball.

◆ $\mathcal{A} \mapsto$ yes

❖**Prompt 4**

➥ $\mathcal{Q} \mapsto$ Is the following sentence plausible? "Sam Darnold passed the puck."

◈ $\mathcal{T} \mapsto$ Sam Darnold -> American football. passing the puck -> hockey.

◆ $\mathcal{A} \mapsto$ no

❖**Prompt 5**

➥ $\mathcal{Q} \mapsto$ Is the following sentence plausible? "Kyle Palmieri was called for slashing."

◈ $\mathcal{T} \mapsto$ Kyle Palmieri -> hockey. called for slashing -> hockey.

◆ $\mathcal{A} \mapsto$ yes

❖**Prompt 6**

➥ $\mathcal{Q} \mapsto$ Is the following sentence plausible? "Carson Wentz set the pick and roll."

◈ $\mathcal{T} \mapsto$ Carson Wentz is -> American football. pick and roll -> basketball.

◆ $\mathcal{A} \mapsto$ no

❖**Prompt 7**

➥ $\mathcal{Q} \mapsto$ Is the following sentence plausible? Malcolm Brogdon banked the shot in.

◈ $\mathcal{T} \mapsto$ th

➥ $\mathcal{Q} \mapsto$ Is the following sentence plausible? "Malcolm Brogdon banked the shot in."

◈ $\mathcal{T} \mapsto$ Malcolm Brogdon -> basketball. banking the shot in -> basketball.

◆ $\mathcal{A} \mapsto$ yes

❖**Prompt 8**

➥ $\mathcal{Q} \mapsto$ Is the following sentence plausible? "Draymond Green threw a touchdown."

◈ $\mathcal{T} \mapsto$ Draymond Green -> basketball. throwing a touchdown -> football.

◆ $\mathcal{A} \mapsto$ no