# OpenReview forum: "What Makes Chain-of-Thought Prompting Effective? A Counterfactual Study"
_EMNLP/2023/Conference — EMNLP 2023 Findings_

### Official Review · Reviewer_jgj8 · 2023-08-03

**Soundness:** 3

**Excitement:**

4: Strong: This paper deepens the understanding of some phenomenon or lowers the barriers to an existing research direction.

**Paper Topic And Main Contributions:**

This paper focuses on Chain-of-thought prompting (CoT) technique of large language models (LLMs) and aims to analyze what makes CoT prompting effective. The main contributions lie in that the authors systematically explore how elements such as symbols (digits, entities) and patterns (equations, sentence structure) in CoT prompt affect model performance by a counterfactual prompting approach. Based on mathematical tasks (GSM-8K), commonsense tasks (date and sports understanding), and symbolic tasks (SORTING), some fine-grained conclusions of how symbols and patterns elicit CoT performance are founded.

**Reasons To Accept:**

1.	Beyond the prior works, this paper systematically and comprehensively explores the influence of fined-grained elements (symbols and patterns) on CoT effectiveness.
2.	The findings of this paper provides valuable insights into the behavior of language models  and will benefit the research on CoT mechanism.
3.	This paper provides sufficient experimental results for reference.


**Reasons To Reject:**

1.	The findings of this paper are dependent on specific tasks and maybe cannot be transferred to other tasks or scenarios.
2.	The paper conducts an empirical study but lacks good summarization to instruct the design of CoT prompt.

**Reproducibility:**

2: Would be hard pressed to reproduce the results. The contribution depends on data that are simply not available outside the author's institution or consortium; not enough details are provided.

**Reviewer Confidence:**

4: Quite sure. I tried to check the important points carefully. It's unlikely, though conceivable, that I missed something that should affect my ratings.

---

> ### Author Rebuttal · Authors · 2023-08-29
>
> Thank you for taking the time to review our paper! We were delighted to read that you recognized the main contributions of our work, especially our systematic exploration into the fined-grained elements in the CoT prompting and how they affect model performance.
>
> We think that all your questions are addressable within this discussion period. Please see our response below. We would be more than happy to address additional questions during the discussion period if anything is unclear.
>
> ---
>
> **Task-specific findings may not generalize to other scenarios.**
>
> Certainly, we agree that findings from specific tasks may not universally generalize. Additionally, for some tasks where differentiating between symbols and patterns is less clear-cut, it might be challenging to confidently assert their individual roles. Our findings from the four tasks have generally led to a few consistent conclusions, particularly regarding the role of CoT in reminding the model of the task. Nevertheless, to evaluate the breadth of our observations, we extended our experiments to the strategy QA task.  The results from `code-davinci-002` are as follows:
>
> | Example Type        | Question                                                                   | Thought                                                                                               | Answer |  Accuracy |
> |---------------------|----------------------------------------------------------------------------|-------------------------------------------------------------------------------------------------------|--------|-------------------|
> | Normal              | "Do hamsters provide food for any animals?"                                | "Hamsters are prey animals. Prey are food for predators. Thus, hamsters provide food for some animals." | "yes"  | 67.837            |
> | Abstracted          | "Do ANIMAL_1 provide food for any ANIMAL_2?"                               | "ANIMAL_1 are PREY ANIMAL_2. PREY are food for predators. Thus, ANIMAL_1 provide food for some ANIMAL_2." | "yes"  | 69.943            |
> | Wrong               | "Do hamsters provide food for any animals?"                                | "Hamsters are domestic animals. Domestic animals are pets, not food. Thus, hamsters provide food for some animals." | "yes"  | 71.215            |
>
>
> The results highlight a consistent trend: prompts utilizing abstract symbols or containing incorrect information can achieve performance levels comparable to, or even surpassing, standard CoT prompts.
>
> ---
>
> **Can we derive some guidelines from this empirical study for CoT prompt design?**
>
>
> A central observation of our study is that when few-shot examples effectively convey the task, LLMs can successfully harness them. Moreover, a consistent pattern within the prompts that aligns with the task significantly enhances the model's ability to generate the correct response.
>
> This led us to develop the Concise CoT (CCoT) prompts. We derived CCoT prompts by trimming redundant tokens from examples. For instance, in the SPORTS category, a thought such as *Jamal Murray is a basketball player. Being perfect from the line is part of basketball* was streamlined to *Jamal Murray → basketball. perfect from the line → basketball.* Similarly, in DATE, *Today is 04/19/1969. 24 hours later is one day after today, which would be 04/20/1969* was converted to *Today is 04/19/1969. 24 hours (one day) later is 04/20/1969.* For GSM-8k, we opted for the shortest examples from its training set.
>
> Our results, summarized in the table below, indicate that Concise CoT not only improves the accuracy, but also cuts down the number of input and output tokens by 1.39$\times$ and 1.58$\times$, respectively.
>
> |              | Type                   | GSM-8K   | Date    | Sports  |
> |--------------|------------------------|---------|---------|---------|
> | GPT-3        | CoT                    | 46.9    | 54.1    | 63.5    |
> |              | Concise CoT (Ours)     | **52.4**| **61.1**| **74.5**|
> | PaLM-62B     | CoT                    | 27.4    | 44.7    | 93.7    |
> |              | Concise CoT (Ours)     | **29.1**| **51.3**| **94.6**|
> | PaLM-540B    | CoT                    | 53.2    | 65.3    | 95.4    |
> |              | Concise CoT (Ours)     | **56.2**| **69.1**| **97.4**|
> | Codex        | CoT                    | **65.6**    | 69.2    | 98.3    |
> |              | Concise CoT (Ours)     | 62.5 | **69.9**| **98.5**|
> | LLAMA2-70B   | CoT                    | 62.17   | 71.83   | **98.26**   |
> |              | Concise CoT (Ours)     | **64.8**| **74.72**|  98.16 |
> | Average      | CoT                    | 51.05  | 61.03  | 89.83  |
> |              | CCoT (Ours)            | **53.0**  | **65.22**| **92.63**|
>
> We hypothesize that the concise nature of these prompts might reduce noise during generation, leading to an improved task solve rate.

---

### Official Review · Reviewer_xMnP · 2023-08-05

**Soundness:** 3

**Excitement:**

3: Ambivalent: It has merits (e.g., it reports state-of-the-art results, the idea is nice), but there are key weaknesses (e.g., it describes incremental work), and it can significantly benefit from another round of revision. However, I won't object to accepting it if my co-reviewers champion it.

**Paper Topic And Main Contributions:**

This paper presents a series of empirical analyses for studying what makes chain-of-thought reasoning effective. Specifically, the paper mainly focuses on studying the impacts of "symbols" and "patterns" via prompting LLMs with counterfactual prompts (where symbols and patterns are manipulated). Based on the experiments on several reasoning datasets, the main findings include that symbols used in the prompt have a limited impact on the model’s performance, and consistent patterns, on the other hand, are more crucial.

**Reasons To Accept:**


The paper studies an interesting and important problem

The paper includes a thorough evaluation, covering a number of settings (manipulated prompts) and language models. The paper also provides extensive details in the appendix

The paper is mostly easy to read.

**Reasons To Reject:**

The definition of the two core concepts, “symbol” and “pattern”, seems not to be very well defined. E.g., the pattern is defined as “a template that is present in all examples within a given task” (line 302). It is unclear to me what exactly does template mean. From the examples, the pattern for SPORTS task looks like a text template (some text with slots to be filled); the pattern for DATE looks like a high-level procedure; the pattern for GSM and SORT seems like a constraint over the generations. It would be good if the pattern could be explained more clearly. Symbols seem to be a bit more intuitive to understand, but the description “token sequences in the prompt which the model employs to reason” and the authors “adopt natural definitions of the symbols” for each task also seem a bit vague.

And symbols and patterns seem to be intertwined. E.g., for the sorting task, does using abstract symbol break pattern as well (assuming the Greek letters don’t have an order)?

---

Some of the claims in the paper seem to be vague and could be clearer. While it is helpful to see summarized takeaways, takeaway two describes “COT helps in eliciting commonsense knowledge”, this is only backed up by qualitative examples and observations.

Another example is in the abstract. The abstract and intro states “specifying text in style frequently found on the web are crucial”, but the main paper does not provide enough details on what are more “frequent” patterns on the web.

---

More discussion on the differences between this paper and related work would be helpful. There have been similar counterfactual probing experiments and similar findings (Wang et al., 2022; Ye et al., 2022). The paper briefly discusses its relation to past work in Section 7. In particular, the paper states "we manipulate various aspects of symbols and patterns to examine their role in the success" (line 534), while Wang et al. (2022) and Ye et al. (2022) also contains similar experiments that perturbing CoTs in multiples ways. (E.g., there are also pattern only experiments and inconsistent pattern experiments in Ye et al. (2022)). It would be helpful to clearly highlight the key differences and new findings




**Reproducibility:**

4: Could mostly reproduce the results, but there may be some variation because of sample variance or minor variations in their interpretation of the protocol or method.

**Reviewer Confidence:**

4: Quite sure. I tried to check the important points carefully. It's unlikely, though conceivable, that I missed something that should affect my ratings.

---

> ### Author Rebuttal · Authors · 2023-08-29
>
> Thank you for taking the time to review our paper! We were happy to read that you found our study on the impacts of symbols and patterns to be both interesting and important.  We also appreciate your recognition of the depth of our evaluation.
>
>
> We think that all your questions are addressable within this discussion period. Please see our response below. We would be more than happy to address additional questions during the discussion period if anything is unclear.
>
> ---
>
>
> **Clarify definitions of "symbol" and "pattern" and their interplay.**
>
> - The term "template" in line 302 intended to convey the structural or rule-based patterns for a task. It could manifest differently: text template for SPORTS, procedural rule for DATE, or constraints for GSM and SORT. Essentially, patterns guide the task.
>
> - Symbols consistently represent specific, tangible entities or values—be it players, sports, or date elements.
>
> - L1075 (Section *Are the definitions of symbols and patterns universal?* - Appendix-Page 14) clarifies our definitions of symbols and patterns are not intended to be universal. We recognize there could be alternative characterizations for these components. Rather than aiming for universality, our study sought to make practical distinctions to understand how manipulating one component affects outcomes while others remain constant. While there isn't a universal definition of symbols and patterns, for any *given* task we have aimed to provide a clear and reasonable characterization.
>
> We will add further clarifications in the next version.
>
> ---
>
> **Sorting task/patterns and symbol intertwined?**
>
> In the sorting tasks, even when using abstract symbols, we ensure that the order presented in the "thought" and "answer" sections is consistent. This consistent order is designed to signal to the model that symbols from the "question" need to be arranged in an increasing order and then emitted in the same sequence as the answer.
>
> ```py
> Example(
>     question="μ , ζ , π , χ , λ , ς , α , γ",
>     thought="α < π < γ < λ < ζ < χ < μ < ς",
>     answer="α , π , γ , λ , ζ , χ , μ , ς",
> )
>
> ```
> Further, following your suggestion, we experimented with sorting where the thought and the answer are arranged based on the order of the Greek alphabet, and found no difference in performance using code-davinci-002.
>
> ---
>
> **Clarify "COT helps in eliciting commonsense knowledge"**
>
> - We agree that qualitative analysis is an important part of our work. To the extent possible, we have tried to be thorough in the qualitative analysis. While space did not allow us to add more examples in the main text, we have added details for analyzing how CoT helps in eliciting commonsense knowledge in Appendix I.2. Specifically, Table 15 presents examples from GSM-8k and DATE Reasoning tasks, and Appendix I.2 discusses these examples in detail.
>
> - Beyond qualitative analysis, we conducted a quantitative study on the sports reasoning dataset, as highlighted in Page-31-Table 17, and Pg-29-L-1555 (“Extracting rare commonsense knowledge.”). Briefly, we find that in instances where CoT proved uniquely effective—being correct when counterfactual prompts were not—the subjects in the question (sports persons and activities) were often more obscure. Specifically, while entities in the SPORTS dataset average a popularity of 399k in Google search results, those uniquely solved by CoT averaged at just 68k, making them roughly 5$\times$ rarer.
>
> **Infrequent web patterns/main paper does not provide enough details on what are more “frequent” patterns on the web**
>
>
> We elaborate on this in Pg-6-L426 (*Text with altered grammatical style*). Specifically, we demonstrate how using Yodish, an infrequent web style of English ~(Kaminski, 2011; Pullum, 2005), as the "thought" in chain-of-thought consistently reduces the solve rate across all tasks (Table 4).  We will make this connection clearer in the next version.
>
> ---
>
> **Highlight key differences from Wang et al. and Ye et al.'s work.**
>
> Our work is concurrent with Wang et al. (2022) and Ye et al. (2022), with our Arxiv preprint slightly predating those papers; we feel that this body of work is nicely mutually reinforcing. In contrast to these papers, one of the key contributions of our work is defining the  two main components (e.g. Symbols and Patterns) in CoT and designing a series of **counterfactual** studies to assess the effectiveness of each component, across four tasks with different characteristics. In addition, we apply surface-level manipulations to text, including altered grammatical style and shuffled text (Section 5). Finally, different from both these works, we additionally explore the behavior of attention patterns under various counterfactual prompts (L285/Figure 2, Pg 18, Appendix G).  In details, these are the main differences between our work and prior work:
>
> (1) Wang et al. (2022) focuses on evaluating the effectiveness of CoT for two tasks: (a) arithmetic reasoning and (b) multi-hop factual question answering. Wang et al. (2022) rather apply drastic changes to sketch invalid reasoning (See Wang et al. (2022)-Section 4.1) in CoT; hence, their study is *not* a counterfactual study (as stated by the authors). They claim that these drastic changes offer “...little valid reasoning exists to help solve the query”. In contrast, our work focuses on counterfactual studies that entail both minimal changes and drastic changes to CoT. Our work evaluates the effectiveness of CoT through counterfactuals in two cases: (1) the flow of reasoning exists but with incorrect symbols and patterns and (2) the flow of reasoning is deliberately eliminated as in our shuffled and random text (Page 7) study.
>
> (2) Ye et al. (2022) studies the impact of incorrect calculations and masking/dropping some words in CoT across three tasks. In addition to these studies, our work expands into using OOD symbols, inconsistent patterns, using symbols or patterns exclusively, different grammatical styles, etc.
>
> We will clarify these differences in the next version.

---

### Official Review · Reviewer_sLkz · 2023-08-05

**Soundness:** 4

**Excitement:**

4: Strong: This paper deepens the understanding of some phenomenon or lowers the barriers to an existing research direction.

**Paper Topic And Main Contributions:**

This work presents experiments to better understand why chain-of-thought (CoT) prompting works or does not work. The authors do this by counterfactual prompting where only one aspect of the CoT prompt is changed (or ablated) to understand its importance in the original prompt. In particular, they present such ablation experiments on various tasks for 3 aspects of the prompts: (i) Symbols (numbers, dates, named entities, etc) (ii) patterns (equations, sentence structure, etc) (iii) surface features (grammatical style).

The authors find that in some cases changing the prompt substantially, like replacing numbers with Greek letters, doesn't change the performance, and in some cases like sorting it can even help. But at the same time, many of these ablations also hurt the performance. Their overall takeaway is that ablation will hurt performance when it hinders understanding of what a task is. E.g., replacing numbers with Greek letters in MWP doesn't hinder understanding, but replacing players with random players in a task about checking if a sports person and activity are related, does hinder task understanding. Overall they argue CoT helps remind the model of the relevant skill/knowledge as opposed to learning it in context.

**Questions For The Authors:**

A. In the ablation where you replace numbers with greek letters: Do the models generate greek letters at test time, and yet generate the correct answer. Or they generate usual numbers. Same question for other ablations you have explored as well. It'd be useful to clarify this in the paper.

**Reasons To Accept:**

- This work explores an important problem of when/why CoT prompting works.
- This work does a thorough analysis of the kinds of ablations in CoT prompts and their effects.
- Past works have explored the reliability of CoT based on its correctness, but this work takes it further by isolating feature-based changes giving a more fine-grained analysis.
- The paper is well written.

**Reasons To Reject:**

I don't see any weakness strong enough to reject this work. It is a good analysis work.

**Reproducibility:**

4: Could mostly reproduce the results, but there may be some variation because of sample variance or minor variations in their interpretation of the protocol or method.

**Reviewer Confidence:**

4: Quite sure. I tried to check the important points carefully. It's unlikely, though conceivable, that I missed something that should affect my ratings.

---

> ### Author Rebuttal · Authors · 2023-08-29
>
> Thank you for taking the time to review our paper! We were pleased to see that you found value in our exploration of CoT prompting and our approach using counterfactual ablations.  Additionally, we are grateful for your acknowledgment of our attempt to provide a fine-grained analysis beyond what has been explored in previous works.
>
> We think that all your questions are addressable within this discussion period. Please see our response below. We would be more than happy to address additional questions during the discussion period if anything is unclear.
>
> ---
>
> **Q: Do the models generate greek letters at test time, and yet generate the correct answer. Or they generate usual numbers. Same question for other ablations you have explored as well.**
>
> We analyze the presence of abstract symbols in the generated output, and find that a large majority does not contain these symbols, however, the exact fraction is dataset dependent:
>
>
> | Dataset | Abstract Symbol Present in generated answer (Yes/No) | Percentage |
> |---------|------------------------------|------------|
> | GSM-8K | Yes               | 12.91%     |
> |                       | No              | 87.09%     |
> | DATE | Yes             | 33.68%     |
> |                       | No              | 66.32%     |
> | SPORTS | Yes | 0.00%      |
> |                       | No              | 100.00%    |
>
> Specifically, outputs for  `GSM-8K` contain abstract symbols (Greek alphabets) in 12.91% of the cases, whereas abstract symbols (placeholders for names and activities) are not present in the output for  `SPORTS Reasoning`.   In cases where the model does generate Greek alphabet symbols, the model regardless reaches the correct answer in 36.5% of the cases.
>
> We will add this analysis in the next version.

---

### Meta-Review · Area_Chair_jkgi · 2023-09-16

**Recommendation:** 3

**Metareview:**

This paper aims to analyze what makes Chain-of-Thought prompting (CoT) prompting of large language models (LLMs) effective. The authors explore how elements such as "symbols" (e.g., digits, entities) and "patterns" (e.g., equations, sentence structure) in CoT prompt affect model performance with counterfactual analysis (where symbols and patterns are manipulated). In general, the reviewers think the work studies an important problem of when/why CoT prompting works, conducts comprehensive experiments, and could be valuable for research on CoT mechanism. They still have remaining concerns such as the definitions of "symbols" and "patterns" being vague. There is also very related work such as Wang et al. (2022) and Ye et al. (2022) that has similar findings, from which the authors need to clarify the distinction (although the work appears concurrently on arXiv, the publication date is way earlier than the EMNLP deadline). Overall, I'd recommend accepting this work to Findings.

---

### Decision · Program_Chairs · 2023-10-07

**Decision:**

Accept-Findings

**Comment:**

This paper aims to analyze what makes Chain-of-Thought prompting (CoT) prompting of large language models (LLMs) effective. The authors explore how elements such as "symbols" (e.g., digits, entities) and "patterns" (e.g., equations, sentence structure) in CoT prompt affect model performance with counterfactual analysis (where symbols and patterns are manipulated). In general, the reviewers think the work studies an important problem of when/why CoT prompting works, conducts comprehensive experiments, and could be valuable for research on CoT mechanism. They still have remaining concerns such as the definitions of "symbols" and "patterns" being vague. There is also very related work such as Wang et al. (2022) and Ye et al. (2022) that has similar findings, from which the authors need to clarify the distinction (although the work appears concurrently on arXiv, the publication date is way earlier than the EMNLP deadline). Overall, I'd recommend accepting this work to Findings.